## OPEN

# Actin remodelling controls proteasome homeostasis upon stress

Thomas David Williams, Roberta Cacioppo, Alexander Agrotis, Ailsa Black, Houjiang Zhou and Adrien Rousseau ⬡ ✉

When cells are stressed, bulk translation is often downregulated to reduce energy demands while stress-response proteins are simultaneously upregulated. To promote proteasome assembly and activity and maintain cell viability upon TORC1 inhibition, 19S regulatory-particle assembly chaperones (RPACs) are selectively translated. However, the molecular mechanism for such selective translational upregulation is unclear. Here, using yeast, we discover that remodelling of the actin cytoskeleton is important for RPAC translation following TORC1 inhibition. mRNA of the RPAC ADC17 is associated with actin cables and is enriched at cortical actin patches under stress, dependent upon the early endocytic protein Ede1. *ede1Δ* cells failed to induce RPACs and proteasome assembly upon TORC1 inhibition. Conversely, artificially tethering *ADC17* mRNA to cortical actin patches enhanced its translation upon stress. These findings suggest that actin-dense structures such as cortical actin patches may serve as a translation platform for a subset of stress-induced mRNAs including regulators of proteasome homeostasis.

Cells require the right amount of each protein to be in the right place at the right time. This intimate balance between protein synthesis, folding, modification and degradation is known as protein homeostasis or proteostasis. Failure to maintain proteostasis causes a build-up of misfolded proteins, and potentially toxic aggregates, including those that cause neurodegenerative diseases[1–3].

Misfolded, damaged and short-lived proteins are degraded by one of two mechanisms: the autophagy–lysosome system or the ubiquitin–proteasome system (UPS)[4,5]. In autophagy, autophagosomes are assembled around proteins or organelles to be degraded, then delivered to the lysosome[5]. With the UPS, proteins are tagged with ubiquitin conjugates that serve as a recognition signal for proteasomal degradation. The proteasome is a large, multiprotein complex comprising a 'core particle' (CP), containing the proteolytic activity of the proteasome, and one or two 'regulatory particles' (RPs). The RP recognizes ubiquitinated proteins and catalyses their unfolding and translocation into the CP, where they are degraded[6,7].

Under optimal energy and nutrient conditions, protein synthesis exceeds degradation to achieve cell growth and proliferation. The TORC1 kinase complex (in yeast, mTORC1 in mammals) is active and promotes anabolic processes such as protein, nucleotide and lipid synthesis[8–10]. Concurrently, TORC1 restricts autophagy induction and proteasomal degradation[11–13]. When nutrients are limited, or certain stresses are applied to cells, TORC1 is inactivated[9,10]. TORC1 inactivation reduces bulk protein synthesis and increases protein degradation. Autophagy is de-repressed, while proteasomal degradation is increased. Enhanced protein degradation capacity allows cells to rapidly degrade misfolded and damaged proteins while simultaneously generating a pool of intracellular amino acids for stress-adaptive protein synthesis[14,15]. While the mechanism of TORC1 autophagy regulation has been well described, its regulation of proteasome function is much less clear. TORC1 inactivation in budding yeast leads to activation of the MAP kinase Mpk1 (also known as Slt2, ERK5 in mammals)[13,16]. Activated Mpk1 is important to induce the translation of proteasome RP assembly chaperones (RPACs) such as Adc17 and Nas6, increasing RP assembly to produce more functional proteasomes[13]. CP assembly is also increased

upon TORC1 inhibition with two CP assembly chaperones, Pba1 and Pba2, being induced. Unlike RPAC induction, CP assembly chaperone induction is Mpk1-independent, the underlying mechanism being unknown[13]. This increase in assembled proteasomes is transient and is followed by an overall reduction via proteaphagy[17].

The mechanism of increased RPAC translation, while most protein synthesis is inhibited, is so far unknown. In this Article, we show that the endocytic protein Ede1 plays a key role in this process. Our findings indicate that actin remodelling is important for regulating the localization and therefore the selective translation of messenger RNAs encoding stress-induced proteins such as RPACs.

## Results

**Identification of potential RPAC translation regulators.** To better understand the mechanisms underlying selective RPAC translation, we established the FGH17 reporter system. The FGH17 reporter encodes two N-terminal Flag epitopes, a green fluorescent protein (GFP) and a C-terminal haemagglutinin tag (HA), under the control of the regulatory elements of the RPAC Adc17 (Fig. 1a). FGH17-containing cells had low basal levels of FGH17, which strongly increased following rapamycin (TORC1 inhibitor) treatment, as for the endogenous RPAC Nas6 (Fig. 1a). We additionally compared the behaviour of endogenous *ADC17* mRNA with that of *FGH17* to confirm that they share similar translation regulation. Using RiboTag[18], we observed that rapamycin increased the recruitment of both *ADC17* mRNA and *FGH17* mRNA to ribosomes for translation (Fig. 1b). This confirmed that FGH17 is a good reporter to interrogate how translation of mRNAs such as *ADC17* are regulated upon stress. We next investigated the contribution of the untranslated regions (UTRs) to FGH17 translation regulation. Deletion of the 3′ UTR had no effect on the regulation of the FGH17 reporter. In contrast, deletion of the 5′ UTR abrogated FGH17 translation, indicating that the 5′ UTR of *ADC17* mRNA contains the required translation regulation element(s) (Fig. 1c). Deletion of the 40 nucleotides upstream of the start codon (FGH17-40ntΔ) only slightly decreased translation of the FGH17 reporter, while deletion of the 70 nucleotides upstream of the start codon

MRC Protein Phosphorylation and Ubiquitylation Unit, School of Life Sciences, University of Dundee, Dundee, UK. ✉e-mail: arousseau@dundee.ac.uk

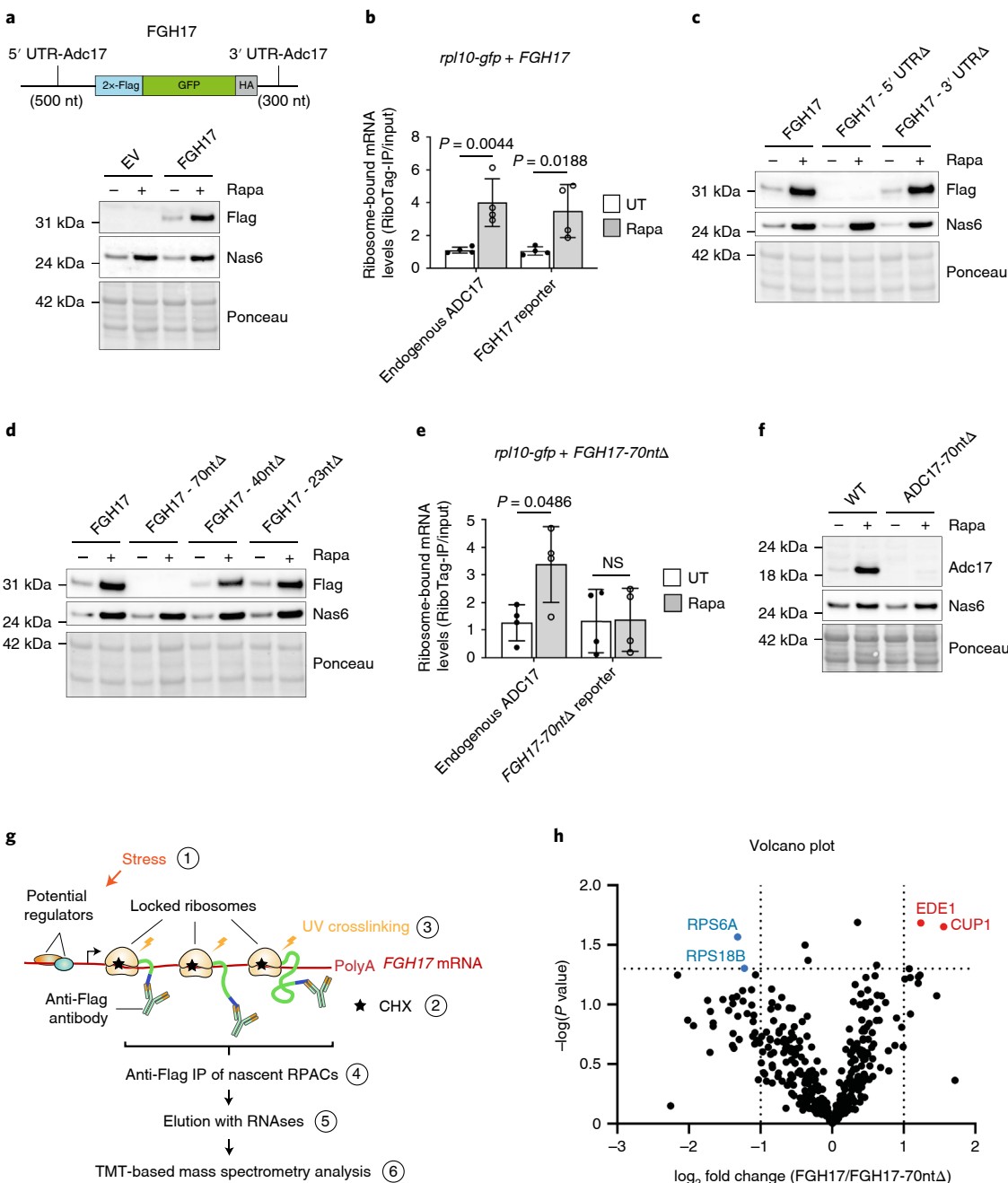

**Fig. 1 | Identification of proteins interacting with translating RPAC reporter mRNAs. a**, Cartoon depicting the FGH17 reporter, consisting of tandem reporters expressed under control of *ADC17* UTRs and western blot analysis of FGH17 expression in untreated cells or cells treated with 200 nM rapamycin (Rapa) for 4 h. Ponceau S staining was used as loading control. Empty vector, EV. **b**, mRNA levels of endogenous *ADC17* and of *FGH17* bound to ribosomes after 1.5 h rapamycin treatment compared with untreated cells. Analysis was performed by RiboTag immunoprecipitation (IP) followed by qRT–PCR and normalized to the housekeeping gene *ALG9*. Ribosome-bound mRNA corresponds to the level of RiboTag IP mRNA normalized to the level of Input mRNA. Data are presented as mean ± s.d., n = 4, unpaired two-tailed Student's t-test. **c,d**, Western blot analysis of WT and mutant FGH17 reporters (anti-Flag) and Nas6 in untreated yeast cells or yeast cells treated with 200 nM rapamycin (Rapa) for 4 h. Ponceau S staining was used as loading control. **e**, mRNA levels of endogenous *ADC17* and of *FGH17-70ntΔ* bound to ribosomes after 1.5 h rapamycin treatment compared with untreated cells. Analysis was performed as in **b**. Data are presented as mean ± s.d., n = 4, unpaired two-tailed Student's t-test. **f**, Western blot analysis of Adc17 and Nas6 expression in WT and ADC17-70ntΔ untreated cells or cells treated with 200 nM rapamycin (Rapa) for 4 h. Ponceau S staining was used as loading control. **g**, Cartoon depicting the proteomics experimental design. Step 1, cells were treated with 200 nM rapamycin for 1.5 h or were left untreated; step 2, ribosomes were locked on mRNAs by treating cells with 35 μM CHX; step 3, cells were treated with 1.2 J cm⁻² UV to covalently crosslink proteins to RNA; step 4, translating *FGH17* mRNAs were immunoprecipitated; step 5, proteins bound to translating *FGH17* mRNAs were recovered by RNase treatment before being subjected to quantitative proteomics (Step 6). **h**, Volcano plot showing the proteins that were differentially recovered from FGH17 and FGH17-70ntΔ immunoprecipitates. Each dot represents a protein. The red and blue dots are proteins significantly more and less bound to *FGH17* mRNA compared with *FGH17-70ntΔ* mRNA, respectively. n = 5 biologically independent samples per condition; P values were determined by multiple unpaired two-tailed t-test. In **a, c, d** and **f**, n = 3 independent biological replicates.

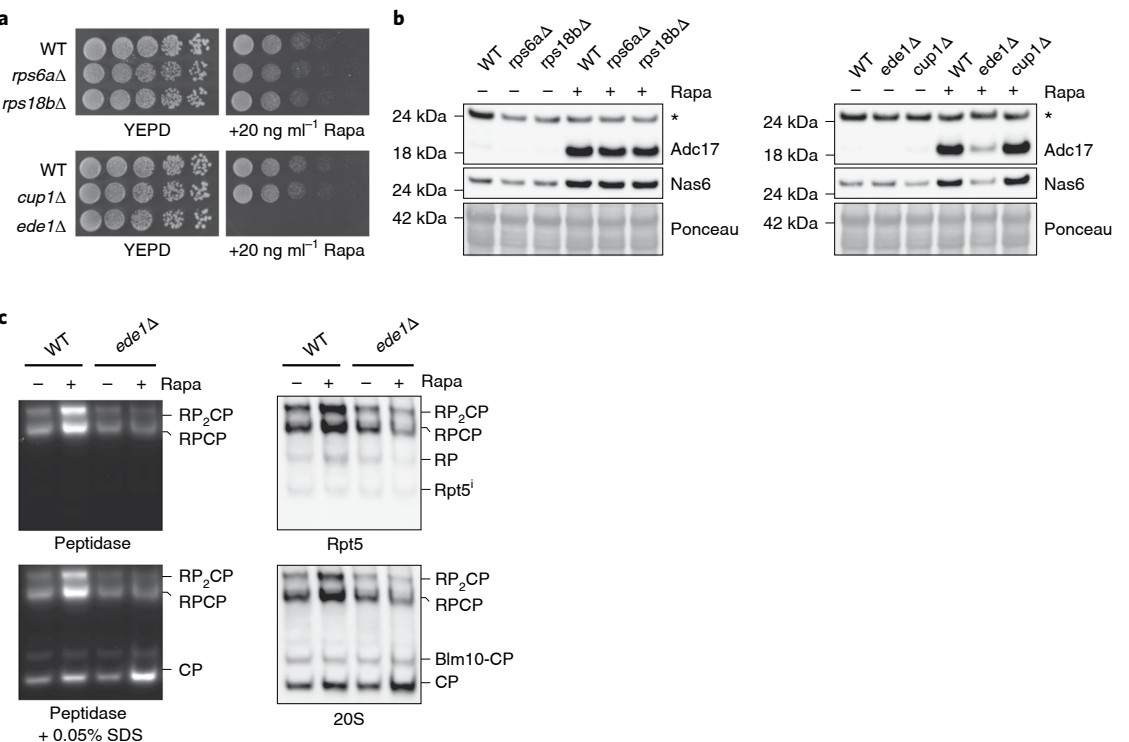

**Fig. 2 | Ede1 regulates proteasome assembly upon TORC1 inhibition. a**, Cells spotted in a fivefold dilution and grown for 3 days on plates with or without 20 ng ml⁻¹ rapamycin. **b**, Western blot analysis of RPACs in WT and deletion strains that were untreated or treated with 200 nM rapamycin (Rapa) for 4 h. Ponceau S staining was used as a loading control. Asterisk indicates non-specific band. Data are representative of three independent biological replicates. **c**, Gradient native polyacrylamide gel electrophoresis (PAGE) (3.8–5%) of yeast extracts from untreated cells or cells treated with 200 nM rapamycin (Rapa) for 3 h, monitored by the fluorogenic substrate Suc-LLVY–AMC (left) and by immunoblots (right). CP, single-capped (RPCP), double-capped (RP₂CP) and Blm10-capped (Blm10-CP) proteasome complexes are indicated. Rpt5 and 20 S antibodies recognize the RP and the CP, respectively. Data are representative of three independent biological replicates.

(FGH17-70ntΔ) prevented translation (Fig. 1d). This was not due to alteration of the Kozak sequence, as re-introducing *ADC17* Kozak sequence to *FGH17-70ntΔ* mRNA (FGH17-70ntΔ+Kozak) was not enough to restore FGH17 expression (Extended Data Fig. 1a). The 70-nucleotide region alone was not sufficient for FGH17 reporter expression (Extended Data Fig. 1b). Comparing FGH17 with FGH17-70ntΔ by RiboTag, we observed that the deletion of this 70-nt sequence prevented the recruitment of *FGH17-70ntΔ* mRNA to ribosomes upon rapamycin treatment (Fig. 1b,e) and decreased its stability by about twofold (Extended Data Fig. 1c). Deleting this 70-nt region at the endogenous *ADC17* locus with clustered regularly interspaced short palindromic repeats (CRISPR)/Cas9, we similarly observed abrogation of Adc17 expression (Fig. 1f). These findings indicated that the FGH17 reporter reflects the regulation of the endogenous *ADC17* gene.

To discover new RPAC translation regulators, we identified RNA-binding proteins with increased recruitment to translating wild-type (WT) *FGH17* mRNAs compared with non-translatable *FGH17-70ntΔ* mRNAs in vivo. To this end, we treated yeast cells with rapamycin to stimulate FGH17 translation (Fig. 1g, step 1). Polysomes were stabilized by adding cycloheximide (CHX) to the cells (Fig. 1g, step 2), and UV-crosslinking covalently linked the RNA and any bound proteins together (Fig. 1g, step 3). We next used anti-FLAG beads to immunoprecipitate translating *FGH17* mRNA complexes where locked ribosomes had already synthesized one or both N-terminal FLAG tags (Fig. 1g, step 4). As *FGH17-70ntΔ* mRNAs are not translated, these samples should only immunoprecipitate proteins that bind non-specifically to the anti-FLAG beads. On the basis of the prediction that potential regulators of

RPAC translation would be UV-crosslinked to translating *FGH17* mRNA (Fig. 1g), we used RNases to specifically elute these potential regulators (Fig. 1g, step 5). We identified proteins in the RNase elution by tandem mass tag (TMT)-based quantitative proteomics (Fig. 1g, step 6). Quantitative analysis (Fig. 1h) revealed that two proteins, Ede1 and Cup1, were enriched in the WT compared with FGH17-70ntΔ samples. In contrast, two proteins were significantly depleted in the WT samples (Rps6a and Rps18b), suggesting they could be translational repressors (Fig. 1h). Taken together, these results identify a region of *ADC17* mRNA essential for translation and discover potential regulators of selective RPAC translation.

**Ede1 is important for RPAC induction upon TORC1 inhibition.**
Defects in RPAC regulation sensitize yeast to rapamycin[13]. Therefore, to examine the involvement of identified proteins in RPAC regulation, we first tested knockout mutants for rapamycin sensitivity. Mutants of the ribosomal subunits Rps6a and Rps18b, which were less associated with translating *FGH17* mRNA, were similarly sensitive to rapamycin as WT cells (Fig. 2a). Cup1 and Ede1 were found to be associated more with translating *FGH17* mRNA, and while the *cup1Δ* mutant showed a similar level of sensitivity to rapamycin as WT cells, *ede1Δ* cells were highly sensitive (Fig. 2a).

We next tested whether these mutants were defective in rapamycin-induced RPAC expression. Unlike *rps6aΔ*, *rps18bΔ* and *cup1Δ* cells, *ede1Δ* cells were severely impaired in both Adc17 and Nas6 induction following rapamycin treatment (Fig. 2b), which, together with the rapamycin sensitivity, was rescued by re-introducing Ede1 (Extended Data Fig. 2a,b). TORC1-mediated Rps6 phosphorylation was still inhibited following rapamycin

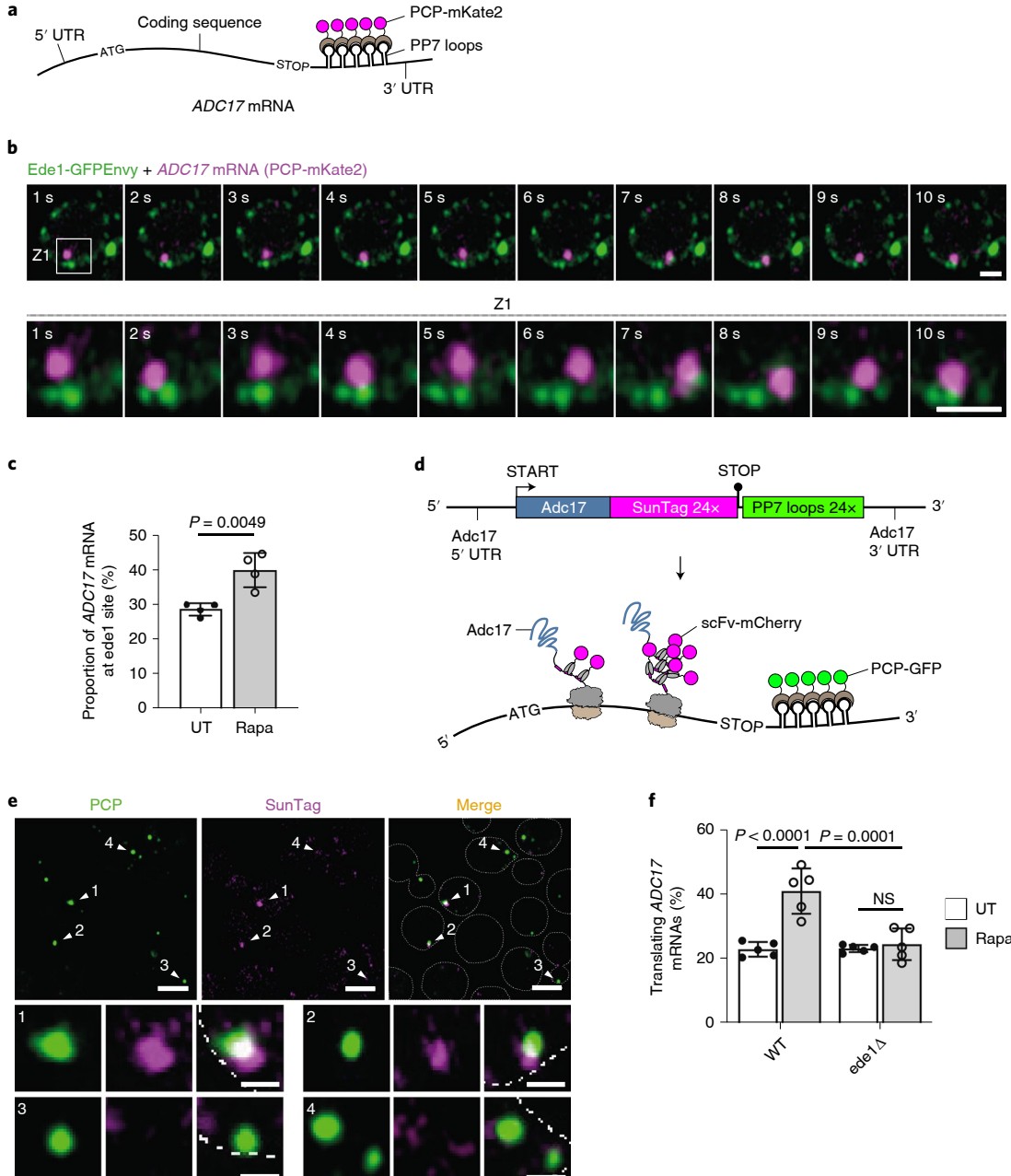

**Fig. 3 | Ede1 controls *ADC17* mRNA translation upon TORC1 inhibition. a**, Cartoon depicting how single-molecule *ADC17* mRNAs are labelled with PCP fused to mKate2 for fluorescence live-cell imaging. PP7 stem loops were introduced into the endogenous *ADC17* mRNA, allowing it to be selectively labelled with PCP-mKate2 in cells expressing Ede1-GFPEnvy. **b**, Montage from time-lapse imaging showing contacts between Ede1-GFPEnvy (green) and *ADC17* mRNA (magenta). Scale bars, 1 μm. *n* = 4 biologically independent experiments. **c**, Frequency of *ADC17* mRNAs (green) co-localizing with Ede1-tdimer2 in cells grown for 3 h with or without 200 nM rapamycin (Rapa). UT, untreated. Data are presented as mean ± s.d. *n* = 4 biologically independent experiments with 521 *ADC17* mRNAs for each condition. Statistical analysis was carried out using unpaired two-tailed Student's *t*-test. **d**, Schematic representation of ADC17-SunTag reporter mRNA used for single-molecule imaging of mRNA translation during stress. PCP-GFP labels *ADC17* mRNA, whereas scFv-mCherry labels translating Adc17 protein. **e**, Representative microscopy images of yeast cells expressing ADC17-SunTag reporter mRNA. Translating *ADC17* mRNAs are GFP (green)- and mCherry (magenta)-positive, while non-translating *ADC17* mRNAs are only positive for GFP. Translating *ADC17* mRNAs are denoted by white arrowheads 1 and 2, while non-translating *ADC17* mRNAs are denoted by white arrowheads 3 and 4. Higher magnification is shown at the bottom. Scale bars, 3 μm. *n* = 5 biologically independent experiments. **f**, Frequency of *ADC17* mRNAs undergoing translation in WT and *ede1Δ* cells that are untreated or treated with 200 nM rapamycin (Rapa) for 3 h using the SunTag labelling method. Data are presented as mean ± s.d., *n* = 5 biologically independent experiments with 547 *ADC17* mRNAs for each condition. Statistical analysis was carried out using two-way ANOVA *t*-test (Tukey multiple comparison test).

treatment in *ede1Δ* cells, indicating that Ede1 acts downstream of TORC1 inhibition (Extended Data Fig. 2c). A defect in RPAC induction following rapamycin treatment is expected to lead to a defect in increased proteasome assembly and activity[13]. Accordingly, *ede1Δ* cells failed to increase assembly and activity of the 26S proteasome, although there was an increase in 20S CPs (Fig. 2c). This

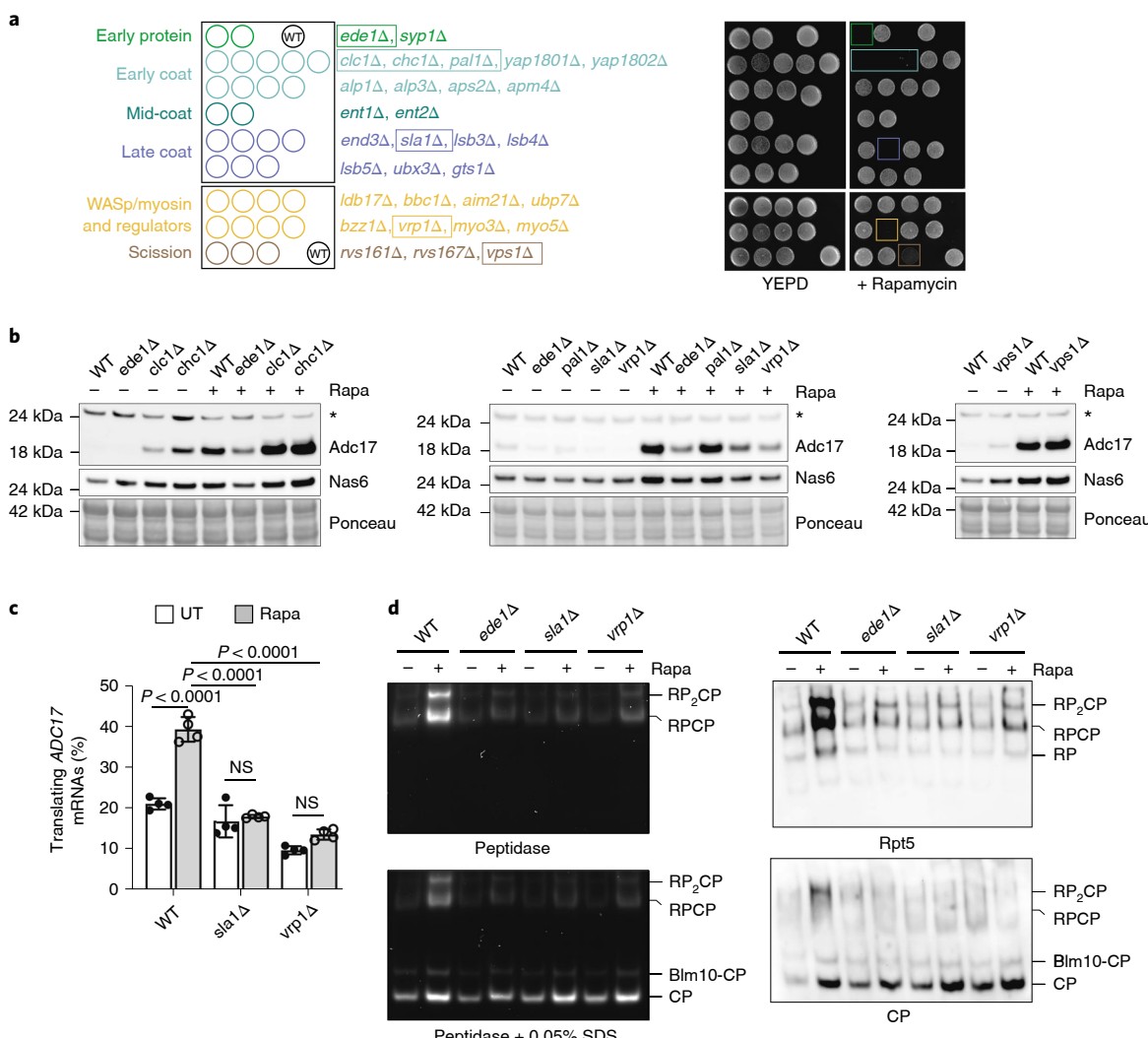

**Fig. 4 | Ede1, Sla1 and Vrp1 are important for proteasome assembly and activity. a**, Screen for rapamycin sensitivity with deletion strains covering all non-essential endocytic genes. Left: schematic representation of YEPD plates indicating the position of deletion strains. WT yeast was used as a control in the indicated positions. Right: yeast growth for 3 days on YEPD plate with or without 20 ng ml⁻¹ rapamycin from the indicated strains. Strains that failed to grow on rapamycin are indicated in coloured boxes in both the schematic and plate image. **b**, Western blot analysis of RPACs in WT and deletion strains that were untreated or treated with 200 nM rapamycin (Rapa) for 4 h. Ponceau S staining was used as a loading control. Asterisk indicates non-specific band. **c**, Frequency of *ADC17* mRNAs undergoing translation in WT, *sla1Δ* and *vrp1Δ* cells that were untreated or treated with 200 nM rapamycin (Rapa) for 3 h using the SunTag labelling method. Data are presented as mean ± s.d. $n = 4$ biologically independent experiments with 511 *ADC17* mRNAs for each condition. Statistical analysis was carried out using two-way ANOVA i-test (Tukey multiple comparison test). **d**, Gradient Native PAGE (3.8–5%) of yeast extracts from cells that were untreated or treated with 200 nM rapamycin (Rapa) for 3 h, monitored by the fluorogenic substrate Suc-LLVY–AMC (left) and by immunoblots (right). CP, RPCP, RP₂CP and Blm10-CP proteasome complexes are indicated. Rpt5 and 20 S antibodies recognize the RP and the CP, respectively. In **a**, **b** and **d**, data are representative of three independent biological replicates.

defect is symptomatic of RP assembly defects and a hallmark of RPAC-deleted cells[13,19–21]. Ede1 is therefore necessary for enhanced proteasome assembly following rapamycin treatment, by increasing the amount of RPACs available.

**Ede1 interacts with *ADC17* mRNA to regulate its translation.** As Ede1 associates with translating *FGH17* mRNA and is critical for proteasome homeostasis upon TORC1 inhibition, we predicted that Ede1 will be in contact with *ADC17* mRNAs upon rapamycin treatment to play a role in their translation. To explore this possibility, PP7 stem loops were introduced into the endogenous *ADC17* mRNA, allowing it to be labelled with PP7 bacteriophage coat protein (PCP) fused to mKate2 in cells expressing GFPEnvy-tagged Ede1 (Fig. 3a). We tracked single molecules of

labelled *ADC17* mRNA and observed frequent contact of Ede1 and *ADC17* mRNA, demonstrating that this interaction is occurring in vivo (Fig. 3b and Supplementary Videos 1 and 2). Around 29% of *ADC17* mRNAs were associated to Ede1 under basal conditions, and this significantly increased to about 40% following rapamycin treatment (Fig. 3c). This is consistent with a recent study that identified Ede1 as a potential RNA-binding protein[22]. To confirm that Ede1 is regulating *ADC17* mRNA at the level of translation, we deployed the SunTag labelling method[23]. This method uses the multimerization of Gcn4 epitope (SunTag) that, when translated, is recognized by multiple single-chain antibodies coupled to a fluorescent protein (scFv-mCherry), enabling quantitative visualization of the translation of individual mRNA molecules in living cells (Fig. 3d). We identified two populations of *ADC17* mRNAs: those

co-localizing with SunTag signal that are translationally active (GFP and mCherry signal) (Fig. 3e, arrowheads 1 and 2) and those devoid of SunTag signal that are translationally inactive (GFP only) (Fig. 3e, arrowheads 3 and 4) (Supplementary Videos 3 and 4). On average, ~23% of *ADC17* mRNAs were translationally active in untreated cells, increasing to ~40% in rapamycin-treated cells, attesting that *ADC17* mRNA translation is increased upon TORC1 inhibition (Fig. 3f). This increased translation of *ADC17* mRNA was lost in *ede1Δ* cells treated with rapamycin, confirming the importance of Ede1 for Adc17 translation regulation (Fig. 3f). Taken together, these results show that *ADC17* mRNAs partly localize to Ede1 sites and demonstrate that Ede1 is critical for *ADC17* mRNA translation upon TORC1 inhibition.

**Ede1, Sla1 and Vrp1 are important for RPAC translation.** Having established that Ede1 regulates RPAC translation, we investigated the underlying mechanism. Ede1 is an early coat protein involved in clathrin-mediated endocytosis (CME)[24]. To determine whether endocytosis is important for stress-mediated proteasome assembly, we tested if mutants of other non-essential proteins involved in CME mimicked the defects of *ede1Δ* cells. We initially screened mutants for increased rapamycin sensitivity, revealing six further proteins that were essential for growth on rapamycin (Fig. 4a). We examined whether these proteins were also involved in RPAC induction following rapamycin treatment. Chc1, Clc1, Pal1 and Vps1 were dispensable for induction of Adc17 and Nas6 after rapamycin treatment; however, *sla1Δ* and *vrp1Δ* cells were severely impaired in RPAC induction (Fig. 4b). Like for *ede1Δ* cells, this was due to a translation defect, as observed using the SunTag labelling method (Fig. 4c). We next confirmed that *sla1Δ* and *vrp1Δ* cells had similar defects to *ede1Δ* in proteasome assembly and activity in response to rapamycin (Fig. 4d).

**Actin remodelling affects *ADC17* mRNA localization.** Ede1 is one of the first proteins to be recruited to endocytic sites (Fig. 5a, step 1). Sla1 forms a heterodimeric complex with the actin nucleation promoting factor (NPF) Las17, which is recruited to endocytic patches via Sla1–Ede1 interaction (Fig. 5a, step 2). Vrp1 is recruited to the endocytic site by Las17 (Fig. 5a, step 3), and contributes to the recruitment and the activation of Myo3 and Myo5 that have both NPF and motor activities (Fig. 5a, step 4). NPFs further recruit and activate the actin nucleator complex Arp2/3 to initiate actin nucleation (Fig. 5a, step 5)[25–27]. As Ede1, Sla1 and Vrp1 localize to and regulate cortical actin patches at the endocytic site, and *ADC17* mRNA makes contacts with Ede1, Sla1 and Vrp1 (Fig. 3c and Extended Data Fig. 3a,b), it seemed likely there might be a role for actin in *ADC17* mRNA regulation. To test this, we fixed cells expressing endogenous PCP-GFP-labelled *ADC17* mRNA and stained them

with phalloidin to visualize actin. Yeast has two major actin structures: actin cables, which are polarized linear bundles of parallel actin filaments extending along the long axis of cells, and cortical actin patches, which are dense dendritic networks of branched actin filaments localized at the plasma membrane[26]. *ADC17* mRNA was mainly seen to localize on actin cables (~70%), with the remainder either on cortical actin patches (~26.5%) or not associated with any phalloidin staining (~3.5%) (Fig. 5b,c). We next sought to examine whether *ADC17* mRNA is associated with the actin cytoskeleton using live-cell imaging. Tagged Abp1 and Abp140 were used to visualize cortical actin patches and cables, respectively. We observed that *ADC17* mRNA is often associated with actin cables in vivo (Fig. 5d and Supplementary Videos 5 and 6), while its interaction with patches is more transient and dynamic, as previously observed for Ede1 (Extended Data Fig. 3c and Supplementary Videos 7 and 8). Overall, these data might suggest movement of *ADC17* mRNA along actin cables, although clear determination of direction and mechanism of movement remain to be determined.

The distribution of cortical actin patches and cables is polarized in budding yeast, with cortical actin patches being found almost exclusively in the bud, and cables being aligned longitudinally from the mother cell into the bud[26]. It has been reported that rapamycin depolarizes the actin cytoskeleton[16], and we have shown that *ADC17* mRNA is largely localized to actin structures (Fig. 5b–d). It is possible, therefore, that actin depolarization is a key step in RPAC induction. We first monitored the kinetics of actin depolarization after rapamycin treatment. Budding cells containing more than six cortical actin patches in the larger mother cell were considered to have a depolarized actin cytoskeleton, as previously described[28,29]. Rapamycin rapidly induced actin depolarization, peaking at 1H and returning to near-normal levels at 4H (Fig. 5e,f). RPAC induction coincides with actin depolarization, indicating that actin remodelling may relocate RPAC mRNAs to trigger their translation (Fig. 5g). To test this possibility, we tracked *ADC17* mRNA in cells stained for actin. Rapamycin treatment induced a shift of *ADC17* mRNA localization from actin cables to patches, from 1 h (1.7-fold increase) onward compared with untreated cells (Fig. 5h). These results show that actin depolarization upon TORC1 inhibition relocates *ADC17* mRNAs from actin cables to cortical actin patches, which could be important for its selective translation.

**Actin disruption induces proteasome assembly and activity.** We next tested whether an alternative means of selectively disrupting actin cables induced RPAC translation. Cells were therefore treated with 25 µM latrunculin B (Lat-B), which at this concentration only disrupts actin cables[30], as observed upon rapamycin treatment. Lat-B treatment completely abolished actin cables and thereby relocated *ADC17* mRNA to either cortical actin patches or to an unbound

---

**Fig. 5 | *ADC17* mRNA associates with actin cables and re-localizes to patches upon stress. a**, Cartoon depicting the role of Ede1, Sla1 and Vrp1. (1) Ede1 is recruited to nascent endocytic sites. (2) Sla1 recruits the NPF Las17 aided by the presence of Ede1. (3) Vrp1 is recruited to the endocytic site by Las17. (4) Vrp1 helps recruit Myo3 and Myo5. (5) The NPFs Las17, Myo3 and Myo5 recruit and activate the actin nucleator complex Arp2/3. **b**, Representative microscopy images (maximum-intensity *Z*-projection) of yeast containing the PCP-GFP-labelled *ADC17* mRNA (cyan) and stained for actin (red). Z1, Z2 and Z3 areas are shown at higher magnifications. White, yellow and green arrowheads indicate *ADC17* mRNAs bound to actin cable, cortical actin patch and not associated to actin, respectively. Scale bars, 3 µm. *n* = 4 biologically independent experiments. **c**, Frequency of *ADC17* mRNAs bound to actin cable, cortical actin patch and not associated to actin structures. Data are presented as mean ± s.d., *n* = 4 biologically independent experiments (*n* = 232 *ADC17* mRNAs per condition). **d**, Representative microscopy images showing *ADC17* mRNA (cyan) interaction with actin cable (Abp140-mKate2, red). Scale bars, 3 µm. *n* = 3 biologically independent experiments. **e**, Representative microscopy images (maximum-intensity *Z*-projection) of yeast that was untreated or treated with 200 nM rapamycin for 1 h and stained with rhodamine phalloidin to visualize actin (hot red LUT). Scale bars, 3 µm. *n* = 4 biologically independent experiments. **f**, Frequency of polarized cells following rapamycin (200 nM) treatment for the indicated time. Data are presented as mean ± s.d. *n* = 4 biologically independent experiments (*n* = 399, 361, 337, 329 and 252 cells for conditions 0H, 1H, 2H, 3H and 4H, respectively). **g**, Western blot analysis of RPACs in WT cells that were untreated or treated with 200 nM rapamycin (Rapa) for the indicated time. Ponceau S staining was used as loading control. *n* = 3 biologically independent experiments. **h**, Frequency of *ADC17* mRNA bound to actin cable, cortical actin patch or not associated to actin in WT cells that were untreated or treated with 200 nM rapamycin for the indicated time. Data are presented as mean ± s.d. *n* = 4 biologically independent experiments (*n* = 220 *ADC17* mRNAs per condition). In **f** and **h**, one-way ANOVA *t*-test (Dunnett multiple comparison test) was used.

state (Fig. 6a,b). Analysing the pathway regulating RPAC levels, we showed that Lat-B activates Mpk1 kinase (Fig. 6c). RPAC expression is induced straight after Mpk1 activation, as previously reported for rapamycin[13] (Fig. 6c). This was further confirmed using genetic disruption of actin cables in the temperature-sensitive mutant *act1-101* (Extended Data Fig. 4a,b). Moreover, actin nucleation at the surface

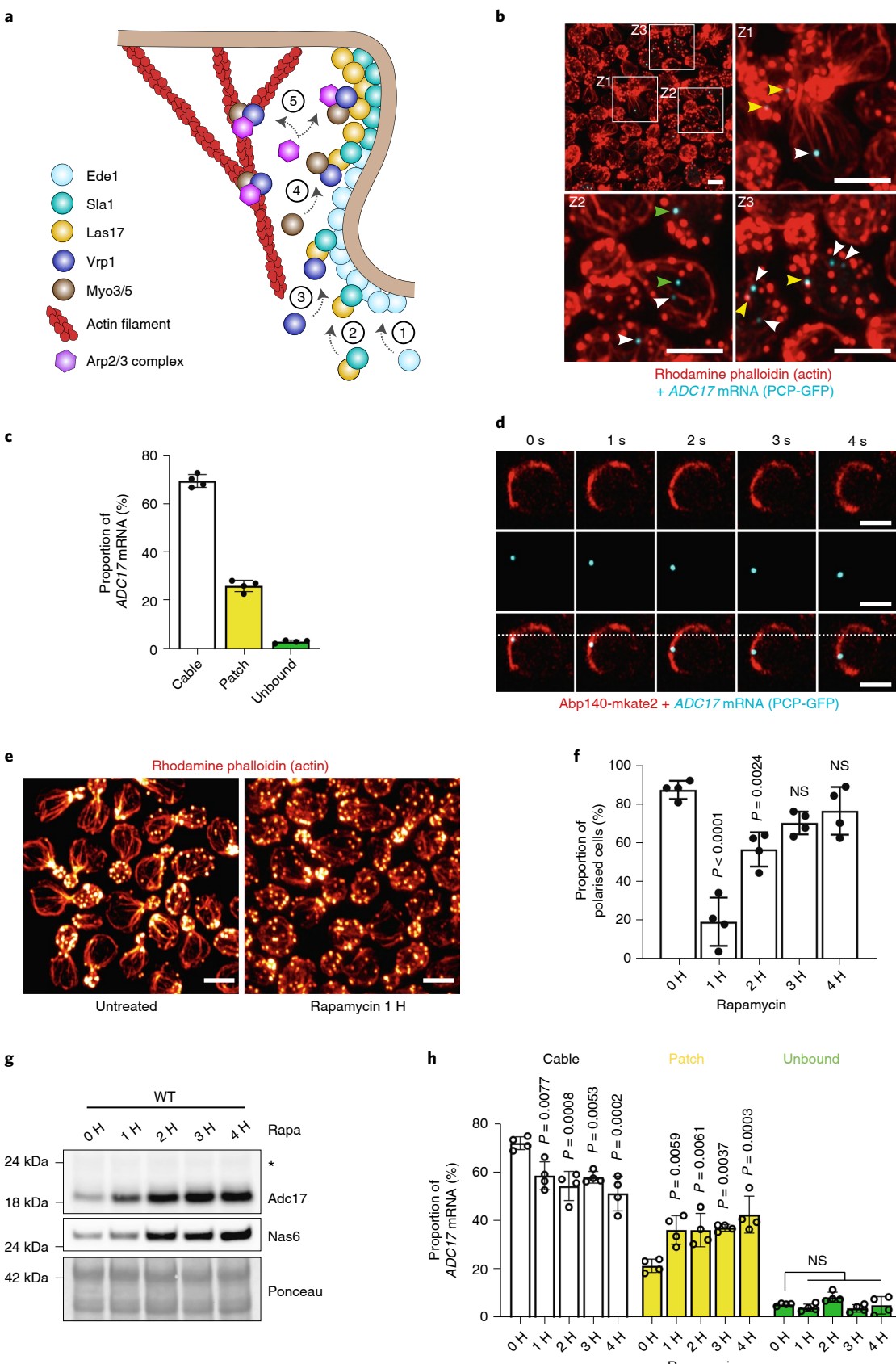

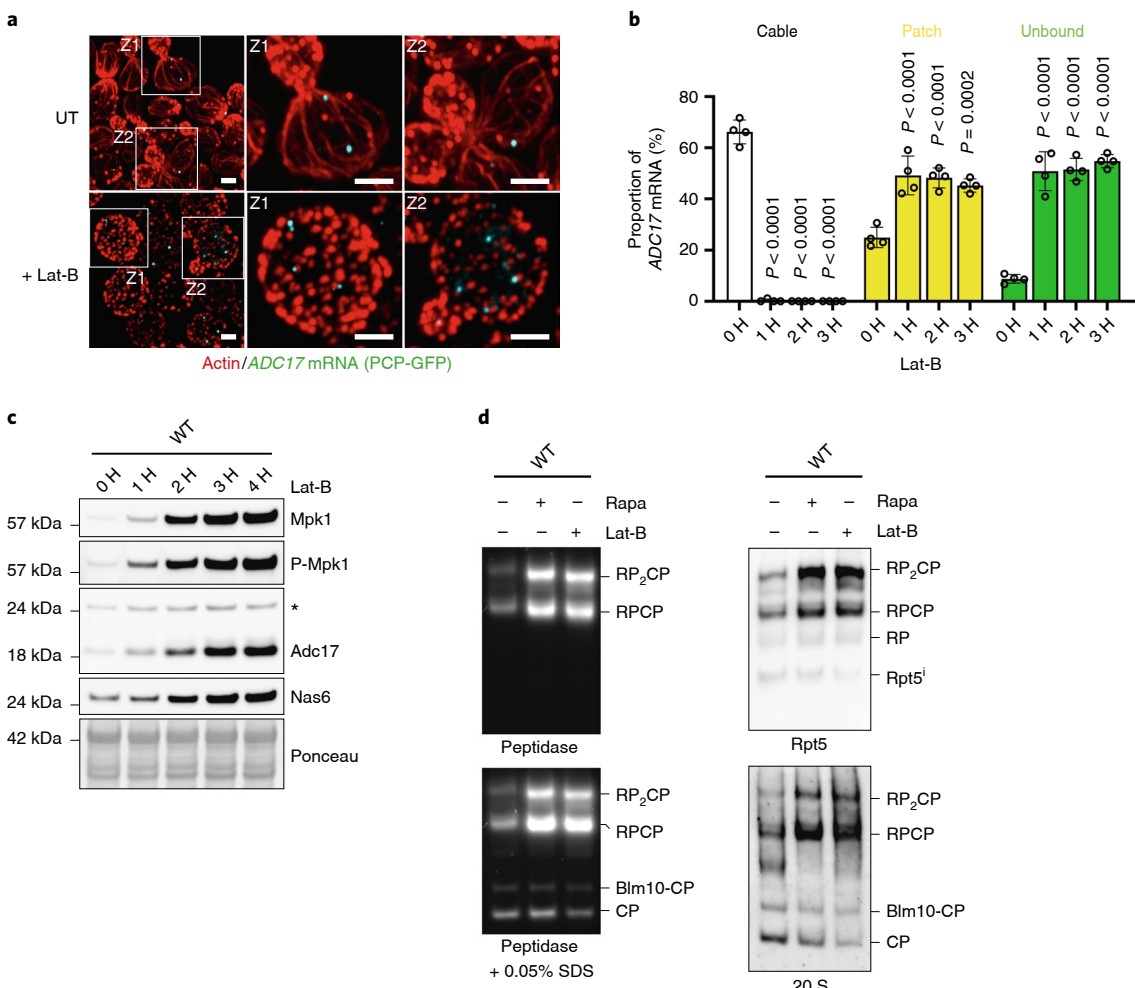

**Fig. 6 | Lat-B induces RPACs and proteasome assembly. a**, Representative microscopy images (maximum-intensity $Z$-projection) of yeast cells containing the PCP-GFP-labelled *ADC17* mRNA (cyan), with or without 25 μM Lat-B for 1 h and stained with rhodamine phalloidin to visualize actin (red). Z1 and Z2 areas are shown at higher magnifications. Scale bars, 2 μm. $n = 4$ biologically independent experiments. **b**, Frequency of *ADC17* mRNA bound to actin cable, cortical actin patch or not associated to actin in WT cells that are untreated or treated with 25 μM Lat-B for the indicated time. Data are presented as mean ± s.d. $n = 4$ biologically independent experiments ($n = 206$ *ADC17* mRNAs for each condition). Statistical analysis was carried out using one-way ANOVA $t$-test (Dunnett multiple comparison test). **c**, Western blot analysis of RPACs and Mpk1 kinase in WT cells that are untreated or treated with 25 μM Lat-B for the indicated time. Ponceau S staining was used as a loading control. **d**, Gradient native PAGE (3.8–5%) of yeast extracts from cells that are untreated or treated with 200 nM rapamycin (Rapa) or 25 μM Lat-B for 3 h, monitored by the fluorogenic substrate Suc-LLVY–AMC (left) and by immunoblots (right). CP, RPCP, RP₂CP and Blm10-CP proteasome complexes are indicated. Rpt5 and 20S antibodies recognize the RP and the CP, respectively. In **c** and **d**, data are representative of three independent biological replicates.

of cortical actin patches was not required for RPAC translation. Latrunculin-A treatment, which disrupts actin at patches as well as cables, had similar effects to that of Lat-B (Extended Data Fig. 4c–e). As the RPAC level mirrors the level of proteasome assembly, we monitored the impact of Lat-B on proteasome activity. In-gel peptidase assays showed that, as for rapamycin, Lat-B is a potent inducer of proteasome assembly and activity (Fig. 6d), attesting that actin remodelling regulates proteasome homeostasis.

**Ede1-tethered *ADC17* mRNAs are more translated upon stress.** Having found that Lat-B induces RPAC expression, we sought to determine whether Ede1 is required for this process, as for rapamycin. Despite actin becoming depolarized after Lat-B treatment (Fig. 7a), RPACs were not induced in *ede1Δ* cells (Fig. 7b). This result shows that Ede1 plays a role downstream of actin depolarization, possibly by stabilizing *ADC17* mRNAs at cortical actin patches. While rapamycin treatment induced a shift of *ADC17* mRNA localization from actin cables to cortical actin patches in WT cells, this

re-localization was lost in *ede1Δ* cells (Fig. 7c). This confirms Ede1 helps stabilize *ADC17* mRNA at cortical actin patches following TORC1 inhibition.

If stabilization of *ADC17* mRNAs at Ede1 sites is important for RPAC translation, artificially targeting *ADC17* mRNA to this location may impact its translation levels upon TORC1 inhibition. Therefore, we set out to establish a targeting system in which Ede1 is fused to a nanobody-recognizing GFP (Ede1-aGFP) that will recruit the PCP-GFP proteins and thence the *ADC17* mRNAs containing the PP7 stem loops (Fig. 7d). We first confirmed that PCP-GFP proteins are tethered to the plasma membrane where Ede1 is localized (Fig. 7e). Using a doubly tagged version of Ede1 (Ede1-tdimer2-aGFP), we also demonstrated that the PCP-GFP proteins are co-localizing with Ede1 proteins (Extended Data Fig. 5a,b). In this system, Ede1 sites are decorated with PCP-GFP and, consequently, all PCP-GFP dots do not correspond to one *ADC17* mRNA molecule. Because of this, we validated that *ADC17* mRNAs are indeed tethered to Ede1 sites using a doubly tagged version of

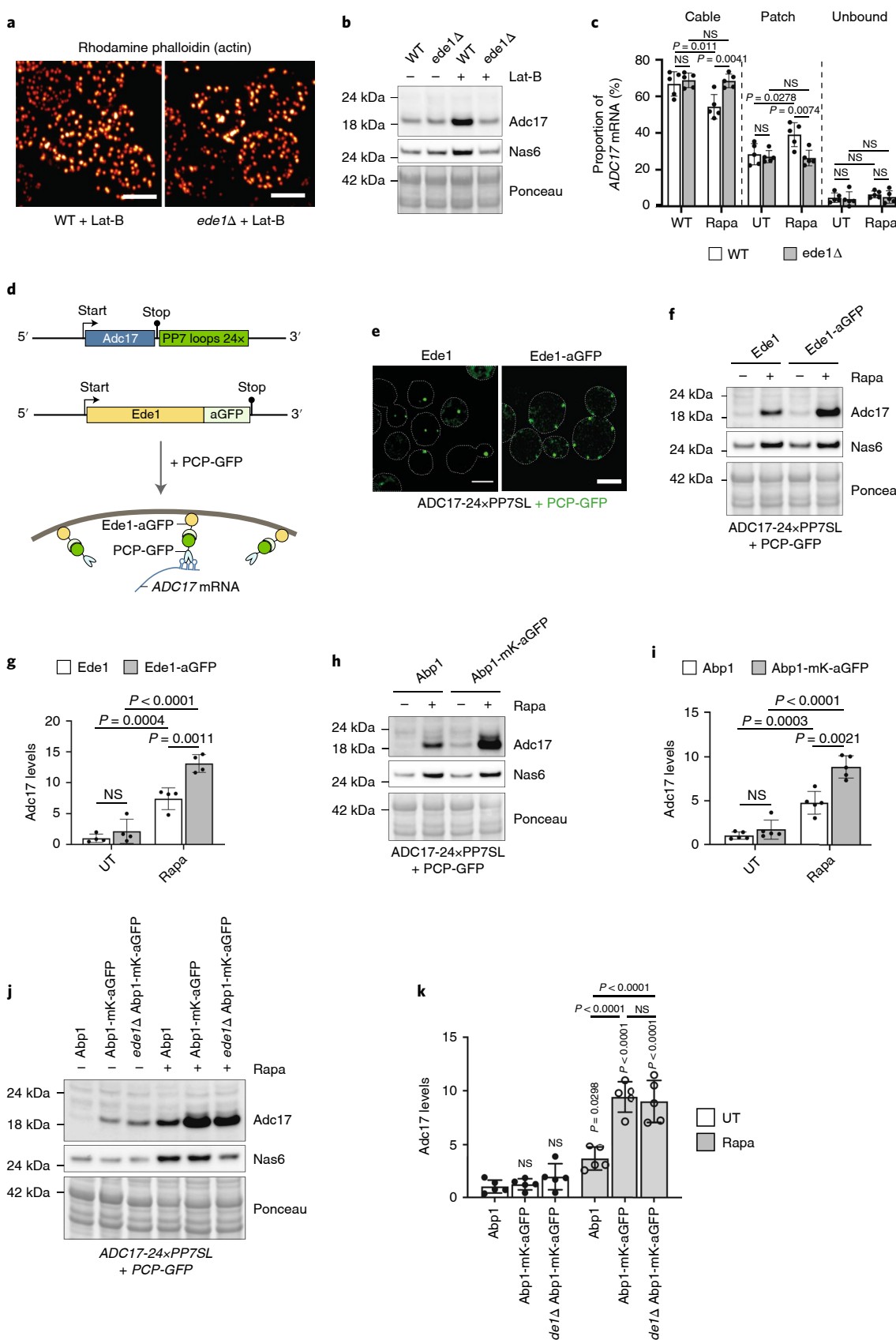

*ADC17* mRNA (Adc17-PP7SL-MS2SL). We observed that *ADC17* mRNAs are robustly tethered to Ede1-aGFP/PCP-GFP sites, indicating that our mRNA targeting system is efficient (Extended Data

Fig. 5c,d). Moreover, the tethering of *ADC17* mRNA to Ede1 sites had no impact on its stability (Extended Data Fig. 5e). We thus used this system to monitor the impact of artificially tethering *ADC17*

**Fig. 7 | Tethering of *ADC17* mRNA to actin patches enhances its translation upon stress. a**, Representative microscopy images (maximum-intensity *Z*-projection) of WT and *ede1*Δ cells treated with 25 μM Lat-B for 1 h and stained for actin (hot red LUT). Scale bars, 3 μm. *n* = 3 biologically independent experiments. **b**, Western blot analysis of RPACs in WT and *ede1*Δ cells that are untreated or treated with 25 μM Lat-B for 3 h. Ponceau S staining was used as loading control. *n* = 3 biologically independent experiments. **c**, Frequency of *ADC17* mRNA bound to actin cable, cortical actin patch or not associated to actin in WT and *ede1*Δ cells that are untreated or treated with 200 nM rapamycin (Rapa) for 1 h. Data are presented as mean ± s.d. *n* = 5 biologically independent experiments (*n* = 212 *ADC17* mRNAs for each condition). **d**, Schematic representation of the system used to artificially tether *ADC17* mRNA to Ede1. aGFP, nanobody against GFP. **e**, Representative microscopy images of yeast cells containing PCP-GFP-labelled *ADC17* mRNA (green) and expressing either WT Ede1 or Ede1 tagged with a nanobody against GFP (Ede1-aGFP). Scale bars, 3 μm. *n* = 3 biologically independent experiments. **f**, Western blot analysis of RPACs in cells shown in **e** that are untreated or treated with 200 nM rapamycin (Rapa) for 4 h. Ponceau S staining was used as loading control. *n* = 4 biologically independent experiments. **g**, Quantification of Adc17 protein level from experiments represented in **f**. Data are presented as mean ± s.d. *n* = 4 biologically independent experiments. **h**, Western blot analysis of RPACs in the indicated cells that are untreated or treated with 200 nM rapamycin (Rapa) for 4 h. Ponceau S staining was used as loading control. *n* = 5 biologically independent experiments. **i**, Quantification of Adc17 protein level from experiments represented in **h**. Data are presented as mean ± s.d. *n* = 5 biologically independent experiments. **j**, Western blot analysis of RPACs in the indicated cells that are untreated or treated with 200 nM rapamycin (Rapa) for 4 h. Ponceau S staining was used as loading control. *n* = 5 biologically independent experiments. **k**, Quantification of Adc17 protein level from experiments represented in **j**. Data are presented as mean ± s.d. *n* = 5 biologically independent experiments. In **c, g, i** and **k**, two-way ANOVA *t*-test (Tukey multiple comparison test).

mRNAs to Ede1 sites on their translation level. We observed that Adc17 induction upon rapamycin treatment was increased around twofold when tethered to Ede1-aGFP/PCP-GFP compared with untethered mRNAs, indicating that recruitment of *ADC17* mRNA to Ede1 sites is important for its translation regulation (Fig. 7f,g). As *NAS6* mRNA does not possess the PP7 stem loops, its induction by rapamycin was unaffected by Ede1-aGFP (Fig. 7f).

As Ede1 localizes to cortical actin patches, its main function in regulating RPAC translation could be to stabilize RPAC mRNAs at these sites. Therefore, we artificially tethered *ADC17* mRNA to the cortical patch marker Abp1 and monitored RPAC levels. We observed that Adc17 induction upon rapamycin treatment was increased by 1.84-fold when tethered to Abp1 (Abp1-mK-aGFP) compared with untethered mRNAs (Abp1), which is similar to that of Ede1 targeting (Fig. 7h,i). When *ADC17* mRNA was targeted to cortical actin patches in *ede1*Δ cells, Adc17 induction upon rapamycin was restored to WT levels, indicating that an important function of Ede1 is to recruit *ADC17* mRNA to cortical actin patches (Fig. 7j,k). Taken together, these results show that Ede1-mediated recruitment of *ADC17* mRNAs at cortical actin patches following rapamycin treatment is important for stimulating Adc17 translation.

## Discussion

Local mRNA translation has been described as important for various processes, including development, cell migration, stress resistance and neuron function[31,32,33,34,35–37]. Yeast mRNAs have been reported to localize to the bud tip, the endoplasmic reticulum (ER), mitochondria and cortical actin patches, where they have either been shown, or are predicted, to be locally translated[38–41]. The most well characterized of these is the *ASH1* mRNA, which is transported along the actin cytoskeleton in a translationally repressed state to the bud tip, where it is anchored, and translation activated. It is possible that a similar mechanism is responsible for RPAC induction. In this scenario, RPAC mRNA is transported along actin cables in a translationally repressed state. When actin cables are disrupted by stress, *ADC17* mRNA re-localizes to Ede1 sites and translation inhibition is relieved. Likewise, in human cells, mRNAs have been reported to localize to the ER, mitochondria, distal parts of neurons and actin-dense structures such as focal adhesions, which are somewhat akin to cortical actin patches[42–47]. The localization of certain mRNAs has been shown to be dependent on F-actin, while other mRNAs are transported along microtubules[41,48–50]. Here we have shown that *ADC17* RPAC mRNA may be transported on actin cables and interacts with cortical actin patches. Upon rapamycin and Lat-B treatment, which respectively weaken and remove actin cables, the mRNA is stabilized at cortical actin patches. This re-localization of RPAC mRNA allows increased RPAC translation

and, ultimately, proteasome assembly in the presence of Ede1, Sla1 and Vrp1.

As cortical actin patches have a higher density of F-actin and are less sensitive to stress than actin cables, they may serve directly as a translation platform or indirectly by recruiting mRNA to a translationally active cellular compartment, helping to translate stress-induced proteins such as RPACs. In agreement with this possibility, it has recently been reported that Ede1 foci are surrounded by fenestrated ER containing membrane-associated ribosomes[49]. Recent findings show that the ER has a far more diverse role in mRNA translation than expected, with ER-bound ribosomes functioning in the synthesis of both cytosolic and ER-targeted proteins[42]. These observations have been reported in diverse organisms and using different methodologies, suggesting that the ER is a favourable environment for translation. Furthermore, ER-localized mRNA translation is less inhibited by stress than their cytosolic counterparts, suggesting the ER represents a protective environment for translation upon stress, allowing the cell to synthesize a specific set of proteins under these conditions[42]. This would support a model in which the role of Ede1 in RPACs translation upon stress may be to recruit their mRNAs to cortical actin patches, so they are near ER-associated ribosomes for translation. As the ER has been reported to be an important site for 20S proteasome assembly, it is possible to imagine that proteasome components are co-translationally assembled at the surface of the ER membrane[50].

Multiple stresses impact upon the actin cytoskeleton, leading it to be proposed as a biosensor for detecting stress[51]. In this work, we showed that rapamycin and Lat-B treatment perturb the actin cytoskeleton and re-localize a stress-responsive mRNA to cortical actin patches. This could potentially allow local translation of the mRNA throughout the mother cell, helping it to cope with the stress before resuming cell growth. While it is becoming clear that local and selective translation is crucially important in regulating cellular functions, less is known about how it is regulated under stress. This work illustrates that actin remodelling controls mRNA localization and translation under stress, helping to adapt the proteome to environmental and physiological challenges. As perturbation of the actin cytoskeleton has been associated with various human diseases such as cancer and autoimmunity, it will be important to better understand the impact of actin cytoskeleton remodelling in controlling selective translation under pathophysiological conditions.

## Online content

Any methods, additional references, Nature Research reporting summaries, source data, extended data, supplementary information, acknowledgements, author contributions and competing interests; and statements of

data and code availability are available at https://doi.org/10.1038/s41556-022-00938-4.

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

## Methods

**Yeast strains, plasmids and growth conditions.** Yeast strains and plasmids used in this study are listed in Supplementary Table 1. All plasmids during this study were made with In-Fusion HD Cloning Plus (Takara, 638909). All yeast strains are isogenic to BY4741. All gene deletions were created using the PCR-based integration system[52] and verified by PCR analysis. Cells were cultured in YEPD (yeast extract peptone) (2% glucose) or SC (synthetic medium) (2% glucose) lacking appropriate amino acids for selection. For yeast treatment, cells were grown on plates overnight at 30 °C, resuspended to $OD_{600nm}$ 0.2 in YEPD and grown at 30 °C until they reached $OD_{600nm}$ 0.5–0.8 to ensure exponential growth. Cells were diluted back to $OD_{600nm}$ 0.2 before being treated with either 200 nM rapamycin (LC Laboratories, R-5000) or 25 μM Lat-B (Abcam, ab144291), or shifted to non-permissive temperature for the indicated time. To assess growth using drop assays, yeast strains were adjusted to $OD_{600nm}$ 0.2 from freshly streaked yeast and 5 μl of 1/5 serial dilutions spotted on YEPD plates with or without 20 ng ml⁻¹ rapamycin. Plates were imaged after 3 days at 30 °C using the Chemidoc Touch imaging system (Bio-Rad) on colorimetric setting.

**Generation of ADC17-70ntΔ strain by CRISPR/CAS9.** Guide RNA designed to cut close to the 70-nucleotide region upstream of the ADC17 start codon (agtaacataatgtgctcagcagg) was inserted into pML104 (Addgene number 67638) to generate pML104-ADC17-70nt. Repair templates to remove the 70-nucleotide upstream region consisting of the flanking region (40 bp) of the 70-nuceotide sequence upstream of ADC17 start codon (f: tcaccaggaaaacaatacttcagaagcttatttctctttgaatgtgctcagcagccggtatcagaagaccaatccagatcg and r: cgatctggattggtcttctgataccggctgctgagcacattcaagagaaataagcttctgaa gtattgttttcctggtga) were generated. Additional mutations were inserted to disrupt the PAM site and prevent any further cleavage by Cas9. The repair template was made by annealing (f) and (r) oligo nucleotides in annealing buffer (10 mM (f) oligo, 10 mM (r) oligo, 50 mM NaCl, 10 mM Tris–HCl 7.8, 10 mM MgCl₂ and 100 mg ml⁻¹ BSA). The annealing reaction was incubated at 95 °C for 6 min, followed by a gradual decrease of the temperature to 25 °C (1.5 °C min⁻¹). pML104-ADC17-70nt was co-transformed in yeast with 10 pmol of annealed repair template, as described above. The clones were verified by PCR using flanking primers and confirmed by sequencing.

**Yeast protein extraction.** Yeast samples were pelleted (3,200g, 4 °C, 4 min). Pellets were washed in 500 μl ice-cold MilliQ water (6,200g, 30 s) and either flash-frozen in dry ice and stored at −20 °C for extraction the following day, or extracted immediately. Pellets were resuspended on ice in 400 μl ice-cold 2 M LiAc for 1 min and spun down. Then the supernatant was removed, and pellets were resuspended on ice in 400 μl ice-cold 0.4 M NaOH for 1 min and spun down again. Pellets were resuspended in 120 μl lysis buffer (0.1 M NaOH, 0.05 M EDTA, 2% SDS, 2% β-mercaptoethanol, PhosStop (Roche) and cOmplete protease inhibitor cocktail (Roche)) and boiled for 10 min. Then, 3 μl 4 M acetic acid was added, and samples were vortexed and boiled for 10 min. Samples were vortexed and spun down (17,000g, 15 min), after which 80 μl supernatant was added to 20 μl 5× loading buffer (0.25 M Tris–HCl pH 6.8, 50% glycerol, 0.05% bromophenol blue) and the remainder was used to quantify protein concentration using NanoDrop ($A_{280nm}$, Thermo). Samples were adjusted to the same concentration and stored at −20 °C.

**Western blot analyses.** Samples were run on homemade 6–14% Bis-Tris acrylamide gels. Two tubes of 6 ml mix were made up of 6% and 14% acrylamide, respectively (0.33 M pH 6.5 Bis-Tris, 0.083% APS and 0.083% TEMED). Then, 600 μl of the 14% solution was added to 2 × 1 mm Mini-Protean casting gels (Bio-Rad). The 14% solution was diluted with 600 μl of the 6% solution, and a further 600 μl was added to the casting gels. This was repeated until the gels were complete and a comb was added. After polymerization, extracts were loaded to a total of 25–50 μg protein per lane and run at 120 V for 2.5 h at 4 °C. Gels were then transferred onto 0.2 μm nitrocellulose membrane (Bio-Rad, 1620112) using a TransBlot Turbo (Bio-Rad) (30 min, 2.5 Amp). Membranes were stained with Ponceau S solution (Santa Cruz, sc-301558), imaged, cut, washed in TBS and blotted for at least 1 h with TBS containing 5% milk, washed three times in TBS-Tween and incubated with primary antibody overnight. Primary antibody was removed, and membranes were washed three times in TBS-T, incubated with secondary antibody for 1 h, washed three times in TBS-T Tween and imaged on a Chemidoc Touch imaging system (Bio-Rad) using Clarity ECL (Bio-Rad, 170-5061). Where indicated, blots were quantified by densitometry using FIJI with expression of the protein of interest normalized to Ponceau staining (loading control).

**Antibody dilutions.** Anti-Adc17 (Bertolotti laboratory; rabbit; 1:1,000), anti-Adc17-(2) (DSTT; sheep; 1:250; DU66321), anti-Nas6 (Abcam; rabbit; 1:2,000; ab91447), anti-Flag (Sigma Aldrich; mouse; 1:2,000; F3165), anti-Rpt5 (Enzo Life Sciences; rabbit; 1:5,000; PW8245), anti-20S (Enzo Life Sciences; rabbit; 1:2,000; PW9355), anti-Mpk1 (Santa Cruz; mouse; 1:500; sc-374434), anti-p-Mpk1 (Cell Signaling Technology; rabbit; 1:1,000; 9101) and anti-p-Rps6 (Cell Signaling Technology; rabbit; 1:1,000; 2211). Rabbit anti-Adc17 antibody from Bertolotti laboratory was used in all figures except for Figs. 1f and 7f,j and Extended Data Fig.

4b, where sheep anti-Adc17 antibody was used instead owing to stock availability. Anti-mouse IgG, HRP-linked antibody (Cell Signaling Technology; 1:10,000; #7076) and anti-rabbit IgG, HRP-linked antibody (Cell Signaling Technology; 1:10,000; #7074).

**Proteasome activity assays.** Yeast was grown in YEPD medium overnight at 30 °C, then diluted to $OD_{600nm}$ 0.2 in 30 ml YEPD, grown at 32 °C to $OD_{600nm}$ 0.5–0.7 and then diluted back to $OD_{600nm}$ 0.2. Treatments were then performed (30 ml with 200 nM rapamycin, 20 ml with 25 μM Lat-B and 20 ml untreated control), and cells returned to 32 °C for 3 h. Then, 15 $OD_{600nm}$ of cells were spun down (3,200g, 4 °C, 4 min), resuspended in 800 μl ice-cold water, transferred to a 2 ml tube and spun down again (6,200g, 30 s, 4 °C). The pellet was resuspended in 300 μl native lysis buffer (50 mM Tris pH 8, 5 mM MgCl₂, 0.5 mM EDTA, 5% glycerol, 1 mM DTT and 5 mM ATP) and lysed with 250 μl acid-washed beads (Sigma, G-8772) (FastPrep 24, MP, 3 × 30 s on, 5 min off). Beads and cell debris were removed by centrifugation (17,000g, 2 min, 4 °C), the supernatant was transferred to a fresh tube and centrifuged again (17,000g, 10 min, 4 °C). Protein concentration was determined on a NanoDrop ($A_{280nm}$, Thermo) and standardized between samples. Then, 75 μg protein in 1× native sample buffer (50 mM Tris–HCl pH 6.8, 10% glycerol and 0.01% bromophenol blue) was loaded onto 1.5 mm 3.8–5% acrylamide gradient native gels (prepared in duplicate as for the western blot gels (above), using 10 ml solutions of 90 mM Tris, 90 mM boric acid, 2 mM MgCl₂, 1 mM DTT, 0.12% APS and 0.12% TEMED with acrylamide added to either 3.8% or 5%). Gels were run for 2.5 h, at 110 V and 4 °C, in ice-cold native running buffer (0.9 M Tris, 0.9 M boric acid, 2 mM MgCl₂, 1 mM ATP and 1 mM DTT). The gels were incubated in 15 ml assay buffer (50 mM Tris–HCl pH 7.5, 150 mM NaCl, 5 mM MgCl₂ and 10% glycerol) containing 100 μl 10 mM suc-LLVY-AMC fluorogenic substrate (Cambridge Biosciences, 4011369), 30 °C in the dark for 15–20 min and imaged using a Chemidoc Touch imaging system (Bio-Rad). To image CP assembly, SDS was added to the assay buffer at a final concentration of 0.05% and the gel was re-incubated for 15 min before imaging again. Gels were transferred onto 0.2 μm nitrocellulose membrane (Bio-Rad; 1620112) using a TransBlot Turbo (Bio-Rad) for western blot analysis.

**Fluorescence microscopy.** Yeast was grown on YEPD plates overnight at 30 °C, resuspended to $OD_{600nm}$ 0.2 in YEPD medium and grown at 30 °C to $OD_{600nm}$ 0.5–0.7. Cultures were split into 4 ml samples, rapamycin (200 nM final) or Lat-B (25 μM final) was added and samples were returned to 30 °C for the indicated time. Formaldehyde (Sigma Aldrich; F8775) was then added to 3.7%, and the sample returned to 30 °C for 20 min. Samples were spun down (3,200g, 4 min), washed twice with 7 ml PBS, transferred to a 1.5 ml tube, spun down (7,800g, 2 min), resuspended in 100 μl PBS/0.1% Triton X-100 containing 1:1,000 dilution of rhodamine phalloidin (Abcam; ab255138) or phalloidin-iFluor 647 (Abcam; ab176759) and incubated at 4 °C on a Stuart SB3 vertical rotator in the dark. After 1 h, samples were spun down as before, washed in 1 ml PBS and resuspended in 10 μl ProLong Glass antifade (Thermo Fisher, P36980). Then, 3–4 μl was mounted on a SuperFrost microscope slide (VWR; 631-0847), covered with a glass cover slip (VWR; 631-0119) and cured in the dark overnight before imaging on a Zeiss 880 Airyscan microscope (Airyscan mode, Alpha Plan-APO 100×/1.46 oil DIC VIS objective (Zeiss) and Alpha Plan-APO 63×/1.4 oil objective (Zeiss)). ZEN 2.3 SP1 FP3 software was used to acquire images. For the quantification of mRNAs per cell, adc17Δ cells with either Ede1 WT or Ede1-aGFP expressing ADC17 mRNA containing MS2 and PP7 stem loops, PCP-GFP and MCP-mCherry were grown and treated with rapamycin as described then fixed and mounted on slides as described, but without phalloidin staining.

For live-cell imaging, 2–3 ml of logarithmically growing yeast cells in DOA medium was added to a 35 mm FluoroDish (Fisher Scientific, 15199112) that had been pre-incubated at 30 °C with concanavalin A (Sigma Aldrich, C2010) and allowed to attach for 0.5–1 h. Plates were washed twice with 2 ml medium to remove unadhered cells and imaged on a Zeiss 880 Airyscan microscope (Airyscan mode, Alpha Plan-APO 100×/1.46 oil DIC VIS objective (Zeiss, 420792-9800-720)) at 30 °C.

All microscopy analysis was carried out using FIJI. For quantification of co-localization of Ede1, Sla1 and Vrp1 with ADC17 mRNA, protein (red) and mRNA (green) punctae (circular punctae, >0.2 μm diameter, >50% brighter than the local cell background) were detected and assessed for co-localization using the ComDet v.0.5.1 plugin with the standard settings. All detected particles were manually checked. For quantification of co-localization of red (translating mRNA) and green (all mRNA) puncta in SunTag experiments, the ComDet v.0.5.1. plugin was used as above. ADC17 mRNA interaction with actin in fixed cells was analysed by performing a maximum-intensity Z-projection, then the number of PP7-GFP-labelled ADC17 mRNAs (circular punctae, >0.2 μm diameter, >50% brighter than the local cell background) in contact with cortical actin patches (circular punctae, >0.5 μm diameter, twofold brighter than actin cables), actin cables (linear structures, twofold brighter than cell background) and no actin (remaining ADC17 puncta) was counted manually. To quantify polarity, maximum-intensity projections were again performed and budding cells with more than six cortical actin patches in the larger mother cell were counted as depolarized, while those with six or fewer were counted as polarized. To quantify

mRNAs per cell, a standard deviation $Z$-projection was performed and ComDet v.0.5.1 used to detect mRNAs (MCP-mCherry), while cells were counted manually.

**RiboTag RNA isolation.** Rpl10-GFP yeast expressing either FGH17 or FGH17-70ntΔ was grown in YEPD medium overnight at 30 °C, then diluted to $OD_{600nm}$ 0.5 in 50 ml YEPD medium and grown at 30 °C to $OD_{600nm}$ ~1 and diluted back to $OD_{600nm}$ 0.5 before being treated with 200 nM rapamycin for 1.5 h at 30 °C, or remaining untreated. After treatment, polysomes were stabilized by washing cells in 20 ml ice-cold water containing 0.1 mg ml⁻¹ cycloheximide (CHX) before being resuspended in 1 ml RiboTag Lysis Buffer (50 mM Tris pH 7.5, 100 mM KCl, 12 mM MgCl₂, 1% Nonidet P-40, 1 mM DTT, 100 U ml⁻¹ Promega RNasin, 100 mg ml⁻¹ CHX and cOmplete EDTA-free protease inhibitor cocktail (Roche)) and bead lysis was performed using 500 μl acid-washed beads (Sigma Aldrich, G8772) (FastPrep 24, MP, 5 × 30 s on, 5 min off). Ribosome-RNA-containing supernatants were cleared of cell debris by centrifugation (12,000$g$, 10 min, 4 °C). Then, 100 μl slurry GFP binder Sepharose beads (MRC-PPU Reagents) per sample were pre-washed twice in wash buffer (10 mM Tris–HCl, pH 7.5, 0.15 M NaCl and 0.5 mM EDTA), before being added to samples and incubated overnight under gentle inversion at 4 °C. Beads were washed three times for 10 min with gentle rotation in high-salt buffer (50 mM Tris pH 7.5, 300 mM KCl, 12 mM MgCl₂, 1% Nonidet P-40, 1 mM DTT, 100 U ml⁻¹ Promega RNasin, 100 mg ml⁻¹ CHX and cOmplete EDTA-free protease inhibitor cocktail (Roche)). RNA was eluted from beads using Qiagen RLT buffer containing 2-mercaptoethanol and by vortexing 30 s. RNA was isolated using RNeasy Kit (74004, Qiagen) following the manufacturer's instructions, before being analysed by qRT–PCR.

**qRT–PCR.** Total yeast RNA was extracted using RNeasy Kit (74004, Qiagen) following the manufacturer's instructions. Then, 1 μg of RNA from untreated and 1.5 h rapamycin-treated cells prepared as described above was synthesized into complementary DNA using SuperScript III reverse transcriptase (18080093, Thermo Fisher). cDNA was diluted 1:10 before qRT–PCR was performed. qRT–PCR with primers *ALG9* (f): cacggatagtggctttggtgaacaattac, *ALG9* (r): tatgattatctggcagcaggaaagaacttggg, *FGH17* (f): gtcctgctggagttcgtgac, *FGH17* (r): cgtaatctggaacatcgtatggg, *ADC17* (f): cgacgacttggagaacacattg, *ADC17* (r): caatgcgtccactctctcat was performed using PowerUp SYBR Green Master Mix (A25741, Thermo Fisher) on a CFX384 real-time PCR detection system (Bio-Rad). Expression of each gene was normalized to the housekeeping gene *ALG9* and expressed as fold change after 1.5 h rapamycin treatment calculated using the delta-delta Ct method.

**Immunoprecipitation of nascent RPACs using FGH construct.** Yeast expressing FGH or FGH17-70ntΔ was grown in SC-URA medium overnight at 30 °C, then diluted to $OD_{600nm}$ 0.2 in 100 ml SC-URA medium and grown at 30 °C to $OD_{600nm}$ ~1 and diluted back to $OD_{600nm}$ 0.5 before being treated with 200 nM rapamycin for 1.5 h at 30 °C. After treatment, ribosomes were locked on mRNAs by adding 0.1 mg ml⁻¹ cycloheximide (CHX) (final concentration) to the cultures and immediately incubated 10 min on ice, collected by centrifugation at 4 °C (3,200$g$, 4 min), washed in 20 ml ice-cold water containing 0.1 mg ml⁻¹ CHX and resuspended in 20 ml of the same solution. Proteins were crosslinked to RNA by treating cells with 254 nm UV (1,200 mJ cm⁻² total; 2 × 6,000 mJ cm⁻² with 2 min off in between on ice), then spun down as before. Cells were resuspended in 1 ml lysis buffer (0.1 M Tris–HCl pH 7.5, 0.5 M LiCl, 10 mM EDTA, 1% Triton-X100, 5 mM DTT, 100 U ml⁻¹ RNasin (Promega, N2611) and cOmplete EDTA-free protease inhibitor cocktail (Roche)), and bead lysis was performed using 500 μl acid-washed beads (Sigma Aldrich, G8772) (4 °C, 5 × 2 min on, 2 min off, using a Disruptor Genie). The supernatant was cleared of cell debris by centrifugation (17,000$g$, 15 min, 4 °C). Protein concentration was determined on a NanoDrop ($A_{280nm}$, Thermo), and protein concentration was standardized between samples. Per sample, 60 μl M2 anti-Flag beads (Sigma Aldrich, M8823) was pre-washed twice in wash buffer (10 mM Tris–HCl pH 7.5, 0.6 M LiCl and 1 mM EDTA), then incubated with 1 mg of lysates for 1 h at 4 °C under rotation. M2-anti-Flag beads were then washed once with lysis buffer, twice with wash buffer. RNase elution was then performed with 100 μl elution buffer (10 mM Tris–HCl pH 7.5, 1 mM MgCl₂ and 40 mM NaCl) containing 5 μl of RNase A/T1 mix (Thermo Fisher, EN0551) and placed at 37 °C for 1 h under agitation. Elution fractions were subjected to tryptic digestion and TMT-based quantitative proteomics (see below).

**Tryptic digestion of RNase elution.** In total, 25 μl of protein denaturation buffer (8 M urea, 50 mM ammonium bicarbonate pH 8.0 and 5 mM DTT) was added to each RNase eluent, which was denatured at 45 °C for 30 min with gentle shaking (Eppendorf, Thermomixer C, 800 rpm). Samples were centrifuged at 5,000$g$ for 1 min and cooled to room temperature. Each sample was then incubated with iodoacetamide (10 mM final concentration) in the dark at room temperature. Unreacted iodoacetamide was then quenched with DTT (5 mM final concentration). Each sample was digested using 0.4 μg trypsin at 37 °C overnight and under agitation. The digestion was stopped by adding trifluoroacetic acid (TFA) to the final 0.2% TFA concentration (v/v), centrifuged at 10,000$g$ for 2 min at room temperature. The supernatant was de-salted on ultra-microspin column silica C18 (The Nest Group). De-salted peptides were dried using a SpeedVac vacuum centrifuge concentrator (Thermo Fisher) before TMT labelling.

**TMT labelling.** Each vacuum-dried sample was resuspended in 50 μl of 100 mM TEAB buffer. The TMT labelling reagents were equilibrated to room temperature, and 41 μl anhydrous acetonitrile was added to each reagent channel and gently vortexed for 10 min. Then, 4 μl of each TMT reagent was added to the corresponding sample and labelling was performed at room temperature for 1 h with shaking before quenching with 1 μl of 5% hydroxylamine, after which 2 μl of labelled sample from each channel was analysed by liquid chromatography with tandem mass spectrometry (LC–MS/MS) to ensure complete labelling before mixing. After evaluation, the complete TMT-labelled samples were combined, acidified and dried. The mixture was then de-salted with ultra-microspin column silica C18, and the eluent from C18 column was dried.

**LC–MS/MS analysis.** LC separations were performed with a Thermo Dionex Ultimate 3000 RSLC Nano liquid chromatography instrument. The dried peptides were dissolved in 0.1% formic acid and then loaded on C18 trap column with 3% ACN/0.1%TFA at a flow rate of 5 μl min⁻¹. Peptide separations were performed using EASY-Spray columns (C18, 2 μm, 75 μm × 50 cm) with an integrated nano electrospray emitter at a flow rate of 300 nl min⁻¹. Peptides were separated with a 180 min segmented gradient as follows: the first ten fractions starting from ~7–32% buffer B in 130 min, ~32–45% in 20 min and ~45–95% in 10 min. Peptides eluted from the column were analysed on an Orbitrap Fusion Lumos (Thermo Fisher Scientific) mass spectrometer. Spray voltage was set to 2 kV, RF lens level was set at 30%, and ion transfer tube temperature was set to 275 °C. The Orbitrap Fusion Lumos was operated in positive-ion data-dependent mode with high-resolution MS2 for reporter ion quantitation. The mass spectrometer was operated in data-dependent top speed mode with 3 s per cycle. The full scan was performed in the range of 350–1,500 $m/z$ at nominal resolution of 120,000 at 200 $m/z$ and AGC set to 4×10⁵ with maximal injection time 50 ms, followed by selection of the most intense ions above an intensity threshold of 5×10⁴ for high collision-induced dissociation (HCD)-MS2 fragmentation in the HCD cell with 38% normalized collision energy. The isolation width was set to 1.2 $m/z$ with no offset. Dynamic exclusion was set to 60 s. Monoisotopic precursor selection was set to peptide. Charge states between 2 and 7 were included for MS2 fragmentation. The MS2 scan was performed in the Orbitrap using 50,000 resolving power with auto normal range scan from $m/z$ 100–500 and AGC target of 5×10⁴. The maximal injection time for MS2 scan was set to 120 ms.

**Proteomic data analysis.** All the acquired LC–MS data were analysed using Proteome Discoverer v.2.2 (Thermo Fisher Scientific) with Mascot search engine. Maximum missed cleavage for trypsin digestion was set to 2. Precursor mass tolerance was set to 10 ppm. Fragment ion tolerance was set to 0.02 Da. Carbamidomethylation on cysteine (+57.021 Da) and TMT-10plex tags on N-termini as well as lysine (+229.163 Da) were set as static modifications. Variable modifications were set as oxidation on methionine (+15.995 Da) and phosphorylation on serine, threonine and tyrosine (+79.966 Da). Data were searched against a complete UniProt *Saccharomyces cerevisiae* (reviewed 6,721 entries downloaded February 2018). Peptide spectral match error rates with a 1% false discovery rate were determined using the forward-decoy strategy modelling true and false matches.

Both unique and razor peptides were used for quantitation. Reporter ion abundances were corrected for isotopic impurities on the basis of the manufacturer's data sheets. Reporter ions were quantified from MS2 scans using an integration tolerance of 20 ppm with the most confident centroid setting. Signal-to-noise (S/N) values were used to represent the reporter ion abundance with a co-isolation threshold of 50% and an average reporter S/N threshold of 10 and above required for quantitation from each MS2 spectra to be used. The S/N value of each reporter ion from each peptide spectral match was used to represent the abundance of the identified peptides. The summed abundance of quantified peptides was used for protein quantitation. The total peptide amount was used for the normalization. Protein ratios were calculated from medians of summed sample abundances of replicate groups. Standard deviation was calculated from all biological replicate values. The standard deviation of all biological replicates lower than 25% was used for further analyses. Multiple unpaired $t$-test was used to determine the significant differences between FGH or FGH17-70ntΔ.

**Statistics and reproducibility.** Each experiment was repeated independently a minimum of three times, as indicated. The standard deviation (s.d.) of the mean of at least four independent experiments is shown in the graphs, or as indicated. $P$ values are as stated, or not significant (NS). $P$ values were obtained from one-way analysis of variance (ANOVA) (Dunnett multiple comparison test: Figs. 5f,h and 6b), two-way ANOVA (Tukey multiple comparison test: Figs. 3f, 4c and 7c,g,i,k, and Extended Data Fig. 5e) or unpaired two-tailed Student's $t$-test (Figs. 1b,e and 3c, and Extended Data Figs. 1c, 3a, 4d and 3b) to probe for statistical significance. All statistics were performed using Graph Pad Prism 9 software (version 9.1.2) (Graph Pad Software Inc.). No statistical method was used to pre-determine sample size. No data were excluded from the analyses, and the experiments were not randomized. The investigators were not blinded to allocation during experiments or outcome assessment.

**Reporting summary.** Further information on research design is available in the Nature Research Reporting Summary linked to this article.

## Data availability

All the data generated or analysed during the current study are included in this published article and its supplementary files (Supplementary Information and source data). The mass spectrometry proteomics data have been deposited to the ProteomeXchange Consortium via the PRIDE partner repository with the dataset identifier PXD027655. All other data supporting the findings of this study are available from the corresponding author on reasonable request. Source data are provided with this paper.

## References

52. Janke, C. et al. A versatile toolbox for PCR-based tagging of yeast genes: new fluorescent proteins, more markers and promoter substitution cassettes. *Yeast* **21**, 947–962 (2004).

## Acknowledgements

We thank A. Bertolotti for the gift of Adc17 antibody. We thank the MRC PPU reagents and services, including Cloning team and DNA sequencing services, MRC-PPU Mass Spectrometry and the Dundee Imaging facility. We thank K. Labib and D. Alessi for reading and discussing the manuscript. This research was supported by the Medical Research Council (grant number MC_UU_00018/8 to A.R.).

## Author contributions

A.R. and T.W. designed experiments and wrote the manuscript. A.R., T.W., R.C., A.A., A.B. and H.Z performed experiments and analysed data. All authors edited the manuscript.

## Competing interests

The authors declare no competing interests.

## Additional information

**Extended data** is available for this paper at https://doi.org/10.1038/s41556-022-00938-4.

**Correspondence and requests for materials** should be addressed to Adrien Rousseau.

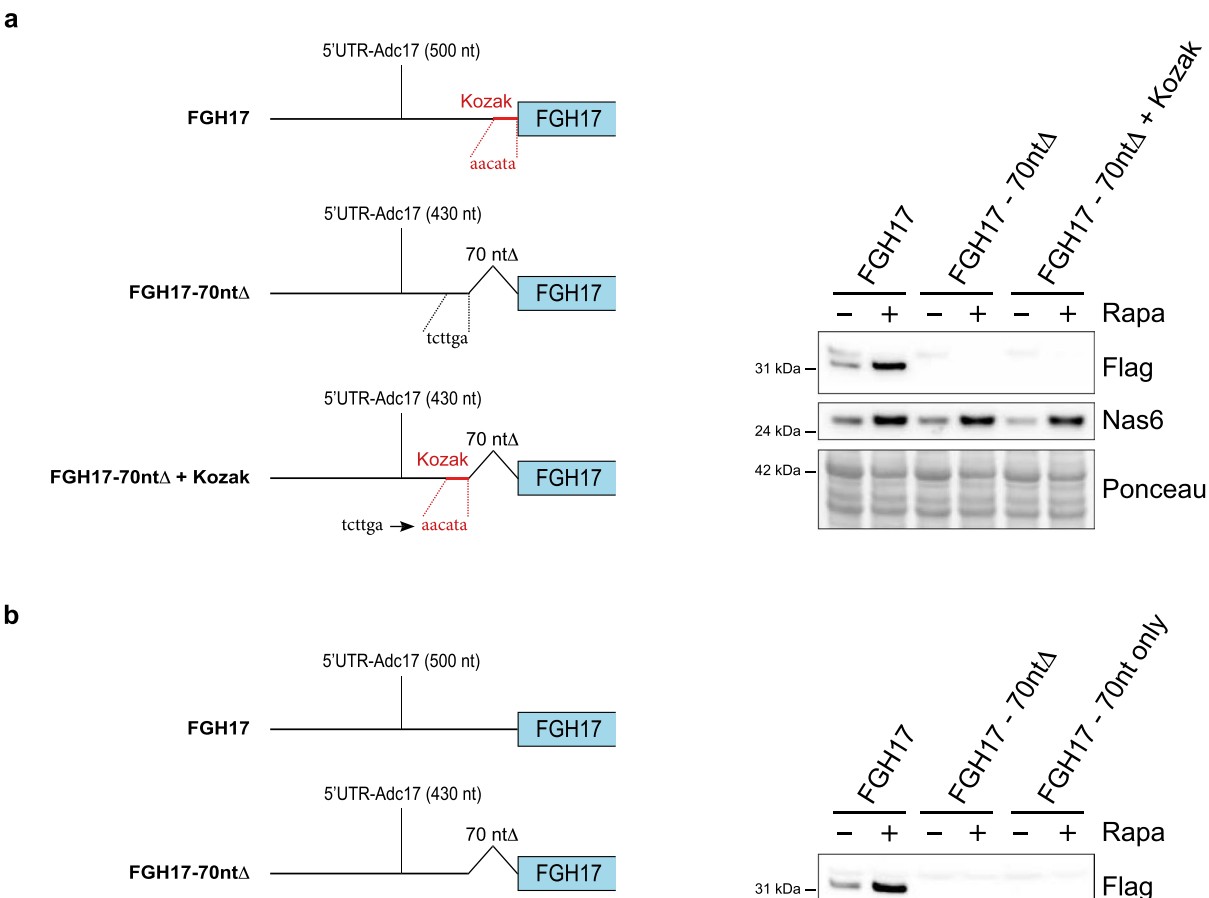

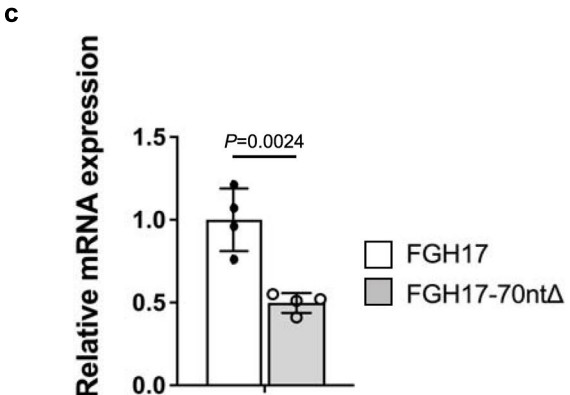

**Extended Data Fig. 1 | Characterisation of the region important for FGH17 regulation. a**, Schematic showing the introduction of the *ADC17* 5′UTR Kozak sequence into the FGH17-70ntΔ vector. Western blot analysis showing the impact on FGH17 levels in cells treated ± 200 nM rapamycin (Rapa) for 4 h. Ponceau S staining was used as a loading control. **b**, FGH17 vectors with the full 5′UTR, the 5′UTR lacking the 70 nucleotides upstream of the start codon (FGH17-70ntΔ) or containing only the 70 nucleotides upstream of the start codon (FGH17-70nt only). Western blot analysis showing the impact on FGH17 levels in cells treated ± 200 nM rapamycin (Rapa) for 4 h. Ponceau S staining was used as a loading control. **c**, Relative abundance of *FGH17* and *FGH17-70ntΔ* mRNA (mRNA of interest normalised to *ALG9* housekeeping mRNA) in yeast cells. Data are presented as mean ± s.d., n = 4 biologically independent experiments. Statistical analysis was carried out using unpaired two-tailed Student's *t*-test. **a, b**, Data are representative of three independent biological replicates.

**a**

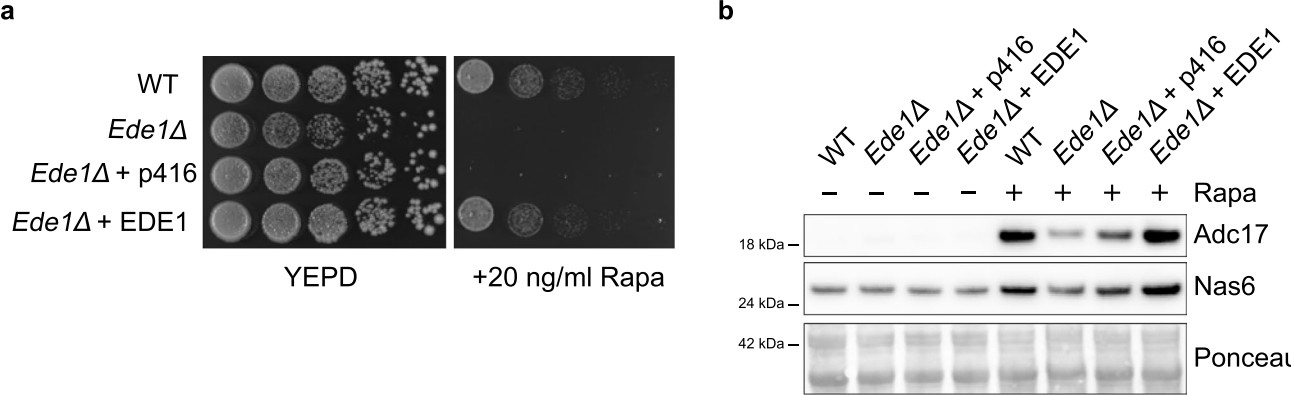

**b**

**c**

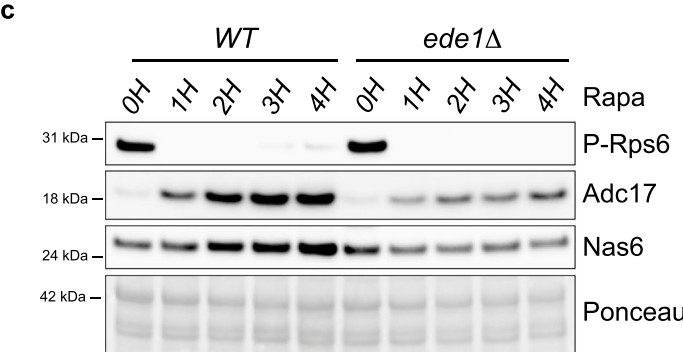

**Extended Data Fig. 2 | Ede1 regulates RPAC levels downstream of TORC1 inhibition. a**, Cells spotted in a fivefold dilution and grown for 3 days on plates ± 20 ng/ml rapamycin. **b**, Western blot analysis of RPACs in WT and *ede1Δ* cells treated ± 200 nM rapamycin (Rapa) for 4 h. Ponceau S staining was used as a loading control. **c**, Western blot analysis of RPACs and P-Rps6 in WT and *ede1Δ* cells treated ± 200 nM rapamycin (Rapa) for the indicated time. Ponceau S staining was used as a loading control. **a-c**, Data are representative of three independent biological replicates.

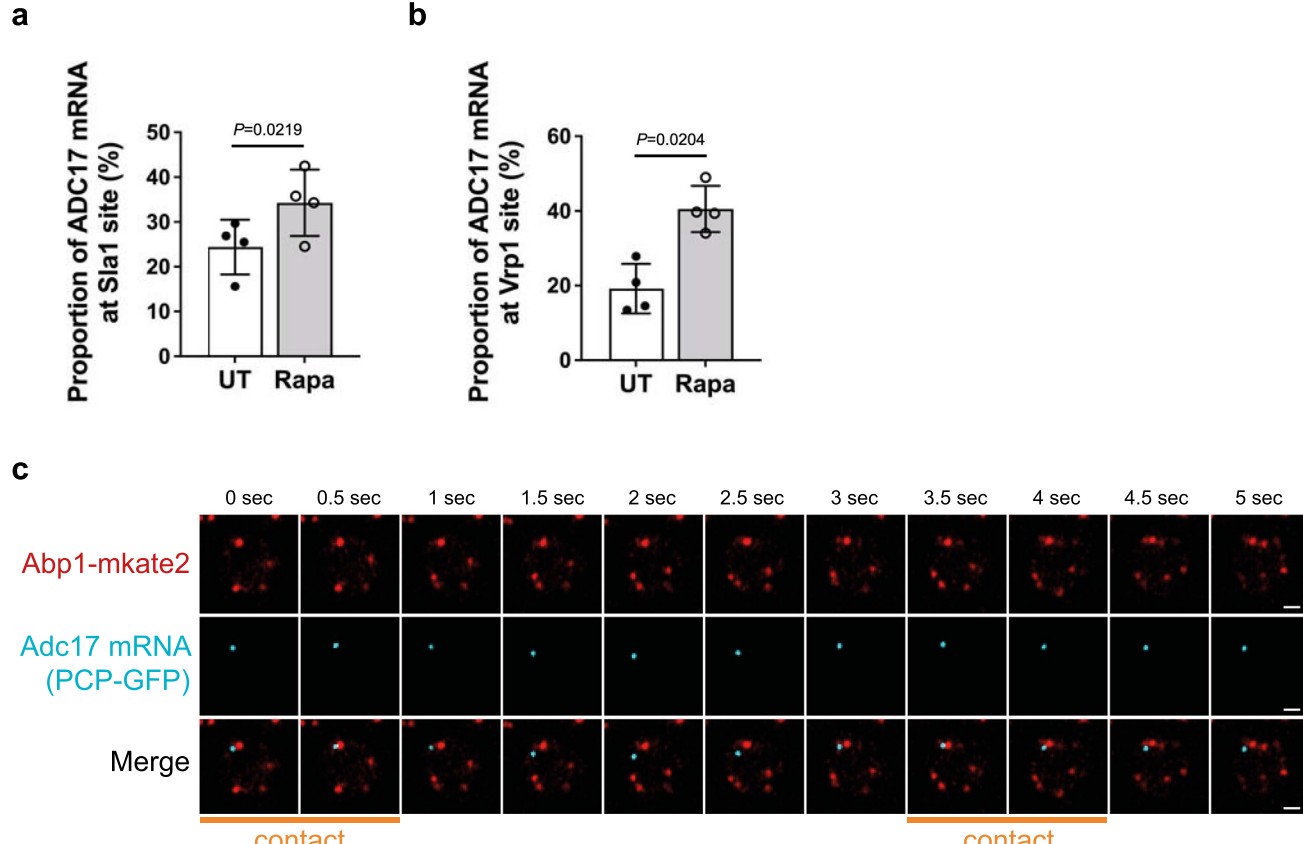

**Extended Data Fig. 3 | *ADC17* mRNA interacts with cortical actin patch proteins. a**, Frequency of *ADC17* mRNAs colocalizing with Sla1-mKate2 in cells grown for 3 h ± 200 nM rapamycin (Rapa). Untreated (UT). Data are presented as mean ± s.d., n = 4 biologically independent experiments. (n = 529 *ADC17* mRNAs per condition). Statistical analysis was carried out using unpaired two-tailed Student's *t*-test. **b**, Frequency of *ADC17* mRNAs colocalizing with Vrp1-mKate2 in cells grown for 3 h ± 200 nM rapamycin (Rapa). Untreated (UT). Data are presented as mean ± s.d., n = 4 biologically independent experiments. (n = 408 *ADC17* mRNAs per condition). Statistical analysis was carried out using unpaired two-tailed Student's *t*-test. **c**, Representative single frames from time-lapse imaging showing contacts between Abp1-mKate2 (red) and PCP-GFP-labelled *ADC17* mRNA (cyan). Scale bars, 1 μm. n = 3 biologically independent experiments.

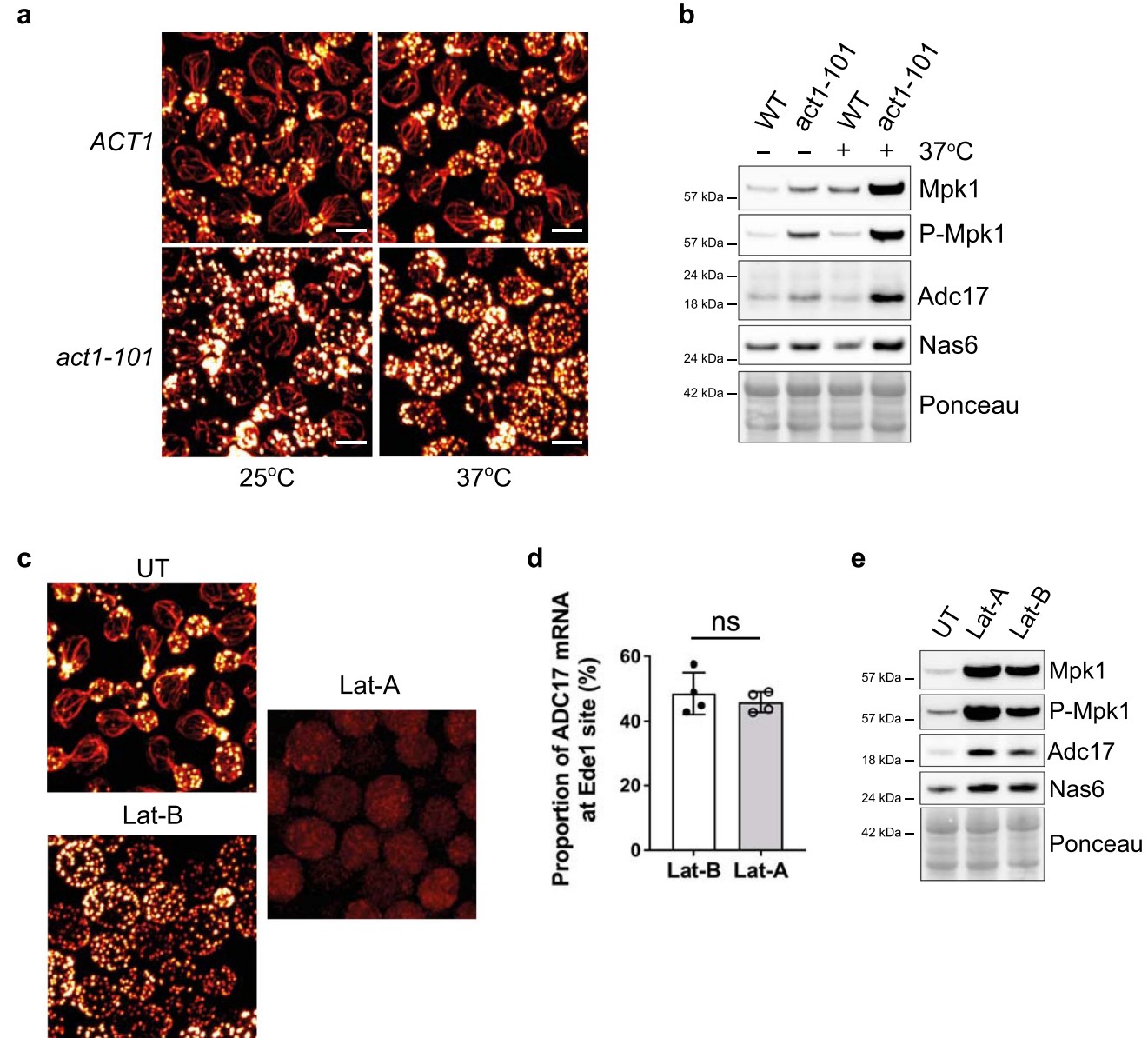

**Extended Data Fig. 4 | Genetic and chemical disruptions of actin induces RPAC levels.** Representative microscopy images (maximum intensity Z-projection) of WT (*ACT1*) and *act1-101* cells either grown at the permissive temperature (25 °C) or shifted to the non-permissive temperature (37 °C) for 4 h and stained with Rhodamine phalloidin to visualise actin (hot red LUT). Scale bars, 3 μm. *n* = 3 biologically independent experiments. **b**, Western blot analysis of RPACs in WT and *act1-101* cells either grown at the permissive temperature (25 °C) or shifted to the non-permissive temperature (37 °C) for 4 h. Ponceau S staining was used as a loading control. *n* = 3 biologically independent experiments. **c**, Representative microscopy images (maximum intensity Z-projection) of WT cells treated or not with either 25 μM Latrunculin-B or 12.5 μM Latrunculin-A for 3 h and stained for actin (hot red LUT). Scale bars, 3 mm. *n* = 3 biologically independent experiments. **d**, Frequency of *ADC17* mRNAs colocalising with Ede1-tdimer2 in cells treated with either 25 μM Latrunculin-B (Lat-B) or 12.5 μM Latrunculin-A (Lat-A) for 3 h. Data are presented as mean ± s.d., *n* = 4 biologically independent experiments (>500 *ADC17* mRNAs per condition). Statistical analysis was carried out using unpaired two-tailed Student's *t*-test. ns not significant. **e**, Western blot analysis of RPACs and Mpk1 kinase in WT cells treated or not with either 25 μM Latrunculin-B (Lat-B) or 12.5 μM Latrunculin-A (Lat-A) for 3 h. Ponceau S staining was used as a loading control. *n* = 3 biologically independent experiments.

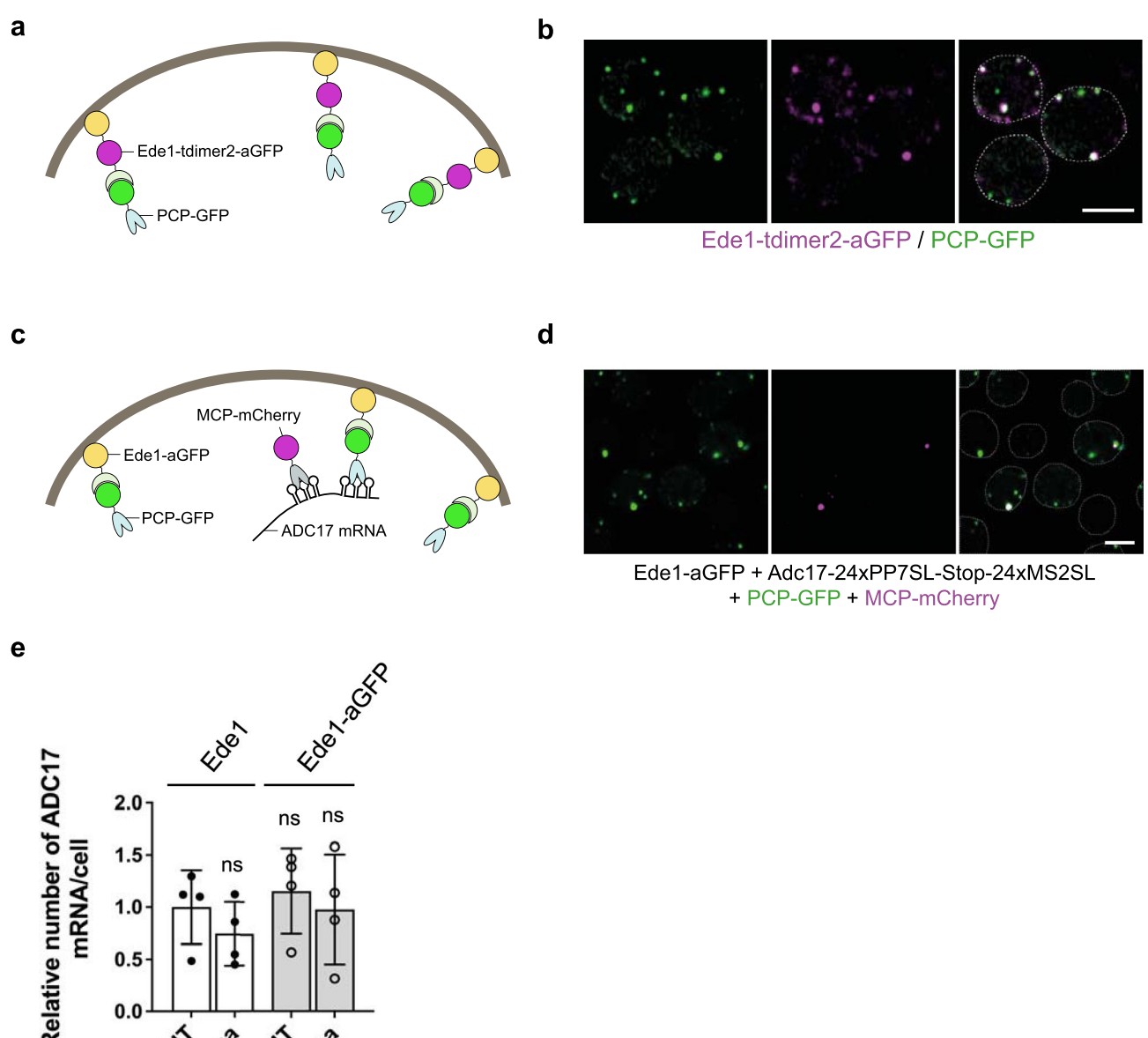

**Extended Data Fig. 5 | Validation of *ADC17* mRNA tethering to Ede1. a**, Schematic representation of PCP-GFP recruitment to Ede1 fused with tdimer2 and aGFP (nanobody against GFP). **b**, Representative microscopy images of yeast cells containing PCP-GFP (green) and Ede1-tdimer2 (magenta) tagged with a nanobody against GFP (Ede1-tdimer2-aGFP). Scale bars, 3 μm. *n* = 3 biologically independent experiments. **c**, Schematic representation of doubly tagged *ADC17* mRNA (magenta) recruitment to Ede1-aGFP/PCP-GFP complex. **d**, Representative microscopy images of yeast cells expressing PCP-GFP (green), MCP-mCherry (magenta), Ede1 tagged with a nanobody against GFP (Ede1-aGFP) and *ADC17* mRNA containing PP7-stem-loops (PP7SL) and MS2-stem-loops (MS2SL) which are recognised by PCP-GFP and MCP-mCherry, respectively. Scale bars, 3 μm. *n* = 3 biologically independent experiments. **e**, Quantification of the number of *ADC17* mRNAs per cell in *adc17*Δ Ede1 WT and Ede1-aGFP cells expressing *ADC17* mRNA containing MS2 and PCP stem loops, PP7-GFP and MCP-mCherry grown for 2 h ± 200 nM rapamycin (Rapa). Untreated (UT). mRNAs were quantified using MCP-mCherry as a marker. Data are presented as mean ± s.d., *n* = 4 biologically independent experiments. (*n* = 261 *ADC17* mRNAs per condition). Statistical analysis was carried out using two-way ANOVA t-test (Tukey multiple comparison test). ns, not significant.

# Reporting Summary

## Statistics

For all statistical analyses, confirm that the following items are present in the figure legend, table legend, main text, or Methods section.

| n/a | Confirmed | |
|---|---|---|
| ☐ | ☒ | The exact sample size (*n*) for each experimental group/condition, given as a discrete number and unit of measurement |
| ☐ | ☒ | A statement on whether measurements were taken from distinct samples or whether the same sample was measured repeatedly |
| ☐ | ☒ | The statistical test(s) used AND whether they are one- or two-sided *Only common tests should be described solely by name; describe more complex techniques in the Methods section.* |
| ☒ | ☐ | A description of all covariates tested |
| ☒ | ☐ | A description of any assumptions or corrections, such as tests of normality and adjustment for multiple comparisons |
| ☐ | ☒ | A full description of the statistical parameters including central tendency (e.g. means) or other basic estimates (e.g. regression coefficient) AND variation (e.g. standard deviation) or associated estimates of uncertainty (e.g. confidence intervals) |
| ☐ | ☒ | For null hypothesis testing, the test statistic (e.g. *F*, *t*, *r*) with confidence intervals, effect sizes, degrees of freedom and *P* value noted *Give P values as exact values whenever suitable.* |
| ☒ | ☐ | For Bayesian analysis, information on the choice of priors and Markov chain Monte Carlo settings |
| ☒ | ☐ | For hierarchical and complex designs, identification of the appropriate level for tests and full reporting of outcomes |
| ☒ | ☐ | Estimates of effect sizes (e.g. Cohen's *d*, Pearson's *r*), indicating how they were calculated |

*Our web collection on statistics for biologists contains articles on many of the points above.*

## Software and code

Policy information about availability of computer code

| Data collection | ZEN 2.3 SP1 FP3 version 14.0.21.201, Proteome Discoverer software v.2.2, Chemidoc Touch imaging system, Zeiss LSM 880 Airyscan microscope, CFX384 Real-Time PCR Detection system, and Orbitrap Fusion Lumos mass spectrometer. |
|---|---|
| Data analysis | Images were acquired with Zeiss LSM 880 Airy Scan confocal microscope. Images taken by Zeiss confocal microscope were analysed using ZEN 2.3 SP1 FP3 black software (version 14.0.21.201). For quantifications of confocal images and Western Blot, Fiji ImageJ 2.1.0/1.53c software was used. For quantification of colocalization of red (translating mRNA) and green (all mRNA) puncta in Suntag experiments, the ComDet v.0.5.1. plugin was used. Proteomic data acquired by the Orbitrap Fusion Lumos mass spectrometer were analysed with Proteome Discoverer software v.2.2 with Mascot search engine. Gene expression data collected by the CFX384 Real-Time PCR Detection system was analysed with Bio-Rad CFX Maestro 2.0 (Version 5.0.021.0616). All statistics were performed using Graph Pad Prism 9 software (Version 9.1.2) (Graph Pad Software Inc., La Jolla, CA). |

For manuscripts utilizing custom algorithms or software that are central to the research but not yet described in published literature, software must be made available to editors and reviewers. We strongly encourage code deposition in a community repository (e.g. GitHub). See the Nature Portfolio guidelines for submitting code & software for further information.

# Data

Policy information about availability of data

All manuscripts must include a data availability statement. This statement should provide the following information, where applicable:

- Accession codes, unique identifiers, or web links for publicly available datasets
- A description of any restrictions on data availability
- For clinical datasets or third party data, please ensure that the statement adheres to our policy

All the data generated or analysed during the current study are included in this published article and its supplementary files. The mass spectrometry proteomics data have been deposited to the ProteomeXchange Consortium via the PRIDE partner repository with the dataset identifier PXD027655.

# Field-specific reporting

Please select the one below that is the best fit for your research. If you are not sure, read the appropriate sections before making your selection.

☒ Life sciences    ☐ Behavioural & social sciences    ☐ Ecological, evolutionary & environmental sciences

For a reference copy of the document with all sections, see nature.com/documents/nr-reporting-summary-flat.pdf

# Life sciences study design

All studies must disclose on these points even when the disclosure is negative.

| | |
|---|---|
| Sample size | No statistical test or power analysis were performed to predetermine sample size. We defined sample sizes based on routine practice in the similar studying fields. Experiments were repeated three or more time, as indicated. |
| Data exclusions | No data were excluded from the manuscript. |
| Replication | All experiments were replicated at least three time with similar findings. Sample sizes are provided in each figure legend. |
| Randomization | All images were acquired randomly. |
| Blinding | No blinding was use for this study. Experiments and data analyses were performed by the same investigator. |

# Reporting for specific materials, systems and methods

We require information from authors about some types of materials, experimental systems and methods used in many studies. Here, indicate whether each material, system or method listed is relevant to your study. If you are not sure if a list item applies to your research, read the appropriate section before selecting a response.

## Materials & experimental systems

| n/a | Involved in the study |
|---|---|
| ☐ | ☒ Antibodies |
| ☐ | ☒ Eukaryotic cell lines |
| ☒ | ☐ Palaeontology and archaeology |
| ☒ | ☐ Animals and other organisms |
| ☒ | ☐ Human research participants |
| ☒ | ☐ Clinical data |
| ☒ | ☐ Dual use research of concern |

## Methods

| n/a | Involved in the study |
|---|---|
| ☒ | ☐ ChIP-seq |
| ☒ | ☐ Flow cytometry |
| ☒ | ☐ MRI-based neuroimaging |

## Antibodies

| | |
|---|---|
| Antibodies used | Anti-Adc17 (Bertolotti laboratory; Rabbit; 1:1000), anti-Adc17-(2) (DSTT; Sheep; 1:250; DU66321; Fig. 7f,h), anti-Nas6 (Abcam; Rabbit; 1:2000; ab91447), anti-Flag (Sigma Aldrich; Mouse; 1:2000; F3165), anti-Rpt5 (Enzo life sciences; Rabbit; 1:5000; PW8245), anti-20S (Enzo life sciences; Rabbit; 1:2000; PW9355), anti-Mpk1 (Santa Cruz; Mouse; 1:500; sc-374434), anti-p-Mpk1 (Cell Signalling Technology; phospho-p44/42, Rabbit; 1:1000; 9101) and anti-p-Rps6 (Cell Signalling Technology; Rabbit; 1:1000; 2211). Anti-mouse IgG, HRP-linked Antibody (Cell Signalling Technology; 1:10000; #7076) and Anti-rabbit IgG, HRP-linked Antibody (Cell Signalling Technology; 1:10000; #7074). |
| Validation | Validation statement and product literature references are available here:<br><br>Rabbit and sheep anti-Adc17 antibodies, rabbit anti-Nas6 antibody, mouse anti-Mpk1 and rabbit anti-p-Mpk1antibodies have been validated using the respective knock-out yeast strains. Each antibody gave a specific signal at the expected size in WT cells while the |

signal was absent in a strain where the target has been knocked-out. Re-expressing the target on a vector restored the signal, confirming the specificity of detection.

Mouse anti-Flag antibody is a gold standard extensively used in science (https://www.sigmaaldrich.com/GB/en/product/sigma/f3165?gclid=EAIaIQobChMIrtHQisap8gIVzNPtCh1yagiWEAAYASAAEgJopvD_BwE)

Rabbit anti-Rpt5 (https://www.enzolifesciences.com/BML-PW8245/proteasome-19s-rpt5-s6a-subunit-polyclonal-antibody/)

Rabbit anti-20S (https://www.enzolifesciences.com/BML-PW9355/proteasome-20s-core-subunits-polyclonal-antibody/)

Rabbit anti-Nas6 antibody (https://www.abcam.com/nas6-antibody-ab91447.html)

Mouse anti-Mpk1 antibody (https://www.scbt.com/fr/p/mpk1-antibody-d-1)

Mouse anti-p-Mpk1 antibody (https://www.cellsignal.com/products/primary-antibodies/phospho-p44-42-mapk-erk1-2-thr202-tyr204-antibody/9101) is extensively used in publications to monitor Mpk1 activation, including (Torres J et al., 2002; Liu L et al., 2018 and Sellers-Moya et al., 2021)

Rabbit anti-p-Rps6 antibody has been validated in yeast in these publications (Yerlikaya et al,. 2015 and Gonzalez A et al., 2015)

Anti-mouse IgG, HRP-linked Antibody (https://www.cellsignal.com/products/secondary-antibodies/anti-mouse-igg-hrp-linked-antibody/7076)

Anti-rabbit IgG, HRP-linked Antibody (https://www.cellsignal.com/products/secondary-antibodies/anti-rabbit-igg-hrp-linked-antibody/7074)

# Eukaryotic cell lines

Policy information about cell lines

| | |
|---|---|
| Cell line source(s) | BY4741 (MATa his3Δ1 leu2Δ0 met15Δ0 ura3Δ0) Horizon Discovery<br>BY4741 + FGH17 (MATa his3Δ1 leu2Δ0 met15Δ0 ura3Δ0 [p416:FGH17::URA3]) This study<br>BY4741 + FGH17-5'UTRΔ (MATa his3Δ1 leu2Δ0 met15Δ0 ura3Δ0 [p416:FGH17-5'UTRD::URA3]) This study<br>BY4741 + FGH17-3'UTRΔ (MATa his3Δ1 leu2Δ0 met15Δ0 ura3Δ0 [p416:FGH17-3'UTRD::URA3]) This study<br>BY4741 + FGH17-40ntΔ (MATa his3Δ1 leu2Δ0 met15Δ0 ura3Δ0 [p416:FGH17-40ntD::URA3]) This study<br>BY4741 + FGH17-30ntΔ (MATa his3Δ1 leu2Δ0 met15Δ0 ura3Δ0 [p416:FGH17-30ntD::URA3]) This study<br>BY4741 + FGH17-23ntΔ (MATa his3Δ1 leu2Δ0 met15Δ0 ura3Δ0 [p416:FGH17-23ntD::URA3]) This study<br>rps6AΔ (MATa his3Δ1 met15Δ0 ura3Δ0 rps6a::LEU2) This study<br>rps18BΔ (MATa his3Δ1 met15Δ0 ura3Δ0 rps18b::LEU2) This study<br>ede1Δ (MATa his3Δ1 leu2Δ0 met15Δ0 ura3Δ0 ede1::kanMx) Horizon Discovery<br>cup1-1/2Δ (MATa his3Δ1 leu2Δ0 met15Δ0 ura3Δ0 cup1-1/2::kanMx) Horizon Discovery<br>Adc17-24xPP7SL + PCP-mKate2 (MATa his3Δ1 leu2Δ0 met15Δ0 ura3Δ0 ADC17-24xPP7SL-LoxP [pFA6:cyc1p-PCP-mKate2::HIS3]) This study<br>Adc17-24xPP7SL  Ede1-3xHA-GFPEnvy + PCP-mKate2 (MATa his3Δ1 leu2Δ0 met15Δ0 ura3Δ0 ADC17-24xPP7SL-LoxP EDE1-3xHA-GFPENVY:KanMx [pFA6:cyc1p-PCP-mKate2::HIS3]) This study<br>Adc17-SunTag (MATa his3Δ1 leu2Δ0 met15Δ0 ura3Δ0 [p416:adc17p-Adc17-SunTag(24x)-PP7SL(24x)::URA3 + pFA6:cyc1p-PCP-EGFP(2X)-cyc1p-scFV-GCN4-mCherry::HIS3]) This study<br>Adc17-SunTag ede1Δ (MATa his3Δ1 leu2Δ0 met15Δ0 ura3Δ0 ede1::KanMx [p416:adc17p-Adc17-SunTag(24x)-PP7SL(24x)::URA3 + pFA6:cyc1p-PCP-EGFP(2X)-cyc1p-scFV-GCN4-mCherry::HIS3]) This study<br>syp1Δ (MATa his3Δ1 leu2Δ0 met15Δ0 ura3Δ0 syp1::kanMx) Horizon Discovery<br>clc1Δ (MATa his3Δ1 leu2Δ0 met15Δ0 ura3Δ0 clc1::kanMx) Horizon Discovery<br>chc1Δ (MATa his3Δ1 leu2Δ0 met15Δ0 ura3Δ0 chc1::kanMx) Horizon Discovery<br>pal1Δ (MATa his3Δ1 leu2Δ0 met15Δ0 ura3Δ0 pal1::kanMx) Horizon Discovery<br>yap1801Δ (MATa his3Δ1 leu2Δ0 met15Δ0 ura3Δ0 yap1801::kanMx) Horizon Discovery<br>yap1802Δ (MATa his3Δ1 leu2Δ0 met15Δ0 ura3Δ0 yap1802::kanMx) Horizon Discovery<br>alp1Δ (MATa his3Δ1 leu2Δ0 met15Δ0 ura3Δ0 alp1::kanMx) Horizon Discovery<br>alp3Δ (MATa his3Δ1 leu2Δ0 met15Δ0 ura3Δ0 alp3::kanMx) Horizon Discovery<br>aps2Δ (MATa his3Δ1 leu2Δ0 met15Δ0 ura3Δ0 aps2::kanMx) Horizon Discovery<br>apm4Δ (MATa his3Δ1 leu2Δ0 met15Δ0 ura3Δ0 apm4::kanMx) Horizon Discovery<br>ent1Δ (MATa his3Δ1 leu2Δ0 met15Δ0 ura3Δ0 ent1::kanMx) Horizon Discovery<br>ent2Δ (MATa his3Δ1 leu2Δ0 met15Δ0 ura3Δ0 ent2::kanMx) Horizon Discovery<br>end3Δ (MATa his3Δ1 leu2Δ0 met15Δ0 ura3Δ0 end3::kanMx) Horizon Discovery<br>sla1Δ (MATa his3Δ1 leu2Δ0 met15Δ0 ura3Δ0 sla1::kanMx ) Horizon Discovery<br>lsb3Δ (MATa his3Δ1 leu2Δ0 met15Δ0 ura3Δ0 lsb3::kanMx ) Horizon Discovery<br>lsb4Δ (MATa his3Δ1 leu2Δ0 met15Δ0 ura3Δ0 lsb4::kanMx ) Horizon Discovery<br>lsb5Δ (MATa his3Δ1 leu2Δ0 met15Δ0 ura3Δ0 lsb5::kanMx ) Horizon Discovery<br>ubx3Δ (MATa his3Δ1 leu2Δ0 met15Δ0 ura3Δ0 ubx3::kanMx) Horizon Discovery<br>gts1Δ (MATa his3Δ1 leu2Δ0 met15Δ0 ura3Δ0 gst1::kanMx) Horizon Discovery<br>ldb17Δ (MATa his3Δ1 leu2Δ0 met15Δ0 ura3Δ0 ldb17::kanMx) Horizon Discovery<br>bbc1Δ (MATa his3Δ1 leu2Δ0 met15Δ0 ura3Δ0 bbc1::kanMx) Horizon Discovery<br>aim21Δ (MATa his3Δ1 leu2Δ0 met15Δ0 ura3Δ0 aim21::kanMx) Horizon Discovery<br>ubp7Δ (MATa his3Δ1 leu2Δ0 met15Δ0 ura3Δ0 ubp7::kanMx) Horizon Discovery<br>bzz1Δ (MATa his3Δ1 leu2Δ0 met15Δ0 ura3Δ0 bzz1::kanMx) Horizon Discovery<br>vrp1Δ (MATa his3Δ1 leu2Δ0 met15Δ0 ura3Δ0 vrp1::kanMx) Horizon Discovery |

myo3Δ (MATa his3Δ1 leu2Δ0 met15Δ0 ura3Δ0 myo3::kanMx) Horizon Discovery
myo5Δ (MATa his3Δ1 leu2Δ0 met15Δ0 ura3Δ0 myo5::kanMx) Horizon Discovery
rvs161Δ (MATa his3Δ1 leu2Δ0 met15Δ0 ura3Δ0 rvs161::kanMx) Horizon Discovery
rvs167Δ (MATa his3Δ1 leu2Δ0 met15Δ0 ura3Δ0 rvs167::kanMx) Horizon Discovery
vps1Δ (MATa his3Δ1 leu2Δ0 met15Δ0 ura3Δ0 vps1::kanMx) Horizon Discovery
Adc17-24xPP7SL + PCP-GFP(2x) (MATa his3Δ1 leu2Δ0 met15Δ0 ura3Δ0 ADC17-24xPP7SL-LoxP [pFA6:cyc1p-PCP-EGFP(2X)::HIS3]) This study
Adc17-24xPP7SL  Abp140-3xHA-mKate2 + PCP-EGFP(2X) (MATa his3Δ1 leu2Δ0 met15Δ0 ura3Δ0 ADC17-24xPP7SL-LoxP ABP140-3xHA-mKATE2:KanMx [pFA6:cyc1p-PCP-EGFP(2X)::HIS3]) This study
Adc17-24xPP7SL  Abp1-3xHA-mKate2 + PCP-EGFP(2X) (MATa his3Δ1 leu2Δ0 met15Δ0 ura3Δ0 ADC17-24xPP7SL-LoxP ABP1-3xHA-mKATE2:KanMx [pFA6:cyc1p-PCP-EGFP(2X)::HIS3]) This study
Adc17-24xPP7SL  ede1Δ + PCP-EGFP(2X) (MATa his3Δ1 leu2Δ0 met15Δ0 ura3Δ0 ADC17-24xPP7SL-LoxP ede1::LEU2 [pFA6:cyc1p-PCP-EGFP(2X)::HIS3]) This study
Adc17-24xPP7SL Ede1-aGFP + PCP-GFP(2x) (MATa his3Δ1 leu2Δ0 met15Δ0 ura3Δ0 ADC17-24xPP7SL-LoxP EDE1-aGFP:LEU2 [pFA6:cyc1p-PCP-EGFP(2X)::HIS3]) This study
ede1Δ + p416 (MATa his3Δ1 leu2Δ0 met15Δ0 ura3Δ0 ede1::kanMx [p416]) This study
ede1Δ + p416-Ede1 (MATa his3Δ1 leu2Δ0 met15Δ0 ura3Δ0 ede1::kanMx [p416-ede1p-Ede1]) This study
Adc17-24xPP7SL Ede1-tdimer2-aGFP + PCP-GFP(2x) (MATa his3Δ1 leu2Δ0 met15Δ0 ura3Δ0 ADC17-24xPP7SL-LoxP EDE1-aGFP:LEU2 [pFA6:cyc1p-PCP-EGFP(2X)::HIS3]) This study
Ede1-aGFP + p416-Adc17-24xPP7SL-Stop-24xMS2SL + PCP-GFP(2x) + MCP-mCherry (MATa his3Δ1 leu2Δ0 met15Δ0 ura3Δ0 EDE1-aGFP:LEU2 [p416-Adc17-24xPP7SL-Stop-24xMS2SL + pFA6:cyc1p-PCP-EGFP(2X)-cyc1p-MCP-mCherry::HIS3]) This study
adc17Δ + p416-Adc17-24xPP7SL-Stop-24xMS2SL + PCP-GFP(2x) + MCP-mCherry (MATa his3Δ1 leu2Δ0 met15Δ0 ura3Δ0 ede1::kanMx [p416-Adc17-24xPP7SL-Stop-24xMS2SL + pFA6:cyc1p-PCP-EGFP(2X)-cyc1p-MCP-mCherry::HIS3]) This study
adc17Δ Ede1-aGFP + p416-Adc17-24xPP7SL-Stop-24xMS2SL + PCP-GFP(2x) + MCP-mCherry (MATa his3Δ1 leu2Δ0 met15Δ0 ura3Δ0  ede1::kanMx EDE1-aGFP:LEU2 [p416-Adc17-24xPP7SL-Stop-24xMS2SL + pFA6:cyc1p-PCP-EGFP(2X)-cyc1p-MCP-mCherry::HIS3]) This study
Adc17-24xPP7SL Abp1-mKate2-aGFP + PCP-GFP(2x) (MATa his3Δ1 leu2Δ0 met15Δ0 ura3Δ0 ADC17-24xPP7SL-LoxP ABP1-mKate2-aGFP:LEU2 [pFA6:cyc1p-PCP-EGFP(2X)::HIS3]) This study
Adc17-24xPP7SL Abp1-mKate2 + PCP-GFP(2x) (MATa his3Δ1 leu2Δ0 met15Δ0 ura3Δ0 ADC17-24xPP7SL-LoxP ABP1-mKate2:KanMX [pFA6:cyc1p-PCP-EGFP(2X)::HIS3]) This study
Adc17-24xPP7SL Abp1-mKate2 + PCP-GFP(2x) (MATa his3Δ1 leu2Δ0 met15Δ0 ura3Δ0 ADC17-24xPP7SL-LoxP ABP1-mKate2:KanMX [pFA6:cyc1p-PCP-EGFP(2X)::HIS3]) This study
Adc17-24xPP7SL Ede1-tdimer 2 + PCP-GFP(2x) (MATa his3Δ1 leu2Δ0 met15Δ0 ura3Δ0 ADC17-24xPP7SL-LoxP EDE1-tdimer2:KanMX [pFA6:cyc1p-PCP-EGFP(2X)::HIS3]) This study
Adc17-24xPP7SL Sla1-mKate2 + PCP-GFP(2x) (MATa his3Δ1 leu2Δ0 met15Δ0 ura3Δ0 ADC17-24xPP7SL-LoxP SLA1-mKate2:KanMX [pFA6:cyc1p-PCP-EGFP(2X)::HIS3]) This study
Adc17-24xPP7SL Vrp1-mKate2 + PCP-GFP(2x) (MATa his3Δ1 leu2Δ0 met15Δ0 ura3Δ0 ADC17-24xPP7SL-LoxP VRP1-mKate2:KanMX [pFA6:cyc1p-PCP-EGFP(2X)::HIS3]) This study
Adc17-24xPP7SL Abp1-mKate2-aGFP + PCP-GFP(2x) ede1Δ (MATa his3Δ1 leu2Δ0 met15Δ0 ura3Δ0 ADC17-24xPP7SL-LoxP ede1Δ::LEU2 ABP1-mKate2:KanMX [pFA6:cyc1p-PCP-EGFP(2X)::HIS3]) This study
Adc17-70ntΔ (CRISPR/CAS9) (MATa his3Δ1 leu2Δ0 met15Δ0 ura3Δ0 5'UTR-70ntΔ-ADC17) This study
WT + FGH17-70ntd + Kozak (MATa his3Δ1 leu2Δ0 met15Δ0 ura3Δ0 ADC17-24xPP7SL-LoxP ABP1-mKate2:KanMX [pFA6:cyc1p-PCP-EGFP(2X)::HIS3]) This study
WT + FGH17-70nt only (MATa his3Δ1 leu2Δ0 met15Δ0 ura3Δ0 ADC17-24xPP7SL-LoxP ABP1-mKate2:KanMX [pFA6:cyc1p-PCP-EGFP(2X)::HIS3]) This study
BY4741 + FGH17-70ntΔ + Kozak (MATa his3Δ1 leu2Δ0 met15Δ0 ura3Δ0 [p416:FGH17-70ntΔ + Kozak::URA3]) This study
BY4741 + FGH17-70nt only (MATa his3Δ1 leu2Δ0 met15Δ0 ura3Δ0 [p416:FGH17--70nt only::URA3]) This study
act1-101 (MATa his3Δ1 leu2Δ0 met15Δ0 ura3Δ0 act1-101:kanMX) Euroscarf
Rpl10-GFP + FGH17 (MATa leu2Δ0 met15Δ0 ura3Δ0 RPL10-GFP:His3MX6 [p416:FGH17::URA3]) This study
Rpl10-GFP + FGH17-70ntΔ (MATa leu2Δ0 met15Δ0 ura3Δ0 RPL10-GFP:His3MX6 [p416:FGH17-70ntΔ::URA3]) This study

Authentication | Authentication of cell lines that were not generated in this study, was done by PCR and on the basis of the expected phenotype.

Mycoplasma contamination | Testing for Mycoplasma contamination was not needed, as only yeast has been used in this study

Commonly misidentified lines (See ICLAC register) | No commonly misidentified cell lines were used

