## [Peer Review File · Nature Cell Biology]

Peer Review Information

Journal: Nature Cell Biology

Manuscript Title: Actin remodelling controls proteasome homeostasis upon stress

Corresponding author name(s): Adrien Rousseau

Reviewer Comments & Decisions:

Decision Letter, initial version:

Dear Dr Rousseau,

I hope you are well. After discussion within the editorial team, I am delighted to handle the peer review of your manuscript.

Your manuscript, "Actin remodelling controls proteasome homeostasis upon stress", has now been seen by 3 referees, who are experts in yeast proteasome/UPS (referee 1); translation and mTOR (referee 2); and actin and endocytosis (referee 3). As you will see from their comments (attached below) they find this work of potential interest, but have raised substantial concerns, which in our view would need to be addressed with considerable revisions before we can consider publication in Nature Cell Biology.

Nature Cell Biology editors discuss the referee reports in detail within the editorial team, including the chief editor, to identify key referee points that should be addressed with priority, and requests that are overruled as being beyond the scope of the current study. To guide the scope of the revisions, I have listed these points below. We are committed to providing a fair and constructive peer-review process, so please feel free to contact me if you would like to discuss any of the referee comments further.

In particular, it would be essential to:

A) Assess whether ADC17 5'UTR is necessary and sufficient for translation induction by rapamycin, and validate potential effects on endogenous mRNAs (Reviewers #1 and #2)

B) Precisely characterize which steps of translation may be affected and whether or not mRNA stability is also affected (Reviewer #2)

C) More completely characterize the role of F-actin and the effect of F-actin perturbations (Reviewer

#3)

D) All other referee concerns pertaining to strengthening existing data, providing controls, methodological details, clarifications and textual changes should also be addressed.

E) Finally please pay close attention to our guidelines on statistical and methodological reporting (listed below) as failure to do so may delay the reconsideration of the revised manuscript. In particular please provide:

- a Supplementary Figure including unprocessed images of all gels/blots in the form of a multi-page pdf file. Please ensure that blots/gels are labeled and the sections presented in the figures are clearly indicated.
- a Supplementary Table including all numerical source data in Excel format, with data for different figures provided as different sheets within a single Excel file. The file should include source data giving rise to graphical representations and statistical descriptions in the paper and for all instances where the figures present representative experiments of multiple independent repeats, the source data of all repeats should be provided.

We would be happy to consider a revised manuscript that would satisfactorily address these points, unless a similar paper is published elsewhere, or is accepted for publication in Nature Cell Biology in the meantime.

- ensure that it conforms to our format instructions and publication policies (see below and <https://www.nature.com/nature/for-authors>).
- provide a point-by-point rebuttal to the full referee reports verbatim, as provided at the end of this letter.
- provide the completed Reporting Summary (found here <https://www.nature.com/documents/nr-reporting-summary.pdf>). This is essential for reconsideration of the manuscript will be available to editors and referees in the event of peer review. For more information see <http://www.nature.com/authors/policies/availability.html> or contact me.

When submitting the revised version of your manuscript, please pay close attention to our [href="https://www.nature.com/nature-research/editorial-policies/image-integrity">Digital Image Integrity Guidelines](https://www.nature.com/nature-research/editorial-policies/image-integrity). and to the following points below:

- that unprocessed scans are clearly labelled and match the gels and western blots presented in figures.
- that control panels for gels and western blots are appropriately described as loading on sample processing controls

2-- all images in the paper are checked for duplication of panels and for splicing of gel lanes.

Nature Cell Biology is committed to improving transparency in authorship. As part of our efforts in this direction, we are now requesting that all authors identified as 'corresponding author' on published papers create and link their Open Researcher and Contributor Identifier (ORCID) with their account on the Manuscript Tracking System (MTS), prior to acceptance. ORCID helps the scientific community achieve unambiguous attribution of all scholarly contributions. You can create and link your ORCID from the home page of the MTS by clicking on 'Modify my Springer Nature account'. For more information please visit www.springernature.com/orcid.

This journal strongly supports public availability of data. Please place the data used in your paper into a public data repository, or alternatively, present the data as Supplementary Information. If data can only be shared on request, please explain why in your Data Availability Statement, and also in the correspondence with your editor. Please note that for some data types, deposition in a public repository is mandatory - more information on our data deposition policies and available repositories appears below.

[REDACTED]

We would like to receive a revised submission within six months.

We hope that you will find our referees' comments, and editorial guidance helpful. Please do not hesitate to contact me if there is anything you would like to discuss.

Best wishes,

Daryl Jason David

Daryl J.V. David, PhD

Senior Editor, Nature Cell Biology

3Consulting Editor, Nature Communications
Nature Portfolio

Heidelberger Platz 3, 14197 Berlin, Germany
Email: daryl.david@nature.com
ORCID: <https://orcid.org/0000-0002-9253-4805>

Reviewers' Comments:

Reviewer #1:

Remarks to the Author:

This is a very interesting study using state-of-the-art genetic, proteomic and cell biology methods to get at the question of how selective translation of certain mRNAs under stress conditions occurs in budding yeast. The specific mRNAs are those for proteasome regulatory particle assembly chaperones (RPACs), which the senior author had previously shown, while a postdoc, were strongly induced by TORC1 inhibition with rapamycin. The mRNA for the Adc17 RPAC was used to make a reporter for selective translation; Williams et al. examined nascent ribosomes with a clever mass spectrometry method for proteins that specifically associated with this mRNA but not one with a 5'-UTR deletion that eliminated the translational response to rapamycin. Edc1, an endocytosis factor, was identified. Rapamycin strongly induces accumulation of full proteasomes, and this induction is blocked in edc1D cells. Rapamycin also induced active translation of ADC17, which is lost in edc1D.

Given the known link of Edc1 and endocytosis with the actin network, the authors went on to show that mRNPs with the ADC17 mRNA move along actin cables and redistribute, in an Edc1-dependent fashion, to cortical actin patches when rapamycin induces transient loss of cables (depolarisation; this term should be precisely defined when first used in the text). Directly depolymerizing cables with latrunculin-B also induces ADC17 mRNA relocalization to actin patches and activates RPAC translation and proteasome assembly; the relocalization to actin patches also required Edc1. Artificially tethering ADC17 mRNA to cortical actin patches via binding to a construct containing the patch component Abp1 (or Edc1) enhanced Adc17 levels upon rapamycin treatment.

These data make a convincing story for the importance of ADC17 relocalization, and presumably that of other RPAC mRNAs, to actin patches via Edc1 tethering under rapamycin stress for selectively enhancing their translation. Exactly why translation is enhanced at these sites is unknown, but these sophisticated analyses are what allow this question to be asked now. I have no qualms about the experiments done and have only a few minor suggestions to increase the clarity and impact of the paper.

1. The effect of rapamycin appears to selectively affect RPACs and RP assembly. Core particle (CP) assembly must also increase to allow full 26S proteasome (RP-CP) formation. The authors should address how this might occur in the Discussion.
2. That rapamycin is a TORC1 inhibitor should be noted on line 62 or earlier.
3. The last 40 nt of the ADC17 5'-UTR are necessary for translation induction by rapamycin. Are they

4also sufficient (or is something slightly larger)? It would be a simple experiment to see if this is the case. Identifying a minimal element would allow the RPAC mRNAs, and possibly others, to be explored for related motifs, which presumably would also bind Edc1.

4. Line 222: *ede1D* is epistatic to Lat-B treatment; 'independent of actin depolarisation' does not seem like the right way to phrase.

5. For Fig. 7h,i: Prediction is that this tethering to the patch via Abp1 will bypass the requirement for Ede1. Can delete EDC1 from these cells and test this.

6. The Discussion is pretty thin. In addition to point 1 above, it would be nice to see some hypotheses for why mRNA localization to actin patches might enhance translation. Maybe I missed this.

Reviewer #2:

Remarks to the Author:

In this article, Williams et al, provide evidence suggesting that in yeast, actin cytoskeleton remodeling plays a central role in TORC1 inhibition-mediated translational upregulation of RPAC. To this end, authors present data that point out towards the model whereby Ede1 stimulates stress-dependent enrichment of ADC17 mRNA at actin patches. In support of this model, genetic evidence is provided that Ede1 abrogation blocks induction of proteasome assembly under conditions wherein TORC1 is inhibited. Finally, authors show that they can bolster ADC17 induction by rapamycin by tethering it to actin patches. Overall, it was found that this article is of broad interest inasmuch as it provides further insights into the mechanisms linking TOR and proteasome function while also imparting evidence for the role of actin patches in serving as platforms for selective translation under stress. Notwithstanding my overall enthusiasm for this study, I thought that several issues should be addressed to adequately corroborate authors' conclusions. These are outlined below:

Major points:

1. Notwithstanding that using an unrelated RPAC (*Nas6*) was appreciated, it was thought that it would be advantageous to compare behaviour of endogenous mRNAs to that of the reporter. Also, direct monitoring of distribution of endogenous vs. reporter mRNAs on the polysomes as a function of rapamycin treatment seems to be warranted (as the lack of correlations between mRNA and protein levels with a variety of factors including protein stability). Finally, considering that reporter expression is lost by 5'UTR removal/partial deletions at baseline, the experiments using reporters devoid of 5'UTR and 3'UTR should be accompanied by those exploiting the potential effects of these interventions on the integrity of corresponding transcripts and/or Kozak consensus sequence. Along these lines, although the use of the reporters to dissect the potential underlying mechanisms was highly appreciated, it was thought that the validation experiments with endogenous mRNAs (e.g. showing that the experimental perturbation affect distribution of endogenous mRNAs on polysomes, enrichment of endogenous mRNAs on actin patches etc) were thought to be warranted.

2. The authors should perhaps consider that one of the limitations of their approach is that albeit cycloheximide will block elongation, it does not block initiation (Warner 1966, JMB) nor RNP assembly. With this being said, subsequent validation and functional characterization of Ede1 largely alleviated my concerns pertinent to the methodology of the initial screen. Nonetheless, perhaps this can explain

5lack of the effects of deletion of Cup1 and rpS6a and 18b in subsequent functional screen.

3. "As only 3 mutants of CME proteins had defects in rapamycin-induced proteasome assembly, their role in RPAC induction is likely independent of endocytosis." I would argue that insufficient data are provided to discard the potential link between RPAC induction and endocytosis in the context of TOC1 inhibition. To this end, the authors should either consider removing this statement or performing additional experiments to corroborate it.

4. Although the experiments with mRNA tethering were found to be informative, they do not directly demonstrate that tethering of ADC17 mRNA increases its translational efficiency inasmuch as the only evidence that was provided here were increased ADC17 protein levels. It should be perhaps considered that the tethering may affect factors that may indirectly impact on alterations in protein levels (e.g. mRNA stability). I am therefore of the opinion that the authors should thus at least provide the evidence that the reporter mRNA levels are unaltered in these experiments.

Minor concerns:

1. It was somewhat surprising that nascent polypeptides were not released from polysomes after cycloheximide treatment (e.g. Hobson et al eLIFE 2020). The authors should comment on this.

2. In the introduction, it is not clear to me what the "unwanted" protein would mean.

I hope that the authors will find my assessment of their work constructive and of sufficient pathos.

Sincerely

I/Topisirovic

Reviewer #3:

Remarks to the Author:

Efficient proteostasis in cells demands efficient clearing of damaged or misfolded proteins, and an important pathway involved in such clearance is the ubiquitin-proteasome pathway. Degradation of a protein via the ubiquitin pathway proceeds involves: (i) ubiquitination of target substrates and (ii) degradation of the targeted protein by the 26S proteasome complex. The proteasome complex consists of a core complex and a regulatory particle assembly complex (RPAC).

This manuscript elucidates an important aspect of how the proteasome system is upregulated/assembled during stress for timely and efficient clearing of misfolded/unwanted proteins. The authors demonstrate through an unbiased approach that Ede1 (an important regulator of endocytosis) also serves as a regulator of RPAC translation by binding to the 5'UTR of Adc17 (a subunit of RPAC) following inhibition of TORC1 complex. The authors further demonstrate that two Ede1-associated actin regulators, Sla1 and Vrp1, are also involved in the regulation of RPAC

6translation under stress. The authors finally demonstrate the actin filament remodeling either after rapamycin treatment or following treatment with latrunculin B, an actin monomer sequestering small molecule, is an important step to induce RPAC translation through Ede1.

Overall, the work is highly impactful, demonstrating the participation of novel proteins, including the actin cytoskeleton, in proteostasis. Most of the conclusions drawn in the manuscript are supported by the experimental data. My two major criticisms for the work however are the following, which are detailed in the next section:

A: The data involving actin filaments are often confusing and as such the role of actin filaments is not clear.

B: Some of the data need to be adequately quantified

Major points:

The precise role of actin filaments, and the experiments with Lat B, are confusing. Lat B is a small molecule that prevents actin polymerization by binding actin monomers, and as such should dramatically decrease both actin patches and cables. The authors' data show, if anything, an increase in actin patch-like structures using fluorescent phalloidin. Phalloidin specifically binds to actin filaments (not actin monomers), and it is confusing as to why the authors still see extensive positive staining with phalloidin following Lat B treatment. Questions:

1) Is the LatB treatment similar to that used by others in budding yeast, and how do the results compare to these previous works?

2) Can the authors cause complete actin depolymerization (for example, with longer treatment or higher doses of LatB, or perhaps with LatA, which we have found to be more effective in other cell types), and what effects are seen?

3) Would it be possible to use mutant strains that have specific defects in patch or cable actin (I believe these exist) and show the effects.

A possible alternative hypothesis might be that the actin cables sequester most of the Adc17 mRNA, preventing its translation and/or interaction with Ede1. Depolymerization of actin following rapamycin treatment, frees up such sequestration and leads to its interaction with Ede1 and translation.

In addition, it would be good to discuss the precise role of Ede1, Sla1 and Vrp1 in translation of Adc17 mRNA. Do the authors find the latter two in addition to Ede1 as potential binders from their experiment in Fig1e?

In terms of more sufficient quantification, these are the figures in question:

1. The authors in Fig 3a-b show a partial overlap of Ede1 and Adc17 mRNA, however the data presented consists of one image. They should quantify the extent of this overlap from different image sets and/or puncta combinations and show a: how many of these puncta overlap/co-localize at any given time, and b: The dwell time of such co-localization. Further, what would be interesting to see is which of these parameters is significantly altered between control and stressed condition.

It is also important to have a negative control (rps6a, rps18b or cup1) and show that these puncta are not associated with Adc17 mRNA, or their association is not altered upon stress.

2. In Fig 3c-e, the authors elegantly show that Ede1 controls active translation of Adc17 mRNA. It would be fantastic if the authors can locate Ede1 puncta on these actively translating Adc17 mRNA in

a subset of images.

4. In fig 5b, c, the authors need to clearly define their criteria for judging actin cables and patches to locate Adc17 mRNA. What is the relative abundance of cables to patches? If this is drastically different, the authors need to normalize the data in 4c to the relative abundance. Fig 5d needs to be quantified from a few images.

5. Fig 5e-h must be clarified. It is confusing that the authors see almost an 80% reduction in cells with actin cables while there is only a minor decrease in the abundance of Adc17 mRNA on cables. How do the authors decide on the identity of cables and patches following rapamycin treatment? Do all the phalloidin positive spots after 1H of rapamycin treatment represent actin patches?

Minor comments:

1. It would be good to provide marks for molecular weight markers on the western blots.
2. It will be good to provide a quantification of the different western blots performed.
3. Insets of Fig 3b, 3d need to have scale bars
4. Actin "depolymerization" have been mis-spelt as "depolarization" throughout the text.
5. The barbed ends of the actin filaments in Fig 5a should point towards the membrane. See models from David Drubin's group.

8AUTHOR AFFILIATIONS – should be denoted with numerical superscripts (not symbols) preceding the names. Full addresses should be included, with US states in full and providing zip/post codes. The corresponding author is denoted by: "Correspondence should be addressed to [initials]."

Methods should be written concisely, but should contain all elements necessary to allow interpretation and replication of the results. As a guideline, Methods sections typically do not exceed 3,000 words. The Methods should be divided into subsections listing reagents and techniques. When citing previous methods, accurate references should be provided and any alterations should be noted. Information

must be provided about: antibody dilutions, company names, catalogue numbers and clone numbers for monoclonal antibodies; sequences of RNAi and cDNA probes/primers or company names and catalogue numbers if reagents are commercial; cell line names, sources and information on cell line identity and authentication. Animal studies and experiments involving human subjects must be reported in detail, identifying the committees approving the protocols. For studies involving human subjects/samples, a statement must be included confirming that informed consent was obtained. Statistical analyses and information on the reproducibility of experimental results should be provided in a section titled "Statistics and Reproducibility".

All Nature Cell Biology manuscripts submitted on or after March 21 2016 must include a Data availability statement as a separate section after Methods but before references, under the heading "Data Availability". For Springer Nature policies on data availability see <http://www.nature.com/authors/policies/availability.html>; for more information on this particular policy see <http://www.nature.com/authors/policies/data/data-availability-statements-data-citations.pdf>. The Data availability statement should include:

- Accession codes for primary datasets (generated during the study under consideration and designated as "primary accessions") and secondary datasets (published datasets reanalysed during the study under consideration, designated as "referenced accessions"). For primary accessions data should be made public to coincide with publication of the manuscript. A list of data types for which submission to community-endorsed public repositories is mandated (including sequence, structure, microarray, deep sequencing data) can be found here <http://www.nature.com/authors/policies/availability.html#data>.
- Unique identifiers (accession codes, DOIs or other unique persistent identifier) and hyperlinks for datasets deposited in an approved repository, but for which data deposition is not mandated (see here for details <http://www.nature.com/sdata/data-policies/repositories>).
- At a minimum, please include a statement confirming that all relevant data are available from the authors, and/or are included with the manuscript (e.g. as source data or supplementary information), listing which data are included (e.g. by figure panels and data types) and mentioning any restrictions on availability.
- If a dataset has a Digital Object Identifier (DOI) as its unique identifier, we strongly encourage including this in the Reference list and citing the dataset in the Methods.

We recommend that you upload the step-by-step protocols used in this manuscript to the Protocol Exchange. More details can found at www.nature.com/protocolexchange/about.

FIGURES – Colour figure publication costs \$600 for the first, and \$300 for each subsequent colour

10figure. All panels of a multi-panel figure must be logically connected and arranged as they would appear in the final version. Unnecessary figures and figure panels should be avoided (e.g. data presented in small tables could be stated briefly in the text instead).

All imaging data should be accompanied by scale bars, which should be defined in the legend. Cropped images of gels/blots are acceptable, but need to be accompanied by size markers, and to retain visible background signal within the linear range (i.e. should not be saturated). The boundaries of panels with low background have to be demarked with black lines. Splicing of panels should only be considered if unavoidable, and must be clearly marked on the figure, and noted in the legend with a statement on whether the samples were obtained and processed simultaneously. Quantitative comparisons between samples on different gels/blots are discouraged; if this is unavoidable, it should only be performed for samples derived from the same experiment with gels/blots were processed in parallel, which needs to be stated in the legend.

The total number of Supplementary Figures (not including the “unprocessed scans” Supplementary Figure) should not exceed the number of main display items (figures and/or tables (see our Guide to Authors and March 2012 editorial <http://www.nature.com/ncb/authors/submit/index.html#suppinfo>;

<http://www.nature.com/ncb/journal/v14/n3/index.html#ed>). No restrictions apply to Supplementary Tables or Videos, but we advise authors to be selective in including supplemental data.

GUIDELINES FOR EXPERIMENTAL AND STATISTICAL REPORTING

REPORTING REQUIREMENTS – We are trying to improve the quality of methods and statistics reporting in our papers. To that end, we are now asking authors to complete a reporting summary that collects information on experimental design and reagents. The Reporting Summary can be found here <https://www.nature.com/documents/nr-reporting-summary.pdf>) If you would like to reference the guidance text as you complete the template, please access these flattened versions at <http://www.nature.com/authors/policies/availability.html>.

Author Rebuttal to Initial comments

Reviewers' Comments:

Reviewer #1:

Remarks to the Author:

This is a very interesting study using state-of-the-art genetic, proteomic and cell biology methods to get at the question of how selective translation of certain mRNAs under stress conditions occurs in budding yeast. The specific mRNAs are those for proteasome regulatory particle assembly chaperones (RPACs), which the senior author had previously shown, while a postdoc, were strongly induced by TORC1 inhibition with rapamycin. The mRNA for the Adc17 RPAC was used to make a reporter for selective translation; Williams et al. examined nascent ribosomes with a clever mass spectrometry method for proteins that specifically associated with this mRNA but not one with a 5'-UTR deletion that eliminated the translational response to rapamycin. Edc1, an endocytosis factor, was identified. Rapamycin strongly induces accumulation of full proteasomes, and this induction is blocked in *edc1D* cells. Rapamycin also induced active translation of ADC17, which is lost in *edc1D*.

Given the known link of Edc1 and endocytosis with the actin network, the authors went on to show that mRNPs with the ADC17 mRNA move along actin cables and redistribute, in an Edc1-dependent fashion, to cortical actin patches when rapamycin induces transient loss of cables (depolarisation; this term should be precisely defined when first used in the text).

“Depolarisation” has now been clearly defined in the main text at lines 185-192: “The distribution of cortical actin patches and cables is polarized in budding yeast with cortical actin patches being found almost exclusively in the bud, and cables being aligned longitudinally from the mother cell into the bud²⁶. It has been reported that rapamycin depolarises the actin cytoskeleton¹⁶, and we have shown that ADC17 mRNA is largely localised to actin structures (Fig. 5b-d). It is possible, therefore, that actin depolarisation is a key step in RPAC induction. We first monitored the kinetics of actin depolarisation after rapamycin treatment. Budding cells containing >6 cortical actin patches in the larger mother cell were considered to have a depolarized actin cytoskeleton, as previously described^{28,29}.”

14Directly depolymerizing cables with latrunculin-B also induces ADC17 mRNA relocalization to actin patches and activates RPAC translation and proteasome assembly; the relocalization to actin patches also required Edc1. Artificially tethering ADC17 mRNA to cortical actin patches via binding to a construct containing the patch component Abp1 (or Edc1) enhanced Adc17 levels upon rapamycin treatment.

These data make a convincing story for the importance of ADC17 relocalization, and presumably that of other RPAC mRNAs, to actin patches via Edc1 tethering under rapamycin stress for selectively enhancing their translation. Exactly why translation is enhanced at these sites is unknown, but these sophisticated analyses are what allow this question to be asked now. I have no qualms about the experiments done and have only a few minor suggestions to increase the clarity and impact of the paper.

We thank the reviewer for his/her assessment of our manuscript and for providing beneficial comments. We direct him/her to our responses to each individual question below.

1. The effect of rapamycin appears to selectively affect RPACs and RP assembly. Core particle (CP) assembly must also increase to allow full 26S proteasome (RP-CP) formation. The authors should address how this might occur in the Discussion.

This is an interesting point raised by the reviewer and CP assembly is also increased upon rapamycin treatment to allow the formation of functional 26S proteasome (Rousseau et al, Nature, 2016). This is the reason why in *ede1*Δ cells, as previously observed for *mpk1*Δ cells, the level of 26S proteasome is not changing while the level of free CP is still increasing upon rapamycin (Fig.2c), a hallmark of RP assembly defect. The mechanism is currently unknown, but we've previously shown that two CP assembly chaperones, Pba1 and Pba2, are increased following rapamycin treatment, giving a potential clue for the underlying mechanism (Rousseau et al, Nature, 2016). This has been added at lines 49-51: "CP assembly is also increased upon TORC1 inhibition with two CP assembly chaperones, Pba1 and Pba2, being induced. Unlike RPAC induction, CP assembly chaperone induction is Mpk1-independent, the underlying mechanism being unknown¹³."

2. That rapamycin is a TORC1 inhibitor should be noted on line 62 or earlier.

This has been added line 63-65: “FGH17-containing cells had low basal levels of FGH17, which strongly increased following rapamycin (TORC1 inhibitor) treatment, as for the endogenous RPAC Nas6 (Fig. 1a).”.

3. The last 40 nt of the ADC17 5'-UTR are necessary for translation induction by rapamycin. Are they also sufficient (or is something slightly larger)? It would be a simple experiment to see if this is the case. Identifying a minimal element would allow the RPAC mRNAs, and possibly others, to be explored for related motifs, which presumably would also bind Edc1.

We found typos on the FGH17 deletion mutants (40nt Δ -> 70nt Δ and 30nt Δ -> 40nt Δ) which have been corrected in the manuscript (Fig. 1). We've performed the suggested experiment and we observed that this sequence is not sufficient for translation induction by rapamycin. This has been added to the main text at lines 77-78: “The 70-nucleotide region alone was not sufficient for FGH17 reporter expression (Extended data Fig. 1b).”.

In a follow-up study, we are trying to identify the minimal sequence required for Adc17 induction to use it as bait to identify more translation regulators.

4. Line 222: *ede1D* is epistatic to Lat-B treatment; 'independent of actin depolarisation' does not seem like the right way to phrase.

The text has been modified accordingly at lines 218-219: “This result shows that Ede1 plays a role downstream of actin depolarisation, possibly by stabilising Adc17 mRNAs at cortical actin patches.”.

5. For Fig. 7h,i: Prediction is that this tethering to the patch via Abp1 will bypass the requirement for Ede1. Can delete EDC1 from these cells and test this.

This is indeed a very interesting point raised by the reviewer. We have performed this experiment and we observed that Adc17 induction was significantly restored by targeting it to actin patches (Abp1-aGFP) in *ede1 Δ* cells (Fig. 7j, k). This has been added to the main text at lines 246-249: “When ADC17 mRNA was targeted to cortical actin patches in *ede1 Δ* cells, Adc17 induction upon rapamycin was restored to

WT levels, indicating that an important function of Ede1 is to recruit ADC17 mRNA to cortical actin patches (Fig. 7j, k).”.

6. The Discussion is pretty thin. In addition to point 1 above, it would be nice to see some hypotheses for why mRNA localization to actin patches might enhance translation. Maybe I missed this.

We now provide such hypotheses/models in the Discussion:

- on lines 265-270: “The most well-characterised of these is the ASH1 mRNA, which is transported along the actin cytoskeleton in a translationally repressed state to the bud tip, where it is anchored, and translation activated. It is possible a similar mechanism is responsible for RPAC induction. In this scenario, RPAC mRNA is transported along actin cables in a translationally repressed state. When actin cables are disrupted by stress, ADC17 mRNA relocates to Ede1 sites and translation inhibition is relieved.”.
- On lines 279-294: “As cortical actin patches have a higher density of F-actin and are less sensitive to stress than actin cables, they may serve directly as a translation platform or indirectly by recruiting mRNA to a translationally active cellular compartment, helping to translate stress-induced proteins such as RPACs. In agreement with this possibility, it has been recently reported that Ede1 foci are surrounded by fenestrated ER containing membrane-associated ribosomes⁴⁹. Recent findings showed that the ER has a far more diverse role in mRNA translation than expected, with ER-bound ribosomes functioning in the synthesis of both cytosolic and ER-targeted proteins⁴². These observations have been reported in diverse organisms and using different methodologies, suggesting that the ER is a favourable environment for translation. Furthermore, ER-localized mRNA translation is less inhibited by stress than their cytosolic counterparts, suggesting the ER represents a protective environment for translation upon stress, allowing the cell to synthesise a specific set of proteins under these conditions⁴². This would support a model in which the role of Ede1 in RPACs translation upon stress may be to recruit their mRNAs to cortical actin patches, so they are near ER-associated ribosomes for translation. As the ER has been reported to be an important site for 20S proteasome assembly, it would be possible to imagine that proteasome component are co-translationally assembled at the surface of ER membrane⁵⁰.”.

Reviewer #2:

Remarks to the Author:

17In this article, Williams et al, provide evidence suggesting that in yeast, actin cytoskeleton remodeling plays a central role in TORC1 inhibition-mediated translational upregulation of RPAC. To this end, authors present data that point out towards the model whereby Ede1 stimulates stress-dependent enrichment of ADC17 mRNA at actin patches. In support of this model, genetic evidence is provided that Ede1 abrogation blocks induction of proteasome assembly under conditions wherein TORC1 is inhibited. Finally, authors show that they can bolster ADC17 induction by rapamycin by tethering it to actin patches. Overall, it was found that this article is of broad interest inasmuch as it provides further insights into the mechanisms linking TOR and proteasome function while also imparting evidence for the role of actin patches in serving as platforms for selective translation under stress. Notwithstanding my overall enthusiasm for this study, I thought that several issues should be addressed to adequately corroborate authors' conclusions. These are outlined below:

We thank the reviewer for his/her in-depth assessment of our manuscript and for providing constructive comments and suggestions. We addressed each of them in the revised manuscript. We direct the reviewer to our responses to each individual question below.

Major points:

1. Notwithstanding that using an unrelated RPAC (Nas6) was appreciated, it was thought that it would be advantageous to compare behaviour of endogenous mRNAs to that of the reporter. Also, direct monitoring of distribution of endogenous vs. reporter mRNAs on the polysomes as a function of rapamycin treatment seems to be warranted (as the lack of correlations between mRNA and protein levels with a variety of factors including protein stability).

We agree with the reviewer that comparing the behaviour of endogenous ADC17 mRNA to that of the FGH17 reporter would be a good addition to the paper. We've therefore performed RiboTag on cells expressing FGH17 to monitor the recruitment of its mRNA, along with that of ADC17, to ribosomes for translation. We observed that recruitment of both ADC17 mRNA and FGH17 mRNA to ribosomes is increased upon rapamycin treatment (Fig. 1b). This confirmed that both mRNAs are behaving similarly, thereby confirming that FGH17 is a good reporter to interrogate how translation of mRNAs such as ADC17 are regulated. This, in association to the Suntag experiment (Fig.3f), also confirms that ADC17 is

translationally regulated upon rapamycin treatment. This has been added to the manuscript at lines 65-69: “We additionally compared the behaviour of endogenous ADC17 mRNA to FGH17 to confirm that they share similar translation regulation. Using RiboTag¹⁸, we observed that rapamycin increased the recruitment of both ADC17 mRNA and FGH17 mRNA to ribosomes for translation (Fig. 1b). This confirmed that FGH17 is a good reporter to interrogate how translation of mRNAs such as Adc17 are regulated upon stress.”.

Moreover, we have previously shown via a cycloheximide time-course that RPAC proteins are stable over the 4h period of rapamycin treatment used in this manuscript (Rousseau et al, Nature, 2016), thereby excluding that changes in protein stability explain the induction of RPACs upon rapamycin treatment.

Finally, considering that reporter expression is lost by 5'UTR removal/partial deletions at baseline, the experiments using reporters devoid of 5'UTR and 3'UTR should be accompanied by those exploiting the potential effects of these interventions on the integrity of corresponding transcripts and/or Kozak consensus sequence.

We found typos on the FGH17 deletion mutants (40nt Δ -> 70nt Δ and 30nt Δ -> 40nt Δ) which have been corrected in the manuscript (Fig. 1). Comparing the behaviour of the FGH17 reporter with that of FGH17-70nt Δ in RiboTag cells, we observed that the deletion of this 70nt sequence prevented the recruitment of FGH17-70nt Δ mRNA to ribosomes upon rapamycin treatment (Fig. 1b, e). This has been added to the manuscript at lines 78-81: “Comparing FGH17 with FGH17-70nt Δ by RiboTag, we observed that the deletion of this 70nt sequence prevented the recruitment of FGH17-70nt Δ mRNA to ribosomes upon rapamycin treatment (Fig. 1b, e) and decreased its stability by about 2-fold (Extended data Fig.1c).”.

RNA stability has also been analysed by RT-qPCR, and we now show that the level of FGH17-70nt Δ is ~ 2-fold lower than that of WT FGH17 (Extended data Fig.1c). This indicates that the deletion of these 70nt modestly destabilised FGH17 mRNA, as well as blocking ribosome recruitment. This is now discussed on lines 78-81: “Comparing FGH17 with FGH17-70nt Δ by RiboTag, we observed that the deletion of this 70nt sequence prevented the recruitment of FGH17-70nt Δ mRNA to ribosomes upon rapamycin treatment (Fig. 1b, e) and decreased its stability by about 2-fold (Extended data Fig.1c).”.

Regarding the Kozak sequence, we've reintroduced the Kozak sequence of *Adc17* in *FGH17-70ntΔ* reporter and we observed that it was not sufficient to restore *FGH17* expression. This has been added on lines 74-77: "while deletion of the 70 nucleotides upstream of the start codon (*FGH17-70ntΔ*) prevented translation (Fig. 1d). This was not due to alteration of the Kozak sequence, as reintroducing *ADC17* Kozak sequence to *FGH17-70ntΔ* (*FGH17-70ntΔ+Kozak*) was not enough to restore *FGH17* expression (Extended data Fig. 1a).".

Along these lines, although the use of the reporters to dissect the potential underlying mechanisms was highly appreciated, it was thought that the validation experiments with endogenous mRNAs (e.g. showing that the experimental perturbation affect distribution of endogenous mRNAs on polysomes, enrichment of endogenous mRNAs on actin patches etc) were thought to be warranted.

Regarding the validation of endogenous *ADC17* mRNA recruitment to ribosome, it has now been added to the new version of the manuscript, as explained above (Fig. 1b, e). Regarding the enrichment of endogenous *ADC17* mRNA to actin patches, this was done in our manuscript. Indeed, the monitoring of *ADC17* mRNA localisation relative to actin structures was performed by endogenously tagging *ADC17* mRNA with 24xPP7SL. We now specify it more clearly on lines 171-172 "To test this, we fixed cells expressing endogenous PCP-GFP-labelled *ADC17* mRNA and stained them with phalloidin to visualise actin.".

Being inspired by the reviewer's comment on analysing more endogenous mRNAs, we've used CRISPR/Cas9 to delete the 70nt upstream of the start codon at endogenous *ADC17* locus and we observed that, like *FGH17-70ntΔ*, the deletion of this region abrogated the expression of endogenous *Adc17* (Fig. 1f). This indicates that observations made with the *FGH17* reporter are physiologically relevant; this has been added to the main text on lines 81-84: "Deleting this 70nt region at the endogenous *ADC17* locus with CRISPR/CAS9, we similarly observed abrogation of *ADC17* expression (Fig. 1f). These findings indicated that the *FGH17* reporter reflects the regulation of the endogenous *ADC17* gene.".

2. The authors should perhaps consider that one of the limitations of their approach is that albeit cycloheximide will block elongation, it does not block initiation (Warner 1966, JMB) nor RNP assembly.

20With this being said, subsequent validation and functional characterization of Ede1 largely alleviated my concerns pertinent to the methodology of the initial screen. Nonetheless, perhaps this can explain lack of the effects of deletion of Cup1 and rpS6a and 18b in subsequent functional screen.

We understand the reviewer's concern here, but we think that it doesn't apply here as in our setup we only use cycloheximide to lock ribosomes on mRNAs and therefore stabilise polysomes for further immunoprecipitation (the drug is added to cells and cells are directly shifted to 4°C, as used for ribosome immunoprecipitation; Panasenko et al, bio-protocol, 2012). This has been now clarified on lines 88-89: "Polysomes were stabilised by adding cycloheximide to the cells (Fig. 1g, step 2) and UV-crosslinking covalently linked the RNA and any bound proteins together (Fig. 1g, step 3).".

3. "As only 3 mutants of CME proteins had defects in rapamycin-induced proteasome assembly, their role in RPAC induction is likely independent of endocytosis." I would argue that insufficient data are provided to discard the potential link between RPAC induction and endocytosis in the context of TOC1 inhibition. To this end, the authors should either consider removing this statement or performing additional experiments to corroborate it.

We agree with the reviewer, so we have now removed this statement.

4. Although the experiments with mRNA tethering were found to be informative, they do not directly demonstrate that tethering of ADC17 mRNA increases its translational efficiency inasmuch as the only evidence that was provided here were increased ADC17 protein levels. It should be perhaps considered that the tethering may affect factors that may indirectly impact on alterations in protein levels (e.g. mRNA stability). I am therefore of the opinion that the authors should thus at least provide the evidence that the reporter mRNA levels are unaltered in these experiments.

We completely agree with the reviewer's point, we have therefore compared the number of tethered with that of untethered mRNA per cell using the doubly tagged ADC17 mRNA (PP7 + MS2). We didn't see a difference in the number of ADC17 mRNAs in Ede1-aGFP (tethered) cells compared to that of Ede1 (untethered) cells (Extended data Fig.5e). This indicate that ADC17 mRNA levels are unaltered by artificially targeting them to Ede1 sites. This has been added to the main text on lines 235-236: "Moreover, the tethering of ADC17 mRNA to Ede1 sites had no impact on its stability (Extended data Fig. 5e).".

Minor concerns:

1. It was somewhat surprising that nascent polypeptides were not released from polysomes after cycloheximide treatment (e.g. Hobson et al eLIFE 2020). The authors should comment on this.

We think this is specific to puromycin-treated cells (Hobson et al eLIFE 2020; they show that cycloheximide induced the release of puromycylated nascent polypeptide chains from ribosomes). Puromycin releases nascent proteins from the ribosome by nucleophilic attack of the free amine on the ester linking the nascent protein to the tRNA. In our case we don't use puromycin, so there is no nucleophilic attack and therefore cycloheximide will not remove the nascent chain, a process that requires the Ribosome Quality Control (RQC) machinery. Moreover, straight after cycloheximide addition, the yeast sample were place on ice for 10 min to stop cellular reaction and UV-crosslinking was performed directly after this incubation time.

2. In the introduction, it is not clear to me what the “unwanted” protein would mean.

This has been replaced by either “short-lived” or “misfolded”:

- On line 28 “Misfolded, damaged, and short-lived proteins are degraded”
- On line 43: “allows cells to rapidly degrade misfolded and damage proteins”.
- On lines 255-256: “while clearing misfolded and damaged proteins”

I hope that the authors will find my assessment of their work constructive and of sufficient pathos.

As stated above, we thank the reviewer for his/her in-depth assessment of our manuscript and for providing constructive comments and suggestions. This really helped improving our manuscript.

Sincerely

I/Topisirovic

22Reviewer #3:

Remarks to the Author:

Efficient proteostasis in cells demands efficient clearing of damaged or misfolded proteins, and an important pathway involved in such clearance is the ubiquitin-proteasome pathway. Degradation of a protein via the ubiquitin pathway proceeds involves: (i) ubiquitination of target substrates and (ii) degradation of the targeted protein by the 26S proteasome complex. The proteasome complex consists of a core complex and a regulatory particle assembly complex (RPAC).

This manuscript elucidates an important aspect of how the proteasome system is upregulated/assembled during stress for timely and efficient clearing of misfolded/unwanted proteins. The authors demonstrate through an unbiased approach that Ede1 (an important regulator of endocytosis) also serves as a regulator of RPAC translation by binding to the 5'UTR of Adc17 (a subunit of RPAC) following inhibition of TORC1 complex. The authors further demonstrate that two Ede1-associated actin regulators, Sla1 and Vrp1, are also involved in the regulation of RPAC translation under stress. The authors finally demonstrate the actin filament remodeling either after rapamycin treatment or following treatment with latrunculin B, an actin monomer sequestering small molecule, is an important step to induce RPAC translation through Ede1.

Overall, the work is highly impactful, demonstrating the participation of novel proteins, including the actin cytoskeleton, in proteostasis. Most of the conclusions drawn in the manuscript are supported by the experimental data. My two major criticisms for the work however are the following, which are detailed in the next section:

A: The data involving actin filaments are often confusing and as such the role of actin filaments is not clear.

B: Some of the data need to be adequately quantified

We would like to thank the reviewer for valuable and constructive comments and suggestions. We have made substantial revision to the manuscript to address them. We direct the reviewer to our responses to each individual question below.

Major points:

The precise role of actin filaments, and the experiments with Lat B, are confusing. Lat B is a small

23molecule that prevents actin polymerization by binding actin monomers, and as such should dramatically decrease both actin patches and cables. The authors' data show, if anything, an increase in actin patch-like structures using fluorescent phalloidin. Phalloidin specifically binds to actin filaments (not actin monomers), and it is confusing as to why the authors still see extensive positive staining with phalloidin following Lat B treatment.

We understand the point raised by the reviewer and we have modified the manuscript to prevent such confusion. First, regarding the decrease of actin cables but not that of actin patches, this is consistent with the literature (e.g., PMID 15616194: Fig.2A and PMID 16611742: Fig.1A) as it has been shown that either Lat-B (less potent than Lat-A) or low concentration of Lat-A can be used to specifically disrupt actin cables. This is more physiologic than higher dose of Lat-A as it resembles the phenotype observed with most actin temperature-sensitive (ts) mutants or that observed upon different stresses such as heat-shock and rapamycin treatment. To our knowledge, none of these actin ts mutants or stresses induce the loss of actin patches, while cables are very often disrupted. This is because actin cables are more sensitive to actin perturbation than actin patches. To improve clarity, we now describe why we used 25 μ M Lat-B in the main text at lines 202-206: "We next tested whether an alternative means of selectively disrupting actin cables induced RPAC translation. Cells were therefore treated with 25 mM Latrunculin B, which at this concentration only disrupts actin cables³⁰, as observed upon rapamycin treatment. Lat-B treatment completely abolished actin cables and thereby relocated ADC17 mRNA to either cortical actin patches or to an unbound state (Fig. 6a,b).".

Second, regarding the detection of actin patches by phalloidin, this is because actin patches are formed by short and branched actin filaments (not actin monomers), so phalloidin is able to stain these structures. To prevent any confusion, we now define more precisely actin cables and actin patches on lines 172-175: "Yeast has two major actin structures: actin cables which are polarized linear bundles of parallel actin filaments extending along the long axis of cells and cortical actin patches which are dense dendritic networks of branched actin filaments localised at the plasma membrane²⁶".

Questions:

1) Is the LatB treatment similar to that used by others in budding yeast, and how do the results compare to these previous works?

Yes, Lat-B treatment is similar to that used by others in budding yeast and results are comparable: complete disruption of actin cables and actin patches staining at the plasma membrane (e.g., PMID 15616194: Fig.2A and PMID 16611742: Fig.1A). Modification has been made to the main text for clarification, on lines 202-206: “We next tested whether an alternative means of selectively disrupting actin cables induced RPAC translation. Cells were therefore treated with 25 mM Latrunculin B, which at this concentration only disrupts actin cables³⁰, as observed upon rapamycin treatment. Lat-B treatment completely abolished actin cables and thereby relocated ADC17 mRNA to either cortical actin patches or to an unbound state (Fig. 6a,b).”.

2) Can the authors cause complete actin depolymerization (for example, with longer treatment or higher doses of LatB, or perhaps with LatA, which we have found to be more effective in other cell types), and what effects are seen?

We have performed the suggested experiment and only Lat-A was potent enough to inhibit actin nucleation on both actin cables and actin patches, with actin cables being disrupted first (Fig. below, panel a). We observed that the level of ADC17 mRNA recruitment to Ede1 sites following Lat-A treatment is not significantly different from that of Lat-B (Fig. below, panel b), which is in agreement with previous studies showing that cortical patches containing early coat protein such as Ede1 and End3 are not disrupted by Lat-A treatment (Kaksonen M et al., Cell, 2005 and Gagny B et al., JCS, 2000). This is because actin is mainly involved in cortical patch movement and internalisation but not formation. Analysing RPAC induction, we observed that Lat-A was as potent as Lat-B to induce Adc17 which is consistent with similar Ede1 recruitment observed upon both treatments (Fig. below, panel c).

As Lat-A and Lat-B have similar effects on RPAC translation and Lat-B has a more physiological effect than Lat-A (to our knowledge, no other conditions or mutations cause the loss of actin nucleation on cortical patches while actin cable disruption is often seen in temperature-sensitive mutants and upon different stresses), we think it is better to only include Lat-B data in our manuscript for better clarity. Moreover, we now explain why we are using Lat-B in our study.

a, Representative microscopy images (maximum intensity Z-projection) of yeast cells \pm 25 μ M Latrunculin-B (Lat-B) or 12.5 μ M Latrunculin-A (Lat-A) for 3 h and stained with Rhodamine phalloidin to visualise actin (red). **b**, Frequency of ADC17 mRNA localising at Ede1 site in cells treated with either 25 μ M Latrunculin-B (Lat-B) or 12.5 μ M Latrunculin-A (Lat-A) for 3 h. For each bar, $n=4$ biologically independent experiments with at least 200 ADC17 mRNAs for each condition. Statistical analysis was carried out using unpaired t-test. ns, not significant. **c**, Western blot analysis of RPACs and Mpk1 kinase in WT cells treated \pm 25 μ M Latrunculin-B (Lat-B) or 12.5 μ M Latrunculin-A (Lat-A) for 3 h. Ponceau S staining was used as a loading control.

3) Would it be possible to use mutant strains that have specific defects in patch or cable actin (I believe these exist) and show the effects.

To our knowledge, no mutations cause the loss of actin nucleation at cortical patches while maintaining that of actin cables, while the opposite scenario is often seen. Even in Arp2/3 temperature-sensitive mutants (Arp2/3 complex mainly nucleates actin on cortical patches), actin patches are still present at the non-permissive temperature (e.g., PMID: 15657399 and PMID: 10512884). Thus, unfortunately there is currently no setup where actin nucleation on cortical patches can be selectively disrupted (Lat-A disrupts first actin cables and then actin on cortical actin patches). Nonetheless, we have obtained a temperature-sensitive mutant of actin, *act1-101*, and we now show in the manuscript that genetic disruption of the actin cables induces RPACs, as observed following Lat-B treatment. This has been

added at lines 208-209: “This was further confirmed using genetic disruption of actin cables in the temperature-sensitive mutant *act1-101* (Extended data Fig. 4a).”.

A possible alternative hypothesis might be that the actin cables sequester most of the Adc17 mRNA, preventing its translation and/or interaction with Ede1. Depolymerization of actin following rapamycin treatment, frees up such sequestration and leads to its interaction with Ede1 and translation.

We thank the reviewer for this suggestion. It could indeed be that the actin cables sequester most of the Adc17 mRNA, preventing its translation and/or interaction with Ede1. When actin cables are disrupted upon rapamycin treatment, ADC17 mRNAs are released from actin cables and subsequently recruited to Ede1 sites for translation. As suggested by the reviewer, we now discuss this model in the discussion on lines 265-270: “The most well-characterised of these is the ASH1 mRNA, which is transported along the actin cytoskeleton in a translationally repressed state to the bud tip, where it is anchored, and translation activated. It is possible a similar mechanism is responsible for RPAC induction. In this scenario, RPAC mRNA is transported along actin cables in a translationally repressed state. When actin cables are disrupted by stress, ADC17 mRNA relocates to Ede1 sites and translation inhibition is relieved.”.

In addition, it would be good to discuss the precise role of Ede1, Sla1 and Vrp1 in translation of Adc17 mRNA. Do the authors find the latter two in addition to Ede1 as potential binders from their experiment in Fig1e?

No, unfortunately, no peptides from Vrp1 and Sla1 were detected in our mass spectrometry experiment (Fig.1), even in the control cells. This is most probably due to the low abundance of these two proteins. Nonetheless, we now show that ADC17 mRNA colocalises with Ede1, Sla1 and Vrp1 sites, and this was enhanced following rapamycin treatment (Fig. 3c and Extended data Fig. 3a, b). This has been added:

- at lines 128-132: “We tracked single molecules of labelled Adc17 mRNA and observed frequent contact of Ede1 and ADC17 mRNA, demonstrating that this interaction is occurring *in-vivo* (Fig. 3b and Supplementary Video 1,2). Around 29% of Adc17 mRNAs were associated to Ede1 under basal conditions and this significantly increased to about 40% following rapamycin treatment (Fig. 3c).”.

- at lines 168-171: “As Ede1, Sla1 and Vrp1 localise to and regulate cortical actin patches at the endocytic site, and ADC17 mRNA makes contacts with Ede1, Sla1 and Vrp1 (Fig. 3c and Extended data Fig. 3a, b), it seemed likely there might be a role for actin in Adc17 mRNA regulation.”.

Moreover, we now show that targeting ADC17 mRNA back to cortical actin patches in *ede1Δ* cells, restored its induction upon rapamycin treatment. This has been added on lines 244-251: “We observed that Adc17 induction upon rapamycin treatment was increased by 1.84-fold when tethered to Abp1 (Abp1-mK-aGFP) compared to untethered mRNAs (Abp1) which is similar to that of Ede1 targeting (Fig. 7h, i). When ADC17 mRNA was targeted to cortical actin patches in *ede1Δ* cells, Adc17 induction upon rapamycin was restored to WT levels, indicating that an important function of Ede1 is to recruit ADC17 mRNA to cortical actin patches (Fig. 7j, k). Taken together, these results show that Ede1-mediated recruitment of ADC17 mRNAs at cortical actin patches following rapamycin treatment is important for stimulating Adc17 translation.”.

Why the recruitment of ADC17 mRNA to actin patches could be beneficial for its translation is also now discussed on lines 279-294: “As cortical actin patches have a higher density of F-actin and are less sensitive to stress than actin cables, they may serve directly as a translation platform or indirectly by recruiting mRNA to a translationally active cellular compartment, helping to translate stress-induced proteins such as RPACs. In agreement with this possibility, it has been recently reported that Ede1 foci are surrounded by fenestrated ER containing membrane-associated ribosomes⁴⁹. Recent findings showed that the ER has a far more diverse role in mRNA translation than expected, with ER-bound ribosomes functioning in the synthesis of both cytosolic and ER-targeted proteins⁴². These observations have been reported in diverse organisms and using different methodologies, suggesting that the ER is a favourable environment for translation. Furthermore, ER-localized mRNA translation is less inhibited by stress than their cytosolic counterparts, suggesting the ER represents a protective environment for translation upon stress, allowing the cell to synthesise a specific set of proteins under these conditions⁴². This would support a model in which the role of Ede1 in RPACs translation upon stress may be to recruit their mRNAs to cortical actin patches, so they are near ER-associated ribosomes for translation. As the ER has been reported to be an important site for 20S proteasome assembly, it would be possible to imagine that proteasome component are co-translationally assembled at the surface of ER membrane⁵⁰.”.

In terms of more sufficient quantification, these are the figures in question:

1. The authors in Fig 3a-b show a partial overlap of Ede1 and Adc17 mRNA, however the data presented

consists of one image. They should quantify the extent of this overlap from different image sets and/or puncta combinations and show a: how many of these puncta overlap/co-localize at any given time, and b: The dwell time of such co-localization. Further, what would be interesting to see is which of these parameters is significantly altered between control and stressed condition.

We agree with the reviewer, and we have now quantified the extend of Ede1-ADC17 mRNA interaction. Data showed that 28.8% of Adc17 mRNAs are associated to Ede1 at basal level and this significantly raised to 40.1% following rapamycin treatment. This has been added at lines 131-132: "Around 29% of Adc17 mRNAs were associated to Ede1 under basal conditions and this significantly increased to about 40% following rapamycin treatment (Fig. 3c).".

We attempted to quantify the dwell time, but unfortunately both the mRNA and Ede1 move in and out of the confocal plane, making it impossible to get an accurate value.

It is also important to have a negative control (rps6a, rps18b or cup1) and show that these puncta are not associated with Adc17 mRNA, or their association is not altered upon stress.

All these proteins have diffuse cytoplasmic staining, so this is unfortunately not feasible.

2. In Fig 3c-e, the authors elegantly show that Ede1 controls active translation of Adc17 mRNA. It would be fantastic if the authors can locate Ede1 puncta on these actively translating Adc17 mRNA in a subset of images.

We agree with the reviewer that it would have been nice to localise translating ADC17 mRNA to Ede1 puncta but, unfortunately, it didn't work. The first problem being that Ede1 localisation became diffuse cytoplasmic when cells were fixed, so we were restricted to live-cell imaging. The SunTag system already use GFP and mCherry, so we have generated Ede1-eCFP (knock-in at endogenous locus) cells expressing the SunTag system but suffered from strong leakage from the GFP to the eCFP channel making the analysis not possible.

4. In fig 5b, c, the authors need to clearly define their criteria for judging actin cables and patches to locate Adc17 mRNA. What is the relative abundance of cables to patches? If this is drastically different, the authors need to normalize the data in 4c to the relative abundance. Fig 5d needs to be quantified from a few images.

This is specified in the M&M but it has also been added to the figure legend for better clarity at lines 820-822: “Cortical actin patches are defined as circular punctae with a diameter of 150–750 nm. Actin cables are defined as linear structures with a length > 1 μ M.”.

The relative abundance of cables and patches in asynchronous cells is relatively similar, so we don't think that normalisation is required. Moreover, we are not aware of such normalisation being made in yeast.

Fig.5d is a representative image from n=4 biological replicates (this has been now specified in the legend at lines 822) showing that ADC17 mRNA is sliding on actin cable. The associated quantification is presented Fig. 5c (ADC17 mRNA interaction with actin cable).

5. Fig 5e-h must be clarified. It is confusing that the authors see almost an 80% reduction in cells with actin cables while there is only a minor decrease in the abundance of Adc17 mRNA on cables. How do the authors decide on the identity of cables and patches following rapamycin treatment? Do all the phalloidin positive spots after 1H of rapamycin treatment represent actin patches?

It is not a decrease of 80% of cells having cables but a decrease of 80% of cells with polarised actin network. So, depolarised cells still have actin cables even if they are shorter and less abundant. This has been clarified in the main text to prevent any confusion on lines 185-192: “The distribution of cortical actin patches and cables is polarized in budding yeast with cortical actin patches being found almost exclusively in the bud, and cables being aligned longitudinally from the mother cell into the bud²⁶. It has been reported that rapamycin depolarises the actin cytoskeleton¹⁶, and we have shown that ADC17 mRNA is largely localised to actin structures (Fig. 5b-d). It is possible, therefore, that actin depolarisation is a key step in RPAC induction. We first monitored the kinetics of actin depolarisation after rapamycin

treatment. Budding cells containing >6 cortical actin patches in the larger mother cell were considered to have a depolarized actin cytoskeleton, as previously described^{28,29}.”.

To decide on the identity of cables and patches, we used the same parameters as those described in comment 4. This been added to the figure legend at lines 820-822: “Cortical actin patches are defined as circular punctae with a diameter of 150–750 nm. Actin cables are defined as linear structures with a length > 1 μ M.”.

Yes, all phalloidin positive spots after 1H rapamycin treatment represents actin patches. In untreated cells, they accumulate in the daughter cells while in rapamycin treated cells, they are localised in both daughter and mother cells.

Minor comments:

1. It would be good to provide marks for molecular weight markers on the western blots.

This has been added to Figures 1a, 1c, 1d, 1f, 2b, 4b, 5g, 6c, 7b, 7f, 7h, 7j and Extended data figures 1a, 1b, 2b, 2c, and 4b.

2. It will be good to provide a quantification of the different western blots performed.

We’ve limited the quantification of western blots for those where changes were not black and white, as NCB policy is that quantitative comparisons between samples on different gels/blots are discouraged.

3. Insets of Fig 3b, 3d need to have scale bars

Scale bars have been added.

4. Actin “depolymerization” have been mis-spelt as “depolarization” throughout the text.

“Depolarisation” is correct, and this is now defined in the main text at lines 189-192: “We first monitored the kinetics of actin depolarisation after rapamycin treatment. Budding cells containing >6 cortical actin patches in the larger mother cell were considered to have a depolarized actin cytoskeleton, as previously described^{28,29}.”. (See above, comment 5).

5. The barbed ends of the actin filaments in Fig 5a should point towards the membrane. See models from David Drubin’s group.

This has been modified.

Methods should be written concisely, but should contain all elements necessary to allow interpretation and replication of the results. As a guideline, Methods sections typically do not exceed 3,000 words. The Methods should be divided into subsections listing reagents and techniques. When citing previous methods, accurate references should be provided and any alterations should be noted. Information must be provided about: antibody dilutions, company names, catalogue numbers and clone numbers for monoclonal antibodies; sequences of RNAi and cDNA probes/primers or company names and catalogue numbers if reagents are commercial; cell line names, sources and information on cell line identity and authentication. Animal studies and experiments involving human subjects must be reported in detail, identifying the committees approving the protocols. For studies involving human subjects/samples, a statement must be included confirming that informed consent was obtained. Statistical analyses and information on the reproducibility of experimental results should be provided in a section titled “Statistics and Reproducibility”.

All Nature Cell Biology manuscripts submitted on or after March 21 2016 must include a Data availability statement as a separate section after Methods but before references, under the heading “Data Availability”. . For Springer Nature policies on data availability see <http://www.nature.com/authors/policies/availability.html>; for more information on this particular policy see <http://www.nature.com/authors/policies/data/data-availability-statements-data-citations.pdf>. The Data availability statement should include:

- Accession codes for primary datasets (generated during the study under consideration and designated as “primary accessions”) and secondary datasets (published datasets reanalysed during the study under consideration, designated as “referenced accessions”). For primary accessions data should be made public to coincide with publication of the manuscript. A list of data types for which submission to community-endorsed public repositories is mandated (including sequence, structure, microarray, deep sequencing data) can be found here <http://www.nature.com/authors/policies/availability.html#data>.
- Unique identifiers (accession codes, DOIs or other unique persistent identifier) and hyperlinks for datasets deposited in an approved repository, but for which data deposition is not mandated (see here for details <http://www.nature.com/sdata/data-policies/repositories>).
- At a minimum, please include a statement confirming that all relevant data are available from the

authors, and/or are included with the manuscript (e.g. as source data or supplementary information), listing which data are included (e.g. by figure panels and data types) and mentioning any restrictions on availability.

- If a dataset has a Digital Object Identifier (DOI) as its unique identifier, we strongly encourage including this in the Reference list and citing the dataset in the Methods.

We recommend that you upload the step-by-step protocols used in this manuscript to the Protocol Exchange. More details can found at www.nature.com/protocolexchange/about.

All imaging data should be accompanied by scale bars, which should be defined in the legend. Cropped images of gels/blots are acceptable, but need to be accompanied by size markers, and to retain visible background signal within the linear range (i.e. should not be saturated). The boundaries of panels with low background have to be demarked with black lines. Splicing of panels should only be considered if unavoidable, and must be clearly marked on the figure, and noted in the legend with a statement on whether the samples were obtained and processed simultaneously. Quantitative comparisons between samples on different gels/blots are discouraged; if this is unavoidable, it should only be performed for samples derived from the same experiment with gels/blots were processed in parallel, which needs to be stated in the legend.

Figures should be provided at approximately the size that they are to be printed at (single column is 86 mm, double column is 170 mm) and should not exceed an A4 page (8.5 x 11"). Reduction to the scale that will be used on the page is not necessary, but multi-panel figures should be sized so that the whole figure can be reduced by the same amount at the smallest size at which essential details in each panel are visible. In the interest of our colour-blind readers we ask that you avoid using red and green for contrast in figures. Replacing red with magenta and green with turquoise are two possible colour-safe

alternatives. Lines with widths of less than 1 point should be avoided. Sans serif typefaces, such as Helvetica (preferred) or Arial should be used. All text that forms part of a figure should be rewritable and removable.

The total number of Supplementary Figures (not including the “unprocessed scans” Supplementary Figure) should not exceed the number of main display items (figures and/or tables (see our Guide to Authors and March 2012 editorial <http://www.nature.com/ncb/authors/submit/index.html#suppinfo>; <http://www.nature.com/ncb/journal/v14/n3/index.html#ed>). No restrictions apply to Supplementary Tables or Videos, but we advise authors to be selective in including supplemental data.

Each Supplementary Figure should be provided as a single page and as an individual file in one of our

accepted figure formats and should be presented according to our figure guidelines (see above). Supplementary Tables should be provided as individual Excel files. Supplementary Videos should be provided as .avi or .mov files up to 50 MB in size. Supplementary Figures, Tables and Videos must be accompanied by a separate Word document including titles and legends.

GUIDELINES FOR EXPERIMENTAL AND STATISTICAL REPORTING

REPORTING REQUIREMENTS – We are trying to improve the quality of methods and statistics reporting in our papers. To that end, we are now asking authors to complete a reporting summary that collects information on experimental design and reagents. The Reporting Summary can be found here <https://www.nature.com/documents/nr-reporting-summary.pdf> If you would like to reference the guidance text as you complete the template, please access these flattened versions at <http://www.nature.com/authors/policies/availability.html>.

We strongly recommend the presentation of source data for graphical and statistical analyses as a separate Supplementary Table, and request that source data for all independent repeats are provided when representative experiments of multiple independent repeats, or averages of two independent experiments are presented. This supplementary table should be in Excel format, with data for different figures provided as different sheets within a single Excel file. It should be labelled and numbered as one

of the supplementary tables, titled "Statistics Source Data", and mentioned in all relevant figure legends.

Our flexible approach during the COVID-19 pandemic

If you need more time at any stage of the peer-review process, please do let us know. While our systems will continue to remind you of the original timelines, we aim to be as flexible as possible during the current pandemic.

This email has been sent through the Springer Nature Tracking System NY-610A-NPG&MTS

Confidentiality Statement:

This e-mail is confidential and subject to copyright. Any unauthorised use or disclosure of its contents is prohibited. If you have received this email in error please notify our Springer Nature Tracking System Helpdesk team at <http://platformsupport.nature.com>.

Details of the confidentiality and pre-publicity policy may be found here <http://www.nature.com/authors/policies/confidentiality.html>.

Privacy Policy | Update Profile

Decision Letter, first revision:

15th March 2022

Dear Dr. Rousseau,

Thank you for submitting your revised manuscript "Actin remodelling controls proteasome homeostasis upon stress" (NCB-R46190A). It has now been seen by the original referees and their comments are below. The reviewers find that the paper has improved in revision, and therefore we'll be happy in principle to publish it in Nature Cell Biology, pending minor revisions to satisfy the referees' final requests and to comply with our editorial and formatting guidelines.

In particular, we ask for inclusion of the LatA results in addition to those with LatB (please see comments from Reviewer #3). We would also require a toning down of claims of Adc17 mRNA transport along actin cables in both the text and abstract (Reviewer #3).

39If the current version of your manuscript is in a PDF format, please email us a copy of the file in an editable format (Microsoft Word or LaTeX)-- we can not proceed with PDFs at this stage.

Thank you again for your interest in Nature Cell Biology Please do not hesitate to contact me if you have any questions.

Sincerely,
Daryl

Daryl J.V. David, PhD

Senior Editor, Nature Cell Biology
Consulting Editor, Nature Communications
Nature Portfolio

Heidelberger Platz 3, 14197 Berlin, Germany
Email: daryl.david@nature.com
ORCID: <https://orcid.org/0000-0002-9253-4805>

Reviewer #1 (Remarks to the Author):

I was already quite enthusiastic about the original submission, and the changes in the revised manuscript adequately address my relatively minor concerns and, in my opinion, those of the other referees. I believe this can be published in NCB without further revision.

Reviewer #2 (Remarks to the Author):

I thought that the authors addressed my concerns in a satisfactory manner. To this end, I have no further concerns and I congratulate the authors on this very interesting study.

Sincerely

I/Topisirovic

Reviewer #3 (Remarks to the Author):

We thank the authors for their detailed response to questions on LatB versus LatA, and their new experiments shown in the response. We do think, however, that the LatA data are interesting and important, because they show that the actin itself in the patch is not necessary for Ede1's effect on translation. The results are still highly impactful, even more so in fact, with this new result.

One more comment to improve the work. The data in 5d (and movies 5 and 6) are described as showing that Adc17 mRNA "is frequently transported along actin cables". It is very difficult to make a clear assessment of that from the information shown. Both movies show very stop-and-start movement, if any, because the cables move around a lot. The authors should reduce their emphasis on movement here. In addition, it is unclear whether the mRNA is moving toward or away from the bud. The answer to this question would tell us a lot: if toward the bud, it is probably myosin V-mediated. If away from the bud, it is probably mediated by retrograde flow. Overall, the best would be for the authors to reduce their statement to something like "These data might suggest movement of Adc17 mRNA along cables, although clear determination of direction and mechanism of movement remain to be determined."

7th April 2022

Dear Dr. Rousseau,

Thank you for your patience as we've prepared the guidelines for final submission of your Nature Cell Biology manuscript, "Actin remodelling controls proteasome homeostasis upon stress" (NCB-R46190A). Please carefully follow the step-by-step instructions provided in the attached file, and add a response in each row of the table to indicate the changes that you have made. Please also check and comment on any additional marked-up edits we have proposed within the text. Ensuring that each point is addressed will help to ensure that your revised manuscript can be swiftly handed over to our production team.

We would like to start working on your revised paper, with all of the requested files and forms, as soon as possible (preferably within one week). Please get in contact with us if you anticipate delays.

In recognition of the time and expertise our reviewers provide to Nature Cell Biology's editorial process, we would like to formally acknowledge their contribution to the external peer review of your manuscript entitled "Actin remodelling controls proteasome homeostasis upon stress". For those reviewers who give their assent, we will be publishing their names alongside the published article.

41Nature Cell Biology offers a Transparent Peer Review option for new original research manuscripts submitted after December 1st, 2019. As part of this initiative, we encourage our authors to support increased transparency into the peer review process by agreeing to have the reviewer comments, author rebuttal letters, and editorial decision letters published as a Supplementary item. When you submit your final files please clearly state in your cover letter whether or not you would like to participate in this initiative. Please note that failure to state your preference will result in delays in accepting your manuscript for publication.

Cover suggestions

As you prepare your final files we encourage you to consider whether you have any images or illustrations that may be appropriate for use on the cover of Nature Cell Biology.

Nature Cell Biology has now transitioned to a unified Rights Collection system which will allow our Author Services team to quickly and easily collect the rights and permissions required to publish your work. Approximately 10 days after your paper is formally accepted, you will receive an email in providing you with a link to complete the grant of rights. If your paper is eligible for Open Access, our Author Services team will also be in touch regarding any additional information that may be required to arrange payment for your article.

Please note that *Nature Cell Biology* is a Transformative Journal (TJ). Authors may publish their research with us through the traditional subscription access route or make their paper immediately open access through payment of an article-processing charge (APC). Authors will not be required to make a final decision about access to their article until it has been accepted. Find out more about Transformative Journals

Authors may need to take specific actions to achieve compliance with funder and institutional open access mandates. If your research is supported by a funder that requires immediate open access (e.g. according to Plan S principles) then you should select the gold OA route, and we will direct you to the compliant route where possible. For authors selecting the subscription

42publication route, the journal's standard licensing terms will need to be accepted, including self-archiving policies. Those licensing terms will supersede any other terms that the author or any third party may assert apply to any version of the manuscript.

Please use the following link for uploading these materials:
[REDACTED]

Best regards,

Nyx Hills
Staff
Nature Cell Biology

On behalf of

Daryl J.V. David, PhD

Senior Editor, Nature Cell Biology
Consulting Editor, Nature Communications
Nature Portfolio

Heidelberger Platz 3, 14197 Berlin, Germany
Email: daryl.david@nature.com
ORCID: <https://orcid.org/0000-0002-9253-4805>

Reviewer #1:

Remarks to the Author:

I was already quite enthusiastic about the original submission, and the changes in the revised manuscript adequately address my relatively minor concerns and, in my opinion, those of the other

43referees. I believe this can be published in NCB without further revision.

Reviewer #2:

Remarks to the Author:

I thought that the authors addressed my concerns in a satisfactory manner. To this end, I have no further concerns and I congratulate the authors on this very interesting study.

Sincerely

I/Topisirovic

Reviewer #3:

Remarks to the Author:

We thank the authors for their detailed response to questions on LatB versus LatA, and their new experiments shown in the response. We do think, however, that the LatA data are interesting and important, because they show that the actin itself in the patch is not necessary for Ede1's effect on translation. The results are still highly impactful, even more so in fact, with this new result.

One more comment to improve the work. The data in 5d (and movies 5 and 6) are described as showing that Adc17 mRNA "is frequently transported along actin cables". It is very difficult to make a clear assessment of that from the information shown. Both movies show very stop-and-start movement, if any, because the cables move around a lot. The authors should reduce their emphasis on movement here. In addition, it is unclear whether the mRNA is moving toward or away from the bud. The answer to this question would tell us a lot: if toward the bud, it is probably myosin V-mediated. If away from the bud, it is probably mediated by retrograde flow. Overall, the best would be for the authors to reduce their statement to something like "These data might suggest movement of Adc17 mRNA along cables, although clear determination of direction and mechanism of movement remain to be determined."

Author Rebuttal, first revision:

We would like to thank the editors for their time and valuable remarks. As described hereafter, all points have been addressed separately in the point-by-point rebuttal.

Reviewer #1 (Remarks to the Author):

I was already quite enthusiastic about the original submission, and the changes in the revised

44manuscript adequately address my relatively minor concerns and, in my opinion, those of the other referees. I believe this can be published in NCB without further revision.

We thank the reviewer for his/her decision.

Reviewer #2 (Remarks to the Author):

I thought that the authors addressed my concerns in a satisfactory manner. To this end, I have no further concerns and I congratulate the authors on this very interesting study.

We thank the reviewer for his/her decision.

Reviewer #3 (Remarks to the Author):

We thank the authors for their detailed response to questions on LatB versus LatA, and their new experiments shown in the response. We do think, however, that the LatA data are interesting and important, because they show that the actin itself in the patch is not necessary for Ede1's effect on translation. The results are still highly impactful, even more so in fact, with this new result.

We agree with the reviewer and Lat-A data have been added to the manuscript at lines 210-212:" Moreover, actin nucleation at the surface of cortical actin patches was not required for RPAC translation. Latrunculin-A treatment, which disrupts actin at patches as well as cables, had similar effects to that of Lat-B (Extended data Fig. 4c-e).".

One more comment to improve the work. The data in 5d (and movies 5 and 6) are described as showing that Adc17 mRNA "is frequently transported along actin cables". It is very difficult to make a clear assessment of that from the information shown. Both movies show very stop-and-start movement, if any, because the cables move around a lot. The authors should reduce their emphasis on movement here. In addition, it is unclear whether the mRNA is moving toward or away from the bud. The answer to this question would tell us a lot: if toward the bud, it is probably myosin V-mediated. If away from the bud, it is probably mediated by retrograde flow. Overall, the best would be for the authors to reduce

45their statement to something like “These data might suggest movement of Adc17 mRNA along cables, although clear determination of direction and mechanism of movement remain to be determined.”

We agree with the reviewer’s concern, and we have reduced our statement on the traveling of ADC17 mRNA along actin cables. See modifications at lines:

- 15-16: “mRNA of the RPAC ADC17 is associated with actin cables and is enriched at cortical actin patches under stress”.
- 180-185: “We observed that ADC17 mRNA is often associated with actin cables *in-vivo* (Fig. 5d and Supplementary Video 5,6), while its interaction with patches is more transient and dynamic, as previously observed for Ede1 (Extended data Fig. 3c and Supplementary Video 7,8). Overall, these data might suggest movement of Adc17 mRNA along actin cables, although clear determination of direction and mechanism of movement remain to be determined.”.
- 279-280: “Here we have shown that Adc17 RPAC mRNA may be transported on actin cables and interacts with cortical actin patches”.
- 828-829: “Representative microscopy images from time-lapse imaging showing ADC17 mRNA interaction with actin cable.”.

We thank the reviewers for their careful reading of the manuscript and their constructive remarks. We have taken all the comments into consideration to strengthen and clarify the manuscript. We feel that the manuscript has been significantly improve as a consequence.

Final Decision Letter:

Dear Dr Rousseau,

I am pleased to inform you that your manuscript, "Actin remodelling controls proteasome homeostasis upon stress", has now been accepted for publication in Nature Cell Biology.

We do apologize that the public availability of your dataset on the ProteomeXchange was delayed.

Over the next few weeks, your paper will be copyedited to ensure that it conforms to Nature Cell

46Biology style. Once your paper is typeset, you will receive an email with a link to choose the appropriate publishing options for your paper and our Author Services team will be in touch regarding any additional information that may be required.

Please note that *Nature Cell Biology* is a Transformative Journal (TJ). Authors may publish their research with us through the traditional subscription access route or make their paper immediately open access through payment of an article-processing charge (APC). Authors will not be required to make a final decision about access to their article until it has been accepted. Find out more about Transformative Journals

If you have not already done so, we strongly recommend that you upload the step-by-step protocols used in this manuscript to the Protocol Exchange (www.nature.com/protocolexchange), an open online resource established by Nature Protocols that allows researchers to share their detailed experimental know-how. All uploaded protocols are made freely available, assigned DOIs for ease of citation and are fully searchable through nature.com. Protocols and Nature Portfolio journal papers in which they are used can be linked to one another, and this link is clearly and prominently visible in the online versions of both papers. Authors who performed the specific experiments can act as primary authors for the Protocol as they will be best placed to share the methodology details, but the Corresponding Author of the present research paper should be included as one of the authors. By uploading your Protocols to Protocol Exchange, you are enabling researchers to more readily reproduce or adapt the methodology you use, as well as increasing the visibility of your protocols and papers. You can also establish a dedicated page to collect your lab Protocols. Further information can be found at www.nature.com/protocolexchange/about

With kind regards,
Daryl

Daryl J.V. David, PhD

Senior Editor, Nature Cell Biology
Consulting Editor, Nature Communications
Nature Portfolio

Heidelberger Platz 3, 14197 Berlin, Germany
Email: daryl.david@nature.com
ORCID: <https://orcid.org/0000-0002-9253-4805>

** Visit the Springer Nature Editorial and Publishing website at www.springernature.com/editorial-and-publishing-jobs for more information about our career opportunities. If you have any questions please click here.**